# Statistical Optimality of Newton-type Federated Learning with Heterogeneous Data

## Abstract

The mainstream federated learning algorithms only communicate the first-order information across the local devices, i.e., FedAvg and FedProx. However, only using first-order information, these methods are often inefficient and the impact of heterogeneous data is yet not precisely understood. This paper proposes an efficient federated Newton method (FedNewton), by sharing both first-order and second-order knowledge over heterogeneous data. In general kernel ridge regression setting, we derive the generalization bounds for FedNewton and obtain the minimax-optimal learning rates. For the first time, our results analytically quantify the impact of the number of local examples, the data heterogeneity and the model heterogeneity. Moreover, as long as the local sample size is not too small and data heterogeneity is moderate, the federated error in FedNewton decreases exponentially in terms of iterations. Extensive experimental results further validate our theoretical findings and illustrate the advantages of FedNewton over the first-order methods.

## 1 Introduction

Owing to the great potential in privacy preservation and in lowering the computational costs, federated learning (FL) McMahan et al. (2017); Li et al. (2020a); Wang et al. (2025) becomes a promising framework in processing large-scale tasks. However, federated learning is facing massive challenges from the heterogeneous data Zhou et al. (2023); Chen et al. (2025), including both the data heterogeneity and the model heterogeneity. The data heterogeneity comes from that inputs across devices are usually sampled from heterogeneous distributions, while the model heterogeneity measures the response shift due to inconsistency between local models and the global model.

First-order approaches, including FedAvg McMahan et al. (2017) and FedProx Li et al. (2020a), share the first-order information rather than the data across devices and tolerate the heterogeneity in federated learning, while Newton-type FL methods Ghosh et al. (2020); Gupta et al. (2021); Safaryan et al. (2022); Islamov et al. (2023); Liu et al. (2023); Dal Fabbro et al. (2024); Li et al. (2024) utilized second-order information for updating federated model. To the best of our knowledge, most of existing learning guarantees for FL methods are derived in the context of optimization and focused on in-sample predictive errors only, i.e., the convergence analysis (optimization) of first-order FL Li et al. (2020b); Karimireddy et al. (2020); Pathak & Wainwright (2020); Glasgow et al. (2022) and Newton-type FL Ghosh et al. (2020); Safaryan et al. (2022); Qian et al. (2022); Elgabli et al. (2022); Elbakary et al. (2024); Hamidi & Ye (2025). However, beyond the optimization, the generalization guarantees (out-sample predictive performance) are of great practical and theoretical interests for FL. Despite recent efforts and progress on the generalization for first-order algorithms Mohri et al. (2019); Yagli et al. (2020); Su et al. (2021); Yuan et al. (2022), the generalization guarantees for Newton-type FL algorithms remain elusive, especially on heterogeneous data and localized models. Therefore, a challenging problem in FL is *how to quantify the impact of heterogeneity from the generalization perspective?*

In this paper, motivated by sharing second-order information, we propose a second-order federated optimization method, named `FedNewton`. It approximates the global predictor on the entire data by utilizing the global gradient and local Hessians, improving the predictive accuracy in an efficient communications framework. We then study the statistical properties of `FedNewton`, and derive the generalization bounds with the minimax optimal rates. We conclude with experiments on simulated

data and publicly available tasks that complement our theoretical results, exhibiting the computational and statistical benefits of our approach. Due to the length limit, we leave the experiment part in the appendix. We summarize our contributions as below:

**1) On the algorithmic front.** We propose a fast second-order federated learning algorithm, which improves the approximation of the centralized model while only requiring similar computational and communication costs as the first-order methods. The convergence of `FedNewton` is exponentially fast and a few communications, for example, $t \leq 2$, can approximate the global model well.

**2) On the statistical front.** To our best knowledge, in presence of both data heterogeneity and model heterogeneity, we present the optimal generalization guarantees for the first time. Our results further analytically quantify the impacts of the local sample size, the data heterogeneity, and the model heterogeneity. Especially, the federated error decreases exponentially fast in benign cases, i.e., a sufficient number of local examples and moderate data heterogeneity.

## 2 PROBLEM SETUP

In a standard framework of federated learning, there is a global parameter server and $m$ local computational clients. On the $j$-th local machine $\forall j \in [m]$, the local data $\mathcal{D}_j = \{(\boldsymbol{x}_{ij}, y_{ij})\}_{i=1}^{|\mathcal{D}_j|}$ is drawn from a local distribution $\rho_j$ on the joint space $\mathcal{X} \times \mathcal{Y}$. The total sample $\mathcal{D} = \bigcup_{j=1}^{m} \mathcal{D}_j$ is the disjoint union of local data and corresponds to a global distribution $\rho$. For any local devices $j, k \in [m]$ and $j \neq k$, data distributions are identical $\rho_j = \rho_k = \rho$ in the homogeneous setting (iid data), while data distributions are distinct $\rho_j \neq \rho_k$ in the heterogeneous case (non-iid data).

We base our analysis on the standard non-parametric regression setup and assume that the target solution $f^*$ belongs to a reproducing kernel Hilbert space (RKHS) induced by a Mercer kernel $K : \mathcal{X} \times \mathcal{X} \to \mathbb{R}$. Mercer's theorem guarantees the kernel function admits an implicit feature mapping $K(\boldsymbol{x}, \boldsymbol{x}') = \langle \phi(\boldsymbol{x}), \phi(\boldsymbol{x}') \rangle_K$ and the norm by $\| \cdot \|_K$. The predictor can be stated as $f_{\mathcal{D}, \lambda}(\boldsymbol{x}) = \langle \boldsymbol{w}_{\mathcal{D}, \lambda}, \phi(\boldsymbol{x}) \rangle$ where $\boldsymbol{w}_{\mathcal{D}, \lambda}$ minimizes the objective on the entire data $\mathcal{D}$

$$\underset{\boldsymbol{w} \in \mathcal{H}_K}{\arg\min} \left\{ \frac{1}{2|\mathcal{D}|} \sum_{i=1}^{|\mathcal{D}|} (f(\boldsymbol{x}_i) - y_i)^2 + \frac{\lambda}{2} \|\boldsymbol{w}\|_K^2 \right\}, \tag{1}$$

where $(\boldsymbol{x}_i, y_i) \in \mathcal{D}$, and $\lambda > 0$ is the regularity parameter. The above regression problem, known as Kernel Ridge Regression (KRR), admits a closed-form solution

$$\boldsymbol{w}_{\mathcal{D}, \lambda} = (\boldsymbol{\Phi}_{\mathcal{D}}^{\top} \boldsymbol{\Phi}_{\mathcal{D}} + \lambda I)^{-1} \boldsymbol{\Phi}_{\mathcal{D}}^{\top} \boldsymbol{y}_{\mathcal{D}}, \tag{2}$$

where $\boldsymbol{\Phi}_{\mathcal{D}} = \frac{1}{\sqrt{|\mathcal{D}|}} \left[ \phi(\boldsymbol{x}_1), \cdots, \phi(\boldsymbol{x}_{|\mathcal{D}|}) \right]^T \in \mathbb{R}^{|\mathcal{D}|} \times \mathcal{H}_K$ are feature mappings on the training set $\mathcal{D}$ and $\boldsymbol{y}_{\mathcal{D}} = \frac{1}{\sqrt{|\mathcal{D}|}} \left( y_1, \cdots, y_{|\mathcal{D}|} \right)^{\top}$ are the corresponding labels.

By averaging the local models, the simplest federated method only communicates once, known as Distributed Kernel Ridge Regression (DKRR) with the closed-form solution

$$\bar{\boldsymbol{w}}_{\mathcal{D}, \lambda} = \sum_{j=1}^{m} p_j (\boldsymbol{\Phi}_{\mathcal{D}_j}^{\top} \boldsymbol{\Phi}_{\mathcal{D}_j} + \lambda I)^{-1} \boldsymbol{\Phi}_{\mathcal{D}_j}^{\top} \boldsymbol{y}_{\mathcal{D}_j},$$

where $p_j$ is the weight of the $j$-th local model, which is usually set $p_j = |\mathcal{D}_j|/|\mathcal{D}|$. Note that, $\boldsymbol{\Phi}_{\mathcal{D}_j} = \frac{1}{\sqrt{|\mathcal{D}_j|}} \left[ \phi(\boldsymbol{x}_1), \cdots, \phi(\boldsymbol{x}_{|\mathcal{D}_j|}) \right]^T \in \mathbb{R}^{|\mathcal{D}_j|} \times \mathcal{H}_K$ are local feature mappings and $\boldsymbol{y}_{\mathcal{D}_j} = \frac{1}{\sqrt{|\mathcal{D}_j|}} \left( y_1, \cdots, y_{|\mathcal{D}_j|} \right)^{\top}$ are labels on the $j$-th local train set $\mathcal{D}_j = \left\{ (\boldsymbol{x}_{ij}, y_{ij}) \right\}_{i=1}^{|\mathcal{D}_j|}, \quad \forall j \in [m]$.

The solution of KRR equation 2 can be rewritten in the Newton's method form

$$\boldsymbol{w}_{\mathcal{D}, \lambda} = \boldsymbol{w} - \boldsymbol{H}_{\mathcal{D}, \lambda}^{-1} \boldsymbol{g}_{\mathcal{D}, \lambda}. \tag{3}$$

where the gradient and Hessian matrix are defined as

$$\boldsymbol{g}_{\mathcal{D}, \lambda} := (\boldsymbol{\Phi}_{\mathcal{D}}^{\top} \boldsymbol{\Phi}_{\mathcal{D}} + \lambda I)\boldsymbol{w} - \boldsymbol{\Phi}_{\mathcal{D}}^{\top} \boldsymbol{y}_{\mathcal{D}},$$

$$\boldsymbol{H}_{\mathcal{D}, \lambda} := (\boldsymbol{\Phi}_{\mathcal{D}}^{\top} \boldsymbol{\Phi}_{\mathcal{D}} + \lambda I).$$

---

**Algorithm 1** Federated Learning with Newton Method (`FedNewton`)

---

**Input:** Local training data subset $\mathcal{D}_j$, $\forall j \in [m]$. Feature mapping $\phi : \boldsymbol{X} \to \mathbb{R}^M$.

**Output:** The global estimator $\bar{\boldsymbol{w}}_{\mathcal{D},\lambda}^T$.

1: **Local machines:** Compute feature mapping $\boldsymbol{\Phi}_{\mathcal{D}_j}$, $\boldsymbol{H}_{\mathcal{D}_j,\lambda} = (\boldsymbol{\Phi}_{\mathcal{D}_j}^\top \boldsymbol{\Phi}_{\mathcal{D}_j} + \lambda I)$, $\boldsymbol{H}_{\mathcal{D}_j,\lambda}^{-1}$ and $\boldsymbol{\Phi}_{\mathcal{D}_j}^\top \boldsymbol{y}_{\mathcal{D}_j}$ for any $j \in [m]$.

2: **Local machines:** Initialize the local estimators by $\boldsymbol{w}_{\mathcal{D}_j,\lambda}^0 = \boldsymbol{H}_{\mathcal{D}_j,\lambda}^{-1} \boldsymbol{\Phi}_{\mathcal{D}_j}^\top \boldsymbol{y}_{\mathcal{D}_j}$ and upload them to the global server ($\uparrow$).

3: **Global server:** Initialize the solution by $\bar{\boldsymbol{w}}_{\mathcal{D},\lambda}^0 = \sum_{j=1}^m p_j \boldsymbol{w}_{\mathcal{D}_j,\lambda}^0$, and send it to the local nodes ($\downarrow$).

4: **for** $t = 1$ to $T$ **do**

5:     **Local machines:** Compute local gradients $\boldsymbol{g}_{\mathcal{D}_j,\lambda}^{t-1} = \boldsymbol{H}_{\mathcal{D}_j,\lambda} \bar{\boldsymbol{w}}_{\mathcal{D},\lambda}^{t-1} - \boldsymbol{\Phi}_{\mathcal{D}_j}^\top \boldsymbol{y}_{\mathcal{D}_j}$ and upload them to global server ($\uparrow$).

6:     **Global server:** Compute the global gradient $\boldsymbol{g}_{\mathcal{D},\lambda}^{t-1} = \sum_{j=1}^m p_j \boldsymbol{g}_{\mathcal{D}_j,\lambda}^{t-1}$ and send it to local nodes ($\downarrow$).

7:     **Local machines:** Compute the local updates $\boldsymbol{H}_{\mathcal{D}_j,\lambda}^{-1} \boldsymbol{g}_{\mathcal{D},\lambda}^{t-1}$ and upload it to the global server ($\uparrow$).

8:     **Global server:** Update the global estimator $\bar{\boldsymbol{w}}_{\mathcal{D},\lambda}^t = \bar{\boldsymbol{w}}_{\mathcal{D},\lambda}^{t-1} - \sum_{j=1}^m p_j \boldsymbol{H}_{\mathcal{D}_j,\lambda}^{-1} \boldsymbol{g}_{\mathcal{D},\lambda}^{t-1}$ and communicate it to local machines ($\downarrow$).

9: **end for**

---

From equation 3, the global gradient $\boldsymbol{g}_{\mathcal{D},\lambda}$ and Hessian $\boldsymbol{H}_{\mathcal{D},\lambda}$ is the key to achieving the centralized model $\boldsymbol{w}_{\mathcal{D},\lambda}$. Note that, since the fact $\boldsymbol{\Phi}_{\mathcal{D}}^\top \boldsymbol{\Phi}_{\mathcal{D}} = \sum_{j=1}^m p_j \boldsymbol{\Phi}_{\mathcal{D}_j}^\top \boldsymbol{\Phi}_{\mathcal{D}_j}$ for data partition $\mathcal{D} = \bigcup_{j=1}^m \mathcal{D}_j$, one can easily obtain the following property for the global gradient and global Hessian.

**Proposition 1** (Partitonability). *If the loss is squared loss, the global gradient and Hessian matrix consist of the local ones, i.e. $\boldsymbol{g}_{\mathcal{D},\lambda} = \sum_{j=1}^m p_j \boldsymbol{g}_{\mathcal{D}_j,\lambda}$ and $\boldsymbol{H}_{\mathcal{D},\lambda} = \sum_{j=1}^m p_j \boldsymbol{H}_{\mathcal{D}_j,\lambda}$.*

**Remark 1** (Computation of local inverse Hessian). *The compute of the inverse of local Hessians $\boldsymbol{H}_{\mathcal{D}_j,\lambda}^{-1}$ is time consuming $\mathcal{O}(|\mathcal{D}_j|M^2 + M^3)$, which is a common problem in second-order optimization Bottou et al. (2018). There are many classic work to reduce the time complexity of the inverse of Hessian, i.e. BFGS Broyden (1970), L-BFGS Liu & Nocedal (1989), inexact Newton Dembo et al. (1982), Gauss-Newton Schraudolph (2002) and Newton sketch Pilanci & Wainwright (2017). Those techniques can be used to improve the efficiency of `FedNewton`, but it is beyond the scope of this paper. We focus on theoretical novelties and leave further computational improvements in the future.*

**Remark 2** (Feature mapping instead of kernel methods). *Without loss of generality, we assume the feature mappings are finite dimensional $\phi : \boldsymbol{X} \to \mathbb{R}^M$, which covers a wide range of generalized linear models, for example neural networks Neal (1995); Jacot et al. (2018), kernel methods Vapnik (2000), random features Rahimi & Recht (2007); Le et al. (2013); Yang et al. (2014), and random sketching Woodruff et al. (2014); Yang et al. (2017).*

## 3   FEDERATED LEARNING WITH NEWTON METHOD

Motivated by recent gradient-based distributed learning Wang et al. (2018); Lin et al. (2020), we propose a Newton-type federated learning method to quantity the impact of data heterogeneity and model heterogeneity. Using Proposition 1, the exact Federated Newton's method communicate local Hessians $\boldsymbol{H}_{\mathcal{D}_j,\lambda}$ for computing the global Hessian matrix equation 3 whose the communication complexity is $\boldsymbol{O}(M^2)$, which is infeasible in federated learning. To reduce communication costs, we propose `FedNewton` that approximates the Newton's updates with the global gradient and local Hessian matrices, such that

$$\boldsymbol{H}_{\mathcal{D},\lambda}^{-1} \boldsymbol{g}_{\mathcal{D},\lambda} \approx \sum_{j=1}^m p_j \boldsymbol{H}_{\mathcal{D}_j,\lambda}^{-1} \boldsymbol{g}_{\mathcal{D},\lambda}. \tag{4}$$

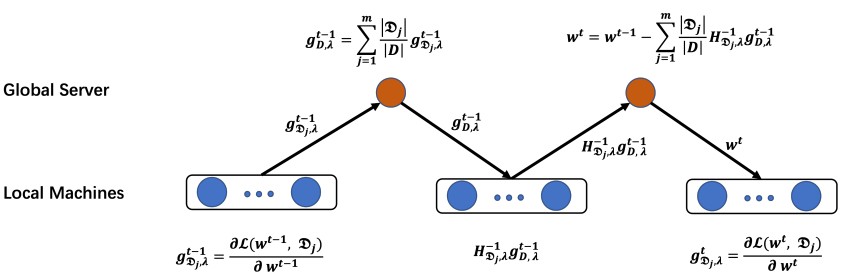

Figure 1: Computation and communication in the $t$-th iteration of `FedNewton`.

The global learner $\bar{f}^t_{\mathcal{D},\lambda}(\boldsymbol{x}) = \langle \bar{\boldsymbol{w}}^t_{\mathcal{D},\lambda}, \phi(\boldsymbol{x}) \rangle$ is updated by

$$\bar{\boldsymbol{w}}^t_{\mathcal{D},\lambda} = \bar{\boldsymbol{w}}^{t-1}_{\mathcal{D},\lambda} - \sum_{j=1}^{m} p_j \boldsymbol{H}^{-1}_{\mathcal{D}_j,\lambda} \boldsymbol{g}^{t-1}_{\mathcal{D},\lambda}, \tag{5}$$

where $\bar{\boldsymbol{w}}^t_{\mathcal{D},\lambda}$ is the model after $t$ iterations and the global gradient is $\boldsymbol{g}^{t-1}_{\mathcal{D},\lambda} = \sum_{j=1}^{m} p_j \boldsymbol{g}^{t-1}_{\mathcal{D}_j,\lambda}$ from Proposition 1. The approximation error between equation 3 and equation 5 is analyzed in Section 4. Without loss of generality, we present the details of `FedNewton` in Algorithm 1 and Figure 1, which includes two times communications as the first-order methods in per round. Note that, the algorithm uploads local Newton updates $\boldsymbol{H}^{-1}_{\mathcal{D}_j,\lambda} \boldsymbol{g}^{t-1}_{\mathcal{D},\lambda} \in \mathbb{R}^M$ instead of local inverse Hessians $\boldsymbol{H}^{-1}_{\mathcal{D}_j,\lambda} \in \mathbb{R}^{M \times M}$, reducing communication costs from $\boldsymbol{O}(M^2)$ to $\boldsymbol{O}(M)$.

**Computational complexity.** With a finite-dimensional feature map $\phi : \boldsymbol{X} \to \mathbb{R}^M$, we now characterize the time, space, and communication complexity of `FedNewton`. On local machine $j$, the space complexity is $\boldsymbol{O}(|\mathcal{D}_j|M + M^2)$ to store $\boldsymbol{\Phi}_{\mathcal{D}_j}$, $\boldsymbol{H}_{\mathcal{D}_j,\lambda}$, and $\boldsymbol{H}^{-1}_{\mathcal{D}_j,\lambda}$, while the server needs $\boldsymbol{O}(mM)$ space to store $\boldsymbol{g}_{\mathcal{D},\lambda}$ and $\boldsymbol{H}^{-1}_{\mathcal{D}_j,\lambda} \boldsymbol{g}_{\mathcal{D},\lambda}$. Before the iterative procedure, computing $\boldsymbol{H}_{\mathcal{D}_j,\lambda}$ and $\boldsymbol{H}^{-1}_{\mathcal{D}_j,\lambda}$ costs $\boldsymbol{O}(|\mathcal{D}_j|M^2 + M^3)$ time. In each iteration, the local time complexity is $\boldsymbol{O}(M^2)$ for computing the local gradient $\boldsymbol{g}_{\mathcal{D}_j,\lambda}$ and the local Newton update $\boldsymbol{H}^{-1}_{\mathcal{D}_j,\lambda} \boldsymbol{g}_{\mathcal{D},\lambda}$, while the server costs $\boldsymbol{O}(mM)$ time to update the global gradient and estimator. Hence the overall time complexity is

$$\boldsymbol{O}\Big( \max_{j \in [m]} |\mathcal{D}_j| M^2 + M^3 + M^2 t + mMt \Big).$$

**Remark 3** (Communication cost). *Each iteration of `FedNewton` requires roughly twice the communication of first-order FL methods such as FedAvg and FedProx, but it needs far fewer iterations. The per-round communication is $\boldsymbol{O}(M)$, so the total communication complexity is $\boldsymbol{O}(Mt)$, matching that of most first-order methods. By Theorem 1, the iteration complexity of `FedNewton` is* linear *in the error, i.e., $t = \Omega(\log(1/\epsilon))$ to achieve federated error $\epsilon$; thus `FedNewton` converges exponentially fast to the global estimator equation 2. In contrast, first-order federated algorithms typically require $t = \Omega(1/\epsilon)$ communication rounds Su et al. (2021). Therefore, while `FedNewton` does not reduce the* per-round *communication as in communication-compressed FL methods Sattler et al. (2019); Reisizadeh et al. (2020); Wu et al. (2022), it substantially reduces the number of communication rounds; in our experiments, `FedNewton` already achieves strong predictive performance with as few as $t \le 2$ rounds.*

### 3.1 FEDNEWTON FOR HEAD FINE-TUNING WITH A FROZEN BACKBONE

In many vision and language applications, large pretrained models (e.g., GPT-2, BERT, ViT) are readily available. A common practice is to freeze the feature layers as a backbone and only fine-tune a small linear head. We adopt this setting and present a `FedNewton` variant where the backbone is globally shared and frozen on all clients, and only a linear head on top of the backbone features is optimized in a federated manner.

---

**Algorithm 2** FedNewton for Head Fine-tuning with a Frozen Backbone

---

1: **Input:** Frozen backbone $\phi_{\text{backbone}}$; local datasets $\{\mathcal{D}_j\}_{j=1}^m$; number of rounds $T$; client fraction $q$; Newton step size $\eta$; damping parameter $\lambda$.
2: **Initialize:** Global head parameter $\bar{\boldsymbol{w}}_{\mathcal{D},\lambda}^0$ on the server.
3: **for** $t = 0$ to $T - 1$ **do**
4:     **Clients** ($j \in S_t$)**:** Compute the local gradient $\boldsymbol{g}_{\mathcal{D}_j,\lambda}^t$ and send $(\boldsymbol{g}_{\mathcal{D}_j,\lambda}^t, |\mathcal{D}_j|)$ to the server.
5:     **Server:** Compute the global gradient $\boldsymbol{g}_{\mathcal{D},\lambda}^t$ via equation 6 and broadcast $\boldsymbol{g}_{\mathcal{D},\lambda}^t$ (and optionally $\bar{\boldsymbol{w}}_{\mathcal{D},\lambda}^t$) to all $j \in S_t$.
6:     **Clients** ($j \in S_t$)**:** Compute the Newton step $\Delta\boldsymbol{w}_j^t$ via equation 7, and send $\Delta\boldsymbol{w}_j^t$ to the server.
7:     **Server:** Update the global head $\bar{\boldsymbol{w}}_{\mathcal{D},\lambda}^{t+1}$ via equation 8.
8: **end for**
9: **Output:** Final head parameter $\bar{\boldsymbol{w}}_{\mathcal{D},\lambda}^T$ on the frozen backbone $\phi_{\text{backbone}}$.

---

Let $\phi_{\text{backbone}}$ denote the parameters of a fixed pretrained backbone shared by all clients. For any input $\boldsymbol{x} \in \boldsymbol{X}$, its feature representation is $\phi(\boldsymbol{x}) = \phi_{\text{backbone}}(\boldsymbol{x}) \in \mathbb{R}^M$, where $M$ is the feature dimension. On these frozen features, we consider a linear model $\bar{f}_{\mathcal{D},\lambda}(\boldsymbol{x}) = \langle \bar{\boldsymbol{w}}_{\mathcal{D},\lambda}, \phi(\boldsymbol{x}) \rangle$, where $\bar{\boldsymbol{w}}_{\mathcal{D},\lambda} \in \mathbb{R}^M$ collects all trainable head parameters. For a twice differentiable loss $\ell(\cdot, \cdot)$ (e.g., cross-entropy), the loss on a sample $(\boldsymbol{x}, y)$ is $\ell\big(\bar{f}_{\mathcal{D},\lambda}(\boldsymbol{x}), y\big)$.

On local machine $j$ with dataset $\mathcal{D}_j = \big\{(\boldsymbol{x}_{ij}, y_{ij})\big\}_{i=1}^{|\mathcal{D}_j|}$, the empirical risk with respect to $\bar{\boldsymbol{w}}_{\mathcal{D},\lambda}$ is $F_j(\bar{\boldsymbol{w}}_{\mathcal{D},\lambda}) = \frac{1}{|\mathcal{D}_j|} \sum_{i=1}^{|\mathcal{D}_j|} \ell\big(\langle \bar{\boldsymbol{w}}_{\mathcal{D},\lambda}, \phi(\boldsymbol{x}_{ij}) \rangle, y_{ij}\big)$. At communication round $t$, all participating clients share the same head parameter $\bar{\boldsymbol{w}}_{\mathcal{D},\lambda}^t$. Each machine $j$ computes its local gradient $\boldsymbol{g}_{\mathcal{D}_j,\lambda}^t := \nabla_{\bar{\boldsymbol{w}}_{\mathcal{D},\lambda}} F_j(\bar{\boldsymbol{w}}_{\mathcal{D},\lambda}^t)$, and uploads $\boldsymbol{g}_{\mathcal{D}_j,\lambda}^t$ and the local sample size $|\mathcal{D}_j|$ to the server. Let $p_j = |\mathcal{D}_j| / \sum_{r=1}^m |\mathcal{D}_j|$. The server then computes the sample-size–weighted global gradient

$$\boldsymbol{g}_{\mathcal{D},\lambda}^t = \sum_{j=1}^m p_j \, \boldsymbol{g}_{\mathcal{D}_j,\lambda}^t, \tag{6}$$

and broadcasts $\boldsymbol{g}_{\mathcal{D},\lambda}^t$ (and optionally $\bar{\boldsymbol{w}}_{\mathcal{D},\lambda}^t$) to all $j \in S_t$.

Based on the global gradient and local Hessian, `FedNewton` instructs clients to compute an exact local Newton step via matrix inversion or an approximate local Newton step via the *Conjugate Gradient (CG)* algorithm to solve a linear system

$$\Delta\boldsymbol{w}_j^t = \big(\boldsymbol{H}_{\mathcal{D}_j,\lambda}^t\big)^{-1} \boldsymbol{g}_{\mathcal{D},\lambda}^t \quad \textit{(exact)}, \qquad \text{or} \qquad \boldsymbol{H}_{\mathcal{D}_j,\lambda}^t \Delta\boldsymbol{w}_j^t = \boldsymbol{g}_{\mathcal{D},\lambda}^t \quad \textit{(CG)} \tag{7}$$

To handle non-convexity and improve stability, the local Hessian is regularized via Tikhonov damping, $\boldsymbol{H}_{\mathcal{D}_j,\lambda}^t = \nabla_{\bar{\boldsymbol{w}}_{\mathcal{D},\lambda}}^2 F_j(\bar{\boldsymbol{w}}_{\mathcal{D},\lambda}^t) + \lambda I$ with $\lambda > 0$. CG method is a matrix-free approach that relies solely on Hessian-vector products, enabling high-precision second-order updates with small memory overhead and zero communications of Hessian matrices.

Given a global step size $\eta > 0$, clients send $\Delta\boldsymbol{w}_j^t$ back to the server and perform the aggregation

$$\bar{\boldsymbol{w}}_{\mathcal{D},\lambda}^{t+1} = \bar{\boldsymbol{w}}_{\mathcal{D},\lambda}^t - \eta \sum_{j=1}^m p_j \Delta\boldsymbol{w}_j^t. \tag{8}$$

Throughout training, the backbone $\phi_{\text{backbone}}$ remains frozen and is neither updated nor communicated. We summarize the protocol in Algorithm 2. It mirrors the RKHS-based `FedNewton` in Algorithm 1, but confines second-order computations to the finite-dimensional head while using a fixed pretrained backbone as the feature map.

**Remark 4.** *Compared to RF-based `FedNewton`, this variant replaces the random feature map with the pretrained backbone $\phi_{\text{backbone}}(\boldsymbol{x})$ and restricts Newton-type updates to the linear head $\bar{\boldsymbol{w}}_{\mathcal{D},\lambda}$. This design keeps second-order computations lightweight, while the strong backbone features ensure competitive accuracy. As shown in Section C, leveraging a powerful frozen backbone allows `FedNewton` to outperform many first-order FL baselines with only a few rounds, aligning well with the pretrain–then–fine-tune paradigm.*

# 4 MAIN RESULTS

In this section, to explore the factors that affect performance, we derive the excess risk bounds for `FedNewton` in homogeneous settings and heterogeneous settings, respectively.

## 4.1 NOTATIONS AND ASSUMPTIONS

We consider a broader scenario for federated learning, where the local training sets contain both heterogenous inputs (covariate shift) $\mathcal{D}_j \sim \rho_j$ and different responses (concept shift) $\boldsymbol{y}_{\mathcal{D}_j} \sim \rho_j(y|\boldsymbol{x})$. The concept shift is represented as

$$f^*(\boldsymbol{x}) = \int_{\mathcal{Y}} y d\rho(y|\boldsymbol{x}), \ \boldsymbol{x} \in \mathcal{X}, \qquad f_j^*(\boldsymbol{x}) = \int_{\mathcal{Y}} y d\rho_j(y|\boldsymbol{x}), \ \boldsymbol{x} \in \mathcal{X}, \ j \in [m], \tag{9}$$

where $f_j^*$ is the underlying mechanism governing the true responses on the $j$-th worker. Give a $\boldsymbol{x} \in \boldsymbol{X}$ and $j, k, \in [m]$, the responses may be different $f_j^*(\boldsymbol{x}) \neq f_k^*(\boldsymbol{x})$ when $j \neq k$.

**Definition 1** (Operators with feature mapping $\phi$). *Using the feature mapping $\phi : \boldsymbol{X} \to \mathcal{H}_K$, $\forall \boldsymbol{\beta} \in \mathcal{H}_K$, the covariance operators $C, C_j, C_{\mathcal{D}}, C_{\mathcal{D}_j} : \mathcal{H}_K \to \mathcal{H}_K$ are defined as*

$$C\boldsymbol{\beta} = \int_X \langle \boldsymbol{\beta}, \phi(\boldsymbol{x}) \rangle \phi(\boldsymbol{x}) d\rho_X(\boldsymbol{x}), \qquad C_{\mathcal{D}}\boldsymbol{\beta} = \frac{1}{|\mathcal{D}|} \sum_{i=1}^{|\mathcal{D}|} \langle \boldsymbol{\beta}, \phi(\boldsymbol{x}_i) \rangle \phi(\boldsymbol{x}_i), \ \forall \ (\boldsymbol{x}_i, y_i) \in \mathcal{D},$$

$$C_j\boldsymbol{\beta} = \int_X \langle \boldsymbol{\beta}, \phi(\boldsymbol{x}) \rangle \phi(\boldsymbol{x}) d\rho_j(\boldsymbol{x}), \qquad C_{\mathcal{D}_j}\boldsymbol{\beta} = \frac{1}{|\mathcal{D}_j|} \sum_{i=1}^{|\mathcal{D}_j|} \langle \boldsymbol{\beta}, \phi(\boldsymbol{x}_i) \rangle \phi(\boldsymbol{x}_i), \ \forall \ (\boldsymbol{x}_i, y_i) \in \mathcal{D}_j.$$

Note that, $C_{\mathcal{D}} = \boldsymbol{\Phi}_{\mathcal{D}}^\top \boldsymbol{\Phi}_{\mathcal{D}}$, $C_{\mathcal{D}_j} = \boldsymbol{\Phi}_{\mathcal{D}_j}^\top \boldsymbol{\Phi}_{\mathcal{D}_j}$ are the empirical covariance operators on $\mathcal{D}$ and $\mathcal{D}_j$, while $C = \mathbb{E}_\rho[C_{\mathcal{D}}], C_j = \mathbb{E}_{\rho_j}[C_{\mathcal{D}_j}]$ are their expected counterparts.

For the sake of readability, we provide some notations

$$\mathcal{P}_{\mathcal{D}_j, \lambda} := \|(C_{\mathcal{D}_j} + \lambda I)^{-1}(C_j + \lambda I)\|, \qquad \mathcal{R}_{\mathcal{D}_j, \lambda} := \|(C_j + \lambda)^{-1}(C_j - C_{\mathcal{D}_j})\|,$$
$$\Delta_{\mathcal{D}_j} := \|C - C_j\|, \qquad\qquad\qquad \Delta_{f_j} := \|f^* - f_j^*\|.$$

The quantities $\mathcal{P}_{\mathcal{D}_j, \lambda}$ and $\mathcal{R}_{\mathcal{D}_j, \lambda}$ measure the similarity between the expected covariance operator and its empirical counterpart. From contraction inequalities for self-adjoint operators, a larger number of local samples $|\mathcal{D}_j|$ leads to smaller $\mathcal{P}_{\mathcal{D}_j, \lambda}$ and $\mathcal{R}_{\mathcal{D}_j, \lambda}$. Note that, $\Delta_{\mathcal{D}_j}$ measures the data heterogeneity on the expected covariance operator, while $\Delta_{f_j}$ measures the model heterogeneity on the true regressions.

We let $\|f\|_2 = \sqrt{\langle f, f \rangle} = \sqrt{\int_X |f(\boldsymbol{x})|^2 d\mathbb{P}(\boldsymbol{x})}$ denote the $L^2(\mathbb{P})$ norm and $L^2(\mathbb{P}) = \{f : \mathcal{X} \to \mathbb{R} \mid \|f\|_2^2 < \infty\}$. Throughout this paper, we assume the outputs are bounded $|y| \leq B$ almost surely for some $B > 0$ and $\kappa := \|\phi(\boldsymbol{x})\|_K < \infty$ for any $\boldsymbol{x} \in \mathcal{X}$.

**Assumption 1** (Federated capacity condition). *For $\lambda \in (0, 1)$, we define the effective dimensions on the global distribution $\rho$ and local distributions $\rho_j, \forall j \in [m]$ as*

$$\mathcal{N}(\lambda) = Tr(C(C + \lambda I)^{-1}), \ \mathcal{N}_j(\lambda) = Tr(C_j(C_j + \lambda I)^{-1}).$$

*Assume there exists $Q > 0$ and $\gamma \in [0, 1]$, such that*

$$\max(\mathcal{N}(\lambda), \mathcal{N}_1(\lambda), \cdots, \mathcal{N}_m(\lambda)) \leq Q^2 \lambda^{-\gamma}.$$

**Assumption 2** (Source condition). *Define the integral operators $L : L^2(\mathbb{P}) \to L^2(\mathbb{P})$,*

$$(Lg)(\cdot) = \int_X \langle \phi(\cdot), \phi(\boldsymbol{x}) \rangle g(\boldsymbol{x}) d\rho_X(\boldsymbol{x}), \quad \forall \ g \in L^2(\mathbb{P}).$$

*Assume there exists $R > 0$, $r > 0$, such that $\|L^{-r} f^*\| \leq R$. where the operator $L^r$ denotes the $r$-th power of $L$ as a compact and positive operator.*

Capacity condition and source condition are standard assumptions in the optimal statistical learning for the KRR related literature Caponnetto & De Vito (2007); Smale & Zhou (2007); Rudi & Rosasco (2017); Lin & Cevher (2020); Liu et al. (2021). The effective dimensions $\mathcal{N}(\lambda)$ and $\mathcal{N}_j(\lambda)$ measure the capacities of the RKHS $\mathcal{H}_K$ on the global distribution $\rho$ and the local distributions $\rho_j$, $\forall j \in [m]$.

Here, we modify the conventional capacity condition for federated learning to impose constraints on local estimators. Note that, for effective dimensions, it holds $1/2 \leq \max(\mathcal{N}(\lambda), \mathcal{N}_1(\lambda), \cdots, \mathcal{N}_m(\lambda)) \leq \kappa^2 \lambda^{-1}$ Rudi et al. (2015). Assumption 1 reflects the variance of the estimator. A larger $\gamma$ leads to a larger $\mathcal{H}_K$ and $\gamma = 1$ corresponds to the capacity independence case. Assumption 2 controls the bias of an estimator, which reflects the regularity of the estimator. The bigger $r$ leads to the stronger regularity of the regression and the easier learning problem. The general settings $(r = 1/2, \gamma = 1)$ lead to $\boldsymbol{O}(1/\sqrt{|D|})$ convergence rates for KRR related approaches.

## 4.2 ERROR DECOMPOSITION

**Theorem 1.** *Let $f_{\mathcal{D},\lambda}, \bar{f}^t_{\mathcal{D},\lambda}, f^*$ be defined according to equation 2, equation 5 and equation 9. Then, the following error decomposition holds*

$$\|\bar{f}^t_{\mathcal{D},\lambda} - f^*\| \leq \underbrace{\|\bar{f}^t_{\mathcal{D},\lambda} - f_{\mathcal{D},\lambda}\|}_{\text{federated error}} + \underbrace{\|f_{\mathcal{D},\lambda} - f^*\|}_{\text{centralized excess risk}}, \tag{10}$$

*and the federated error for* `FedNewton` *is bounded by:*

$$\|\bar{f}^t_{\mathcal{D},\lambda} - f_{\mathcal{D},\lambda}\|_2 \leq \Upsilon^t \left\| (C + \lambda I)^{1/2}(\bar{\boldsymbol{w}}^0_{\mathcal{D},\lambda} - \boldsymbol{w}_{\mathcal{D},\lambda}) \right\|_K,$$

*where $\Upsilon = \sum_{j=1}^m p_j \mathcal{P}_{\mathcal{D}_j,\lambda} \left( 2\mathcal{R}_{\mathcal{D}_j,\lambda} + \frac{\Delta_{\mathcal{D}_j}}{\lambda} \right) \left( 1 + \frac{\Delta_{\mathcal{D}_j}}{\lambda} \right).$*

In the above theorem, we decompose the excess risk for `FedNewton` into two parts: the federated error $\|\bar{f}^t_{\mathcal{D},\lambda} - f_{\mathcal{D},\lambda}\|$ and the excess risk for the centralized KRR $\|f_{\mathcal{D},\lambda} - f^*\|$. Since the generalization analysis for $\|f_{\mathcal{D},\lambda} - f^*\|$ is standard Caponnetto & De Vito (2007); Smale & Zhou (2007), we focus on the federated error $\|\bar{f}^t_{\mathcal{D},\lambda} - f_{\mathcal{D},\lambda}\|$.

From Theorem 1, we find that the value of $\Upsilon$ determines the effectiveness of multiple iterations. If $\Upsilon \geq 1$, `FedNewton` with multiple communications is worse than oneshot federated learning (DKRR). However, when $\Upsilon < 1$, the federated error decreases exponentially and the rate of convergence is referred to as *linear convergence* in the optimization literature Bottou et al. (2018). The quantities $\mathcal{P}_{\mathcal{D}_j,\lambda}$ and $\mathcal{R}_{\mathcal{D}_j,\lambda}$ measure the similarity between $C_{\mathcal{D}_j}$ and $C_j$ where those quantities decrease as the local sample size $|\mathcal{D}_j|$ increases. Because $\Upsilon$ is proportional to $\mathcal{P}_{\mathcal{D}_j,\lambda}$, $\mathcal{P}_{\mathcal{D}_j,\lambda}$ and $\Delta_{\mathcal{D}_j}$, the *linear convergence* requires both a sufficient number of local examples $|\mathcal{D}_j|$ and moderate data heterogeneity $\Delta_{\mathcal{D}_j}$. If $t = 0$, the above error bound degrades into that for DKRR $\|\bar{f}_{\mathcal{D},\lambda} - f_{\mathcal{D},\lambda}\|_2 \leq \left\| (C + \lambda I)^{1/2}(\bar{\boldsymbol{w}}^0_{\mathcal{D},\lambda} - \boldsymbol{w}_{\mathcal{D},\lambda}) \right\|_K.$

**Theorem 2.** *Under Assumption 2, with a high probability $1 - \delta$, $\forall \delta \in (0,1)$, the federated error can be bounded*

$$\|\bar{f}^t_{\mathcal{D},\lambda} - f_{\mathcal{D},\lambda}\|_2$$

$$\lesssim \Upsilon^t \sum_{j=1}^m p_j \sqrt{1 + \frac{\Delta_{\mathcal{D}_j}}{\lambda}} \left( 2\mathcal{R}_{\mathcal{D}_j,\lambda} + \frac{(1 + \mathcal{R}_{\mathcal{D}_j,\lambda})\Delta_{\mathcal{D}_j}}{\lambda} \right) \left( \left( \frac{1}{|\mathcal{D}_j|\sqrt{\lambda}} + \sqrt{\frac{\mathcal{N}(\lambda)}{|\mathcal{D}_j|}} \right) \log \frac{2}{\delta} + \frac{\Delta_{\mathcal{D}_j}}{\lambda} + \Delta_{f_j} \right).$$

Theorem 2 illustrates the key factors that affect the federated error: the discrepancy between expected and empirical covariance operators $\mathcal{R}_{\mathcal{D}_j,\lambda}$, the covariate shift $\Delta_{\mathcal{D}_j}$, and the model heterogeneity $\Delta_{f_j}$. The smaller these factors, the smaller the federated error. The federated error results from three parts: distributed error $\frac{1}{\sqrt{\lambda}|\mathcal{D}_j|} + \sqrt{\frac{\mathcal{N}(\lambda)}{|\mathcal{D}_j|}}$, covariate shift $\Delta_{\mathcal{D}_j}/\lambda$ and concept shift $\Delta_{f_j}$. Specifically, as the increase of local sample size, the distributed error decreases. However, the concept shifts $\Delta_{f_j}$ is a constant and it will dominate the federated error when model heterogeneity $\Delta_{f_j}$ is large. In the case $\Upsilon < 1$, iterators can reduce the federated error.

### 4.3 HOMOGENEOUS SETTING

**Theorem 3.** *Let $\delta \in (0, 1/3]$, $\lambda = |\mathcal{D}|^{\frac{-1}{2r+\gamma}}$ and $2r + \gamma \geq 1$. Under Assumptions 1, 2, if $\Delta_{\mathcal{D}_j} = 0$ and $\Delta_{f_j} = 0$, with the probability at least $1 - 3\delta$, it holds*

$$\|\bar{f}^t_{\mathcal{D},\lambda} - f^*\|_2 \lesssim \Upsilon^t \sum_{j=1}^m p_j \aleph_j \, \log^2 \frac{2}{\delta} + |\mathcal{D}|^{\frac{-r}{2r+\gamma}} \log \frac{2}{\delta}.$$

*Here, $\aleph_j$ and $\Upsilon$ have different values w.r.t local sample size*

$$
\aleph_j =
\begin{cases}
|\mathcal{D}_j|^{-2} |\mathcal{D}|^{\frac{1.5}{2r+\gamma}}, & \text{if } |\mathcal{D}_j| \lesssim |\mathcal{D}|^{\frac{1-\gamma}{2r+\gamma}} \\
|\mathcal{D}_j|^{-1.5} |\mathcal{D}|^{\frac{1+0.5\gamma}{2r+\gamma}}, & \text{if } |\mathcal{D}|^{\frac{1-\gamma}{2r+\gamma}} \lesssim |\mathcal{D}_j| \lesssim |\mathcal{D}|^{\frac{1}{2r+\gamma}} \\
|\mathcal{D}_j|^{-1} |\mathcal{D}|^{\frac{1+\gamma}{4r+2\gamma}}, & \text{if } |\mathcal{D}|^{\frac{1}{2r+\gamma}} \lesssim |\mathcal{D}_j| \lesssim |\mathcal{D}|^{\frac{2r+\gamma+1}{4r+2\gamma}} \\
|\mathcal{D}|^{\frac{-r}{2r+\gamma}}, & \text{if } |\mathcal{D}_j| \gtrsim |\mathcal{D}|^{\frac{2r+\gamma+1}{4r+2\gamma}},
\end{cases}
$$

*and $\Upsilon = 2 \sum_{j=1}^m p_j \mathcal{P}_{\mathcal{D}_j,\lambda} \mathcal{R}_{\mathcal{D}_j,\lambda}$ holds*

$$
\begin{cases}
\Upsilon \geq 1, & \text{if } |\mathcal{D}_j| \lesssim |\mathcal{D}|^{\frac{1}{2r+\gamma}} \\
\Upsilon \lesssim \frac{|\mathcal{D}|^{\frac{1}{2r+\gamma}}}{|\mathcal{D}_j|} < 1, & \text{otherwise.}
\end{cases}
$$

Note that, the second term in the above bound is from the centralized model $\|f_{\mathcal{D},\lambda} - f^*\|_2$, where the learning rate $\boldsymbol{O}(|\mathcal{D}|^{\frac{-r}{2r+\gamma}})$ is optimal in a minimax sense Caponnetto & De Vito (2007). The performance of `FedNewton` in the homogeneous setting is only affected by the local sample size. We discuss the above result in three parts. First, when the number of local examples is limited $|\mathcal{D}_j| \lesssim |\mathcal{D}|^{\frac{1}{2r+\gamma}}$, in another word the number of local machines is larger than $m \gtrsim |\mathcal{D}|^{\frac{2r+\gamma-1}{2r+\gamma}}$, the federated error dominates the excess risk and fails to achieve the optimal rate, where the convergence rates are slower than $\mathcal{O}(|\mathcal{D}|^{\frac{\gamma-1}{4r+2\gamma}})$. Meanwhile, when the number of local examples is limited, it leads to $\Upsilon \geq 1$ and multiple communications hurt the performance. Second, when $|\mathcal{D}|^{\frac{1}{2r+\gamma}} \lesssim |\mathcal{D}_j| \lesssim |\mathcal{D}|^{\frac{2r+\gamma+1}{4r+2\gamma}}$, although the convergence rates of federated error are still not the optimal, the iterator $\Upsilon$ is smaller than one, leading to a linear convergence. As the increase of communications $t \to \infty$, the centralized excess risk will dominate the error bound that achieves the optimal rate. Third, with a large number of local examples $|\mathcal{D}_j| \gtrsim |\mathcal{D}|^{\frac{2r+\gamma+1}{4r+2\gamma}}$, even with insufficient communications $t \to 0$, the error bound still achieves the optimal rate $\boldsymbol{O}(|\mathcal{D}|^{\frac{-r}{2r+\gamma}})$.

Theorem 3 can be further simplified in some special cases. For example, we consider the general case $(r = 1/2, \gamma = 1)$, where $r = 1/2$ is equivalent to assuming $f^* \in \mathcal{H}_K$ and $\gamma = 1$ is the capacity independent case. The learning rate achieves $\boldsymbol{O}(1/\sqrt{|\mathcal{D}|})$ when $|\mathcal{D}_j| \gtrsim |\mathcal{D}|^{0.5}$ with multiple iterations or $|\mathcal{D}_j| \gtrsim |\mathcal{D}|^{0.75}$ with only one communication.

**Remark 5.** *The existing theoretical guarantees for DKRR Zhang et al. (2015); Guo et al. (2017); Lin & Cevher (2020) focused on how to achieve the optimal rate by a sufficient number of local examples (or lower the number of partitions), but they ignored the sub-optimal case that the local sample size is fixed and insufficient. However, in federated learning, the number of partitions is fixed and local examples are generated locally, such that sub-optimal cases are more general. Theorem 3 illustrate that a sufficient number of local examples is crucial for both learning rates (in generalization) and convergence rate (in optimization).*

**Remark 6** (Finite dimensional case). *In the proofs of theoretical findings, we consider the estimator in RKHS with $\boldsymbol{w} \in \mathcal{H}_K$. However, the finite-dimensional cases are more general, i.e. $\boldsymbol{w} \in \mathbb{R}^M$ in Algorithm 1, where the feature mappings are explicit and can be neural networks or random features Rahimi & Recht (2007). With a simple modification of our proofs, one can derive similar results for finite-dimensional cases. In particular, under same assumptions of Theorem 3 and $(r = 1/2, \gamma = 0)$, then with high probability, $\|\bar{f}^t_{\mathcal{D},\lambda} - f^*\|_2 \lesssim |\mathcal{D}_j|^{-2}|\mathcal{D}|^{1.5} + \sqrt{M/|\mathcal{D}|}$, provided that $|\mathcal{D}| \gtrsim M \log M$.*

*As shown in Rudi & Rosasco (2017), a large number of random features $M \gtrsim |\mathcal{D}|^{\frac{1+\gamma(2r-1)}{2r+\gamma}}$ can guarantee the optimal rates for $\|\bar{f}_{\mathcal{D},\lambda} - f^*\|_2$, and thus there are similar results as Theorem 3.*

## 4.4 HETEROGENEOUS SETTING

**Theorem 4.** *Let $\delta \in (0, 1/3]$, $\lambda = |\mathcal{D}|^{\frac{-1}{2r+\gamma}}$ and $2r + \gamma \geq 1$. Under Assumptions 1, 2, with the probability at least $1 - 3\delta$, the excess risk bound for* FedNewton *holds*

$$\|\bar{f}_{\mathcal{D},\lambda}^t - f^*\|_2 \lesssim \Upsilon^t \sum_{j=1}^m p_j \sqrt{1 + \frac{\Delta_{\mathcal{D}_j}}{\lambda}} (\aleph_j + \Pi_j) \log^2 \frac{2}{\delta} + |\mathcal{D}|^{\frac{-r}{2r+\gamma}} \log \frac{2}{\delta}.$$

*Here, $\Upsilon = \sum_{j=1}^m p_j \mathcal{P}_{\mathcal{D}_j,\lambda} (2\mathcal{R}_{\mathcal{D}_j,\lambda} + \frac{\Delta_{\mathcal{D}_j}}{\lambda})(1 + \frac{\Delta_{\mathcal{D}_j}}{\lambda})$, $\aleph_j$ is same to Theorem 3 and*

$$\Pi_j = \begin{cases} \frac{|\mathcal{D}|^{\frac{2}{2r+\gamma}}}{|\mathcal{D}_j|} \Delta_{\mathcal{D}_j} + \frac{|\mathcal{D}|^{\frac{1}{2r+\gamma}}}{|\mathcal{D}_j|} \Delta_{f_j}, & \text{if } |\mathcal{D}_j| \lesssim |\mathcal{D}|^{\frac{1}{2r+\gamma}} \\ (1 + |\mathcal{D}|^{\frac{1}{2r+\gamma}} \Delta_{\mathcal{D}_j})(\Delta_{f_j} + |\mathcal{D}|^{\frac{1}{2r+\gamma}} \Delta_{\mathcal{D}_j}), & \text{otherwise.} \end{cases}$$

We add some comments on the above theorem. First, when the local sample size is insufficient $|\mathcal{D}_j| \lesssim |\mathcal{D}|^{\frac{1}{2r+\gamma}}$ or the data heterogeneity is considerable, we have $\Upsilon \geq 1$, and communications hurt the performance. Meanwhile, since the federated error $\sqrt{1 + \Delta_{\mathcal{D}_j}/\lambda}(\aleph_j + \Pi_j)$ depends on $|\mathcal{D}_j|, \Delta_{\mathcal{D}_j}$, and $\Delta_{f_j}$, the learning rate is far from the optimal rate. Second, when the number of local examples is sufficient $|\mathcal{D}_j| \gtrsim |\mathcal{D}|^{\frac{1}{2r+\gamma}}$ and data heterogeneity is small, it holds $\Upsilon < 1$ where communications can improve the generalization ability of FedNewton. In this case, the federated error $\|\bar{f}_{\mathcal{D},\lambda}^t - f_{\mathcal{D},\lambda}\|$ converge exponentially fast. If $t$ is large enough, the error bound in Theorem 4 depends on the centralized excess risk $\|f_{\mathcal{D},\lambda} - f^*\|_2$ and achieves the optimal learning rate.

The learning rate of generalization bound in Theorem 4 is determined by four factors: the local sample size $|\mathcal{D}_j|$, the covariate shift $\Delta_{\mathcal{D}_j}$, the response shift $\Delta_{f_j}$ and the number of iterations $t$. Furthermore, the iterator value $\Upsilon$ depends on $|\mathcal{D}_j|$ and $\Delta_{\mathcal{D}_j}$, such that these two values are important factors for both fast convergences (in optimization) and the learning rates (in generalization).

**Remark 7** (How to achieve the optimal rate in federated learning?)**.** *The value of $\Upsilon < 1$ is key to obtaining a linear convergence rate and the optimal learning rate, where it depends on both local sample sizes $\Upsilon \propto \mathcal{R}_{\mathcal{D}_j,\lambda} \propto |\mathcal{D}_j|$ and data heterogeneity $\Upsilon \propto \Delta_{\mathcal{D}_j}$. Note that, $\Delta_{\mathcal{D}_j}$ measures the intrinsic discrepancy between local distributions and the global one, and thus it is a fixed value independent from the local sample size. Therefore, since $\Delta_{\mathcal{D}_j}$ is a constant, we can obtain $\Upsilon < 1$ with a large number of local examples generated by local machines. And then, with a large number of iterations when $\Upsilon < 1$, the federated error, depending on both data heterogeneity and model heterogeneity, can become small enough to be negligible. In this case, a large number of local examples can guarantee both a linear convergence rate (for federated error) and the optimal learning rate (from the centralized excess risk). A large number of local examples benefit both optimization and generalization, rather than making tradeoffs between them.*

## 5 COMPARED WITH RELATED WORK

We compare FedNewton with recent Newton-type methods, DKRR methods, and first-order FL algorithms in both algorithmic and theoretical fronts. Table 1 reports the main factors that affect the performance, the computational and generalization properties of related work.

**Compared with DKRR.** DKRR methods in kernel space (Zhang et al., 2015; Guo et al., 2017) incur much higher time complexity than stochastic optimization in feature space. Although both our work and (Guo et al., 2017; Lin & Cevher, 2018; Lin et al., 2020) rely on integral operator techniques, DKRR assumes i.i.d. local data and thus avoids the data/model heterogeneity central to our setting, making their proofs significantly simpler. Key distinctions are: (1) we relax the regularity condition from $r \in [\frac{1}{2}, 1]$ to $r > 0, 2r + \gamma \geq 1$; (2) we handle non-i.i.d. data with both covariate ($\Delta_{\mathcal{D}_j}$) and response ($\Delta_{f_j}$) shifts, whereas DKRR covers only the homogeneous i.i.d. case; (3) to cope with heterogeneity we introduce new error decompositions for the federated excess risk; and (4) we bound excess risk for varying local sample sizes (Theorem 3) covering both optimal and sub-optimal rates, while DKRR analyzes only optimal rates under restrictive partition constraints (Lin et al., 2020).

**Compared with first-order methods.** First-order analyses such as (Su et al., 2021) achieve the optimal rate $\|f - f^*\|_2^2 = \mathcal{O}(1/|\mathcal{D}|)$ via random matrix theory and local Rademacher complexity, but

Table 1: Summary of computational and generalization properties for related work.

| Related Work | $|\mathcal{D}_j|\Delta_{\mathcal{D}_j}\Delta_{f_j}$ | Global Convergence | Communication | Conditions | Local Size $|\mathcal{D}_j|$ | Generalization Bound |
|---|---|---|---|---|---|---|
| DKRR Zhang et al. (2015) | √ × × | $O(1)$ | $|\mathcal{D}|$ | Specific kernels | $\Omega(r^2\kappa^4\log|\mathcal{D}|)$ | $O\left(\frac{1}{|\mathcal{D}|}\right)$ |
| DKRR Guo et al. (2017) | √ × × | $O(1)$ | $|\mathcal{D}|$ | $r\in[1/2,1]$ | $\Omega(|\mathcal{D}|^{\frac{1+\gamma}{2r+\gamma}})$ | $O(|\mathcal{D}|^{\frac{-r}{2r+\gamma}})$ |
| DKRR-SGD Lin & Cevher (2018) | √ × × | $O(|\mathcal{D}|^{\frac{2-\gamma}{2r+\gamma}})$ | $|\mathcal{D}|$ | $r\in[1/2,1]$ | $\Omega(|\mathcal{D}|^{\frac{1}{2r+\gamma}})$ | $O\left(|\mathcal{D}|^{\frac{-r}{2r+\gamma}}\right)$ |
| DKRR-CM Lin et al. (2020) | √ × × | $O(\log\frac{1}{\epsilon})$ | $|\mathcal{D}|t$ | $r\in[1/2,1]$ | $\Omega(|\mathcal{D}|^{\frac{2r+\gamma+1}{4r+2\gamma}})$ | $O\left(|\mathcal{D}|^{\frac{-r}{2r+\gamma}}\right)$ |
| FedAvg Su et al. (2021) | × × √ | $O(\frac{1}{\epsilon})$ | $Mt$ | Specific kernels | / | $O\left(\frac{1}{\eta t}+\frac{\Delta_f^2}{|\mathcal{D}|}\right)$ |
| FedProx Su et al. (2021) | × × √ | $O(\frac{1}{\epsilon})$ | $Mt$ | Specific kernels | / | $O\left(\frac{1}{\eta t}+\frac{\Delta_f^2}{|\mathcal{D}|}\right)$ |
| DistributedNewton Ghosh et al. (2020) | × × × | $O(\log\log\frac{1}{\epsilon}+\log\frac{1}{\epsilon})$ | $Mt$ | / | / | / |
| LocalNewton Gupta et al. (2021) | × × × | $O(\log\frac{1}{\epsilon})$ | $Mt$ | / | / | / |
| FedNew Elgabli et al. (2022) | √ × × | / | $Mt$ | / | / | / |
| FedNL Safaryan et al. (2022) | √ √ × | $O(\log\frac{1}{\epsilon})$ | $Mt$ | / | / | / |
| SHED Dal Fabbro et al. (2024) | √ √ × | $O(\log\log\frac{1}{\epsilon})$ | $M^2t$ | / | / | / |
| FedNS Li et al. (2024) | √ √ × | $O(\log\log\frac{1}{\epsilon})$ | $kMt$ | / | / | / |
| Fed-sofia Elbakary et al. (2024) | √ √ × | / | $Mt$ | / | / | / |
| Theorem 3 | √ × × | $O(\log\frac{1}{\epsilon})$ | $Mt$ | $r>0, 2r+\gamma\geq1$ | $\Omega(|\mathcal{D}|^{\frac{1}{2r+\gamma}})$ | Theorem 3 |
| Theorem 4 | √ √ √ | $O(\log\frac{1}{\epsilon})$ | $Mt$ | $r>0, 2r+\gamma\geq1$ | $\Omega(|\mathcal{D}|^{\frac{1}{2r+\gamma}})$ | Theorem 4 |

Note: The computational complexities are computed in terms of regularized least squared loss. We estimate the upper bounds for $\|f-f^*\|_2 \ \forall f\in L^2(\mathbb{P})$. We denote $\mathcal{D}_{\text{test}}$ the testing data, $\eta$ the step-size for SGD approaches, $\epsilon$ the federated error and $\Delta_f^2=\sum_{j=1}^m p_j\Delta_{f_j}^2$. For Rademacher complexities based bounds Zhang et al. (2015); Su et al. (2021), specific kernels include kernels with finite-rank or polynomial eigenvalues decay. $k$ is the subsampled size for Nyström approximation. Integral operator based bounds Guo et al. (2017); Lin & Cevher (2018); Lin et al. (2020) also assume $\gamma\in[0,1]$.

assume i.i.d. inputs and neglect both local sample size and data heterogeneity. Their results further rely on (i) the target function lying in the hypothesis space ($r\in[\frac{1}{2},1]$), (ii) vanishing complexity ($\gamma\to0$), and (iii) specific kernels that may be sub-optimal for federated tasks. Our integral-operator analysis removes these constraints, explicitly capturing the role of local sample size and heterogeneity, and shows that with sufficient local data and moderate heterogeneity the federated error converges *linearly* at the optimal rate $\mathcal{O}(|\mathcal{D}|^{-2r/(2r+\gamma)})$, whereas the rate in (Su et al., 2021) remains sublinear and deteriorates with model heterogeneity $\mathcal{O}(\frac{\sum_{j=1}^m p_j\Delta_{f_j}^2}{|\mathcal{D}|})$.

**Compared with Newton-type FL methods.** Compared with existing second-order federated optimization methods, our theoretical results (Theorems 3 and 4) advance the state of the art in three key aspects. *(i) Classical distributed Newton methods* such as DistributedNewton (Ghosh et al., 2020) and LocalNewton (Gupta et al., 2021) provide fast convergence ($\mathcal{O}(\log\frac{1}{\epsilon})$ or better) but purely from an optimization perspective, without any formal statistical generalization or minimax risk analysis. *(ii) Federated Newton variants under non-i.i.d. data* including FedNL (Safaryan et al., 2022), SHED (Dal Fabbro et al., 2024) and FedNS (Li et al., 2024) address data heterogeneity and retain logarithmic convergence, yet their theory stops at bounding the optimization gap and lacks explicit excess-risk guarantees. *(iii) Our work* not only preserves the fast $\mathcal{O}(\log\frac{1}{\epsilon})$ convergence and $Mt$ communication complexity typical of second-order approaches, but also establishes tight minimax lower bounds on the excess risk and extends them to heterogeneous settings—offering, to our knowledge, the first unified optimization–generalization theory for federated second-order methods.

# 6 CONCLUSION AND FUTURE WORK

In this paper, we present an efficient second-order optimization method for FL. We derive generalization bounds with the optimal rates, which quantify the impacts of local sample size, the data heterogeneity, and the model heterogeneity. In benign cases, the federated error convergence exponentially fast, and thus communications can be small. Our theoretical findings fill the gap between optimization and generalization for federated learning, rather than focusing on one of them. Overall, the techniques presented here highlight new ways for designing efficient algorithms and analyzing both generalization and optimization for FL.

In future, we first aim to explore superlinear convergence for `FedNewton`, potentially leveraging Hessian-free approximations to reduce computational overhead while maintaining fast convergence. Then, we extend the theoretical framework to nonconvex losses and more general model classes.

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

# A EFFICIENT SOLVERS FOR SCALABLE FEDNEWTON

While FedNewton provides a theoretically superior update direction via the local Hessian $\mathbf{H}_k$, computing the exact Newton step $\Delta \mathbf{w} = \mathbf{H}_k^{-1} \mathbf{g}$ in equation 7 is computationally prohibitive for high-dimensional models. Explicitly forming $\mathbf{H}_k \in \mathbb{R}^{d \times d}$ requires $\mathcal{O}(d^2)$ memory, and inverting it requires $\mathcal{O}(d^3)$ operations. To make FedNewton feasible for modern deep networks, we propose and implement four distinct approximation strategies to solve the linear system $\mathbf{H}_k \Delta \mathbf{w} = \mathbf{g}$ without explicit matrix inversion.

## A.1 MATRIX-FREE CONJUGATE GRADIENT (CG)

To avoid the memory bottleneck of storing the Hessian, we employ the Conjugate Gradient (CG) method. CG is an iterative algorithm that solves linear systems involving symmetric positive-definite matrices using only matrix-vector products (MVPs). Instead of computing $\mathbf{H}_k$ explicitly, we compute the product $\mathbf{H}_k \mathbf{v}$ for an arbitrary vector $\mathbf{v}$ using the "Pearlmutter trick" (automatic differentiation):

$$\mathbf{H}_k \mathbf{v} = \nabla_{\mathbf{w}}(\nabla_{\mathbf{w}} \mathcal{L}(\mathbf{w})^\top \mathbf{v}). \tag{11}$$

This operation has a computational cost of $\mathcal{O}(d)$, similar to a standard gradient backpropagation. By truncating the CG process to a small number of iterations (e.g., $K_{cg} = 10$), we obtain a high-precision approximation of the Newton step with linear memory complexity $\mathcal{O}(d)$, making it applicable to large-scale models.

## A.2 DIAGONAL APPROXIMATION (FEDNEWTON-DIAG)

For scenarios with extreme resource constraints, we implement a diagonal approximation strategy. We approximate the Hessian as $\mathbf{H}_k \approx \mathrm{diag}(\mathbf{h})$, where $\mathbf{h} \in \mathbb{R}^d$ captures the curvature along the axes. In our implementation, we estimate the diagonal elements using the empirical Fisher Information Matrix (FIM) on the current batch of data $B$:

$$\mathbf{h}_{ii} = \frac{1}{|B|} \sum_{x \in B} \left( \frac{\partial \ell(x; \mathbf{w})}{\partial w_i} \right)^2 + \lambda. \tag{12}$$

The update rule simplifies to an element-wise division $\Delta w_i = g_i / h_{ii}$. While this ignores cross-parameter interactions, it reduces the computational overhead to that of a single backward pass, offering a robust baseline similar to adaptive optimizers like Adam or FedSophia.

## A.3 LIMITED-MEMORY BFGS (L-BFGS)

To utilize historical curvature information without accessing the second-order derivatives directly, we adapt the Limited-Memory BFGS (L-BFGS) algorithm for the federated setting. Unlike standard Newton methods that compute the Hessian from scratch at every round, our L-BFGS solver maintains a history of the $m$ most recent updates $\mathbf{s}_t = \mathbf{w}_{t+1} - \mathbf{w}_t$ and gradient differences $\mathbf{y}_t = \mathbf{g}_{t+1} - \mathbf{g}_t$. Using the two-loop recursion algorithm, we compute the direction $\mathbf{d} = -\mathbf{H}_{L-BFGS}^{-1} \mathbf{g}$ directly. Crucially, in our Federated implementation, the L-BFGS state is maintained on the **server** or synchronized across clients to ensure the curvature history reflects the global optimization trajectory, smoothing out the noise inherent in local client updates.

## A.4 LOW-RANK APPROXIMATION

We explore a Low-Rank approximation based on the observation that the Hessian spectrum of neural networks is often dominated by a few large eigenvalues (the "bulk" and "outliers" structure). We employ a truncated Krylov subspace method (implemented via early-stopped CG or Lanczos iterations) to project the optimization problem into a lower-dimensional subspace spanned by the top-$k$ eigenvectors. Formally, if $\mathbf{H}_k \approx \mathbf{U} \Lambda \mathbf{U}^\top$ where $\mathbf{U} \in \mathbb{R}^{d \times k}$ contains the top eigenvectors, the inverse is approximated in this subspace. This method acts as a regularizer, focusing the Newton update on the directions of highest curvature while leaving flat directions to standard gradient descent, thereby preventing instability from noisy small eigenvalues.

Table 2: Complexity Analysis of Proposed Solvers. $d$ denotes the number of parameters, $K_{cg}$ is CG iterations, $m$ is L-BFGS memory size, and $k$ is the rank.

| Solver Strategy | Time Complexity (per step) | Space Complexity | Exact Hessian Required? |
|---|---|---|---|
| Exact Newton | $\mathcal{O}(d^3)$ | $\mathcal{O}(d^2)$ | Yes |
| CG (Matrix-Free) | $\mathcal{O}(K_{cg} \cdot d)$ | $\mathcal{O}(d)$ | No (HVP only) |
| L-BFGS | $\mathcal{O}(m \cdot d)$ | $\mathcal{O}(m \cdot d)$ | No (History only) |
| Diagonal | $\mathcal{O}(d)$ | $\mathcal{O}(d)$ | No (Diag estimation) |
| Low-Rank | $\mathcal{O}(k \cdot d)$ | $\mathcal{O}(d)$ | No (HVP only) |

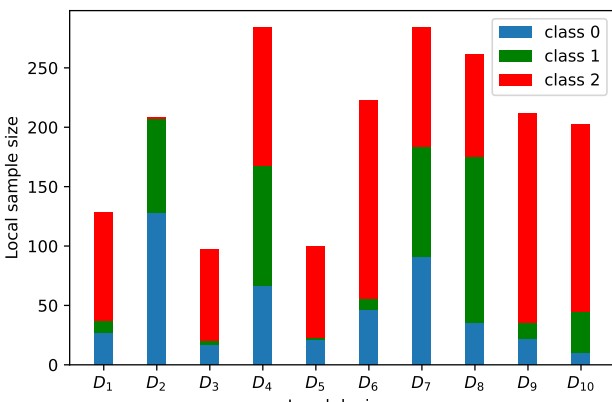

Figure 2: Data partitions for the dna dataset.

## B  EXPERIMENTS ON RF MODELS

In this section, we first carry out simulations to corroborate our theoretical statements. Then, we compare the performance of `FedNewton` with related baselines on real-world datasets.

DATASETS

**1) Synthetic dataset.** Although the existing work Li et al. (2020a); Lin et al. (2020); Su et al. (2021) provide strategies to generate synthetic datasets, these datasets either fail to impose both data heterogeneity and model heterogeneity among devices, or just fit a simple linear problem. Here, we focus on a nonlinear problem $f^*(\boldsymbol{x}) = \min(-\mathbf{1}^\top \boldsymbol{x}, \mathbf{1}^\top \boldsymbol{x})$ with $\boldsymbol{x} \sim \mathcal{N}(0, \mathbf{I})$. On the $j$-th local machine, we generate $\mathcal{D}_j = (\boldsymbol{X}_j, \boldsymbol{y}_j)$ based on $y = \min(-\boldsymbol{w}^\top \boldsymbol{x}, \boldsymbol{w}^\top \boldsymbol{x}) + \epsilon$, where $\epsilon \sim \mathcal{N}(0, 0.2)$ is the label noise, $\boldsymbol{x}_j \sim \mathcal{N}(\boldsymbol{u}_j, \mathbf{I})$, $\boldsymbol{u}_j \sim \mathcal{N}(0, \alpha)$ and $\boldsymbol{w}_j \sim \mathcal{N}(\mathbf{1}, \boldsymbol{v}_j)$, $\boldsymbol{v}_j \sim \mathcal{N}(0, \beta)$. Notably, $\alpha$ and $\beta$ control the data heterogeneity and model heterogeneity, respectively. Data heterogeneity and model heterogeneity increase as $\alpha$ and $\beta$ become larger, and the homogeneous setting corresponds to $\alpha = \beta = 0$. We set $d = 10$ and generate $|\mathcal{D}| = 10000$ samples for training, 2500 samples for testing.

**2) Real-world datasets.** We evaluate the compared algorithms on publicly available datasets from LIBSVM Data [1], which provide both training and testing data. To construct a heterogeneous and unbalanced setting, we split these datasets across 10 clients using a Dirichlet distribution $\text{Dir}_K(c)$ Wang et al. (2020), where $c$ is some constant relevant to the level of heterogeneity and unbalanced distribution. For example, the data partition for the *dna* dataset with $\text{Dir}_K(1)$ is reported in Figure 2 where the local datasets are both heterogeneous and unbalanced, which is common in federated learning scenarios.

---

[1]Available at `https://www.csie.ntu.edu.tw/~cjlin/libsvmtools/`

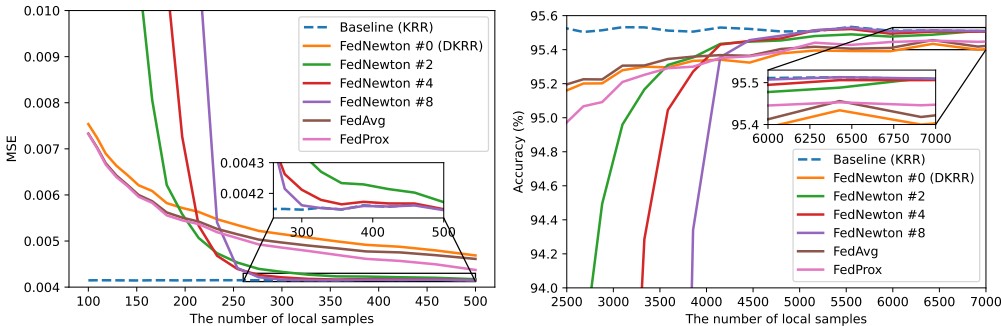

Figure 3: Impact of the number of local samples (left) on the synthetic dataset and MNIST dataset (right). The number of total training samples is fixed, $|\mathcal{D}_j| = |\mathcal{D}|/m$ and $\Delta_{\mathcal{D}_j} = \Delta_{f_j} = 0$. The blue dotted line denotes the exact KRR on all training data.

EXPERIMENTAL SETTINGS

We compared the proposed `FedNewton` with the baseline (KRR on entire data), DKRR (`FedNewton` with $t = 0$), FedAvg McMahan et al. (2017) and FedProx Li et al. (2020a) with the squared loss equation 1. The estimator can be expressed as $f(\boldsymbol{x}) = \langle \boldsymbol{w}, \phi(\boldsymbol{x}) \rangle$, where $\phi(\boldsymbol{x})$ denotes the feature mapping function. Here, we use random Fourier feature $\phi(\boldsymbol{x}) = 1/\sqrt{M} \cos(\boldsymbol{\Omega}^\top \boldsymbol{x} + \boldsymbol{b})$, where $\phi : \mathbb{R}^d \to \mathbb{R}^M, \boldsymbol{\Omega} \in \mathbb{R}^{d \times M}, \boldsymbol{b} \in \mathbb{R}^M$ and $\boldsymbol{\Omega} \sim \mathcal{N}(0, 1/\sigma^2), \boldsymbol{b} \in U[0, 2\pi]$. We set $M = 200$ for synthetic dataset and $M = 2000$ for real-world datasets. We implement all code based Pytorch and tune the hyperparameters over $\sigma^2 \in \{0.01, 0.1, \cdots, 1000\}$ and $\lambda = \{0.1, 0.01, \cdots, 10^{-7}\}$ by grid search. We report the data statistics and parameter setting in Table 3. All experiments are recorded by averaging results after 10 trials.

We initialize all iterative methods, including FedAvg, FedProx and `FedNewton`, by $\boldsymbol{w}_{\mathcal{D}_j,\lambda}^0 = \boldsymbol{H}_{\mathcal{D}_j,\lambda}^{-1} \boldsymbol{\Phi}_{\mathcal{D}_j}^\top \boldsymbol{y}_{\mathcal{D}_j}$ rather than $\boldsymbol{w}_{\mathcal{D}_j,\lambda}^0 = \boldsymbol{0}$. DKRR directly averages the initialized models. FedAvg updates local models with $s = 2$ iterations on all local data in each epoch. In Section B, we estimate the impact of local sample size, data heterogeneity without comparing FedAvg and FedProx. Here, we provide the full comparison with FedAvg and FedProx w.r.t. local sample size and data heterogeneity.

B.1 EMPIRICAL VALIDATIONS

We verify the theoretical findings in theorems by exploring how the factors empirically affect the performance on a synthetic dataset that can capture both data heterogeneity and model heterogeneity and the MNIST dataset.

**Impact of local sample size.** We explore the influence of local sample size $|\mathcal{D}_j|$ by fixing the total sample size $|\mathcal{D}| = 10000$ while varying the number $m$ of local machines, where $|\mathcal{D}_j| = \frac{|\mathcal{D}|}{m}$. As shown in the first two in Figure 3, when the number of local samples is small, i.e. $|\mathcal{D}_j| < 200$ for the synthetic dataset and $|\mathcal{D}_j| < 3300$ for MNIST, `FedNewton` with multiple communications hurts the generalization performance, and more communications lead to worse accuracy, corresponding to the cases $\Upsilon^t > 1$ in Theorem 3. When the local sample size is larger than a threshold, i.e. $|\mathcal{D}_j| \approx 260$ for the synthetic dataset and $|\mathcal{D}_j| \approx 4400$ for MNIST, more communications can significantly improve the predictive performance and get closer to the exact KRR, which coincides with the cases $\Upsilon^t < 1$ in Theorem 3. Note that, even with a large number of local examples, there still is a great gap between DKRR and KRR, while `FedNewton` achieves a good approximation to KRR. Meanwhile, both larger $|\mathcal{D}_j|$ and larger $t$ can improve the approximation ability, validating the theoretical results. Compared to first-order methods, when the local sample size is large enough, `FedNewton` outperforms FedAvg and FedProx. However, `FedNewton` is more sensitive to the number of local examples, and we find that the predictive error explodes when local sample size is small.

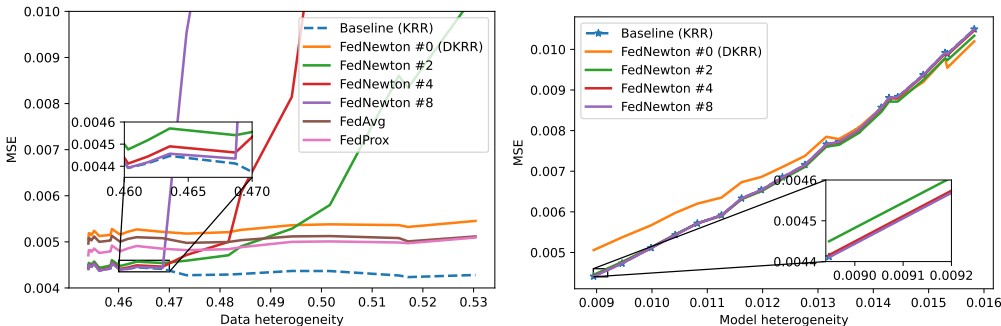

Figure 4: Impact of data heterogeneity (left) and model heterogeneity (right) on the synthetic dataset. We empirically estimate data heterogeneity by $\Delta_{\mathcal{D}_j} = [\boldsymbol{\Phi}_{\mathcal{D}}^{\top}\boldsymbol{\Phi}_{\mathcal{D}} - \boldsymbol{\Phi}_{\mathcal{D}_j}^{\top}\boldsymbol{\Phi}_{\mathcal{D}_j}]$, and model heterogeneity by $\Delta_{f_j} = \frac{1}{|\mathcal{D}_j|}\sum_{i=1}^{|\mathcal{D}_j|}[f^*(\boldsymbol{x}_i) - f_j^*(\boldsymbol{x}_i)]$.

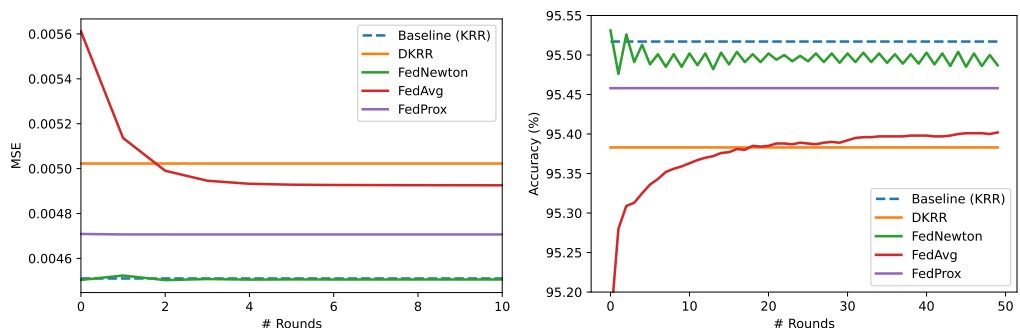

Figure 5: Predictive performance of `FedNewton`, FedAvg and FedProx on heterogeneous synthetic dataset (left) and MNIST (right).

**Impact of heterogeneous data.** Let $m = 20$ and $|\mathcal{D}_j| = 500$ for the synthetic dataset. We explore the impact of data heterogeneity by generating inputs with covariate shifts and explore the impact of model heterogeneity by generating outputs with response shifts. The right of Figure 3 illustrates: 1) Compared to DKRR, `FedNewton` remarkably reduce MSE when the heterogeneity is small. But it enlarges the errors from heterogeneous data when the heterogeneity is bigger than a threshold, i.e., $\Delta_{\mathcal{D}_j} \approx 0.466$. 2) For the benign data heterogeneous settings, more communications for `FedNewton` lead to better approximation to the exact KRR, while the gap between DKRR and KRR still exists. 3) When data heterogeneity is large, `FedNewton` is more sensitive to data heterogeneity than DKRR, and more communications hurt the predictive accuracy. In the line of federated learning, the data heterogeneity is common due to different data distributions while the model heterogeneity is usually small. The left of Figure 4 shows that 1) Model heterogeneity decreases the predictive performance for all methods. 2) More communications lead to better approximation to KRR when model heterogeneity is small. 3) The performance of all methods is similarly poor when model heterogeneity is bigger than 0.0135 and all models finally get similar bad results when model heterogeneity is large enough. These observations coincide with Theorem 4.

**Iterations of `FedNewton` and first-order methods.** We use heterogeneous dataset for iterations, i.e. the synthetic dataset with $\alpha = 0.01$ and $\gamma = 0.001$ and the MNIST dataset partitioned by a Dirichlet distribution $\text{Dir}_K(0.5)$. The last two in Figure 4 reports the generalization performance on heterogeneous data in terms of the communication rounds. We find that: 1) With a few iterations, `FedNewton` converges to KRR on the entire data, outperforming the divide-and-conquer and first-order methods. 2) Since local models are initialized by the closed-form solutions, FedProx converges very fast $t = 1$ and then updates slowly. The performance of FedProx is better than DKRR and

Table 3: Data statistics and hyperparameter settings.

| Dataset | Task | $|\mathcal{D}|$ | $d$ | classes | kernel parameter $\sigma^2$ | $\lambda$ | $\lambda_{\text{prox}}$ | $\text{Dir}_K(\alpha)$ | learning rate $\gamma$ |
|---|---|---|---|---|---|---|---|---|---|
| synthetic | regression | 10000 | 10 | 1 | 0.1 | 1e-06 | 7e-07 | 1 | 0.001 |
| dna | multiclass | 2000 | 180 | 3 | 0.001 | 1e-07 | 1e-08 | 1 | 0.001 |
| letter | multiclass | 15000 | 16 | 26 | 1 | 0.001 | 0.001 | 0.5 | 0.001 |
| pendigits | multiclass | 7494 | 16 | 10 | 0.01 | 0.0001 | 0.0001 | 1 | 0.0001 |
| satimage | multiclass | 4435 | 36 | 6 | 1 | 0.001 | 0.001 | 1 | 0.001 |
| Sensorless | multiclass | 58509 | 48 | 11 | 10 | 1e-06 | 1e-07 | 1 | 0.001 |
| shuttle | multiclass | 43500 | 48 | 11 | 10 | 0.001 | 0.001 | 0.5 | 0.001 |
| usps | multiclass | 7291 | 256 | 10 | 0.1 | 0.0001 | 0.0001 | 0.5 | 0.001 |
| mnist | multiclass | 60000 | 784 | 10 | 0.1 | 1e-05 | 7e-07 | 0.5 | 0.001 |

Table 4: Classification accuracy (%) for classification datasets. We bold the results with the best method and underline the ones that are not significantly worse than the best one.

| Dataset | Compared methods | | | FedNewton | | | |
|---|---|---|---|---|---|---|---|
| | DKRR | FedAvg | FedProx | # 1 | # 2 | # 4 | # 8 |
| dna | 90.91±0.50 | 91.09±0.42 | 89.42±6.98 | **92.23±0.53** | 91.96±0.48 | 92.02±0.40 | 88.19±11.58 |
| letter | 77.18±0.12 | 77.11±0.17 | 77.17±0.12 | 77.30±0.12 | 77.30±0.12 | **77.30±0.12** | 77.30±0.12 |
| pendigits | 97.12±0.09 | 97.12±0.11 | 97.12±0.10 | 97.29±0.13 | **97.31±0.10** | 97.31±0.11 | 97.23±0.31 |
| satimage | 87.70±0.17 | 87.84±0.08 | 87.74±0.11 | **88.49±0.19** | 88.26±0.17 | 88.31±0.14 | 88.31±0.15 |
| Sensorless | 96.81±0.12 | 96.87±0.14 | 96.84±0.13 | **97.32±0.11** | 96.87±0.14 | 96.44±0.17 | 84.43±1.20 |
| shuttle | 98.46±0.06 | 98.53±0.08 | 98.51±0.07 | **98.54±0.07** | 98.51±0.07 | 98.50±0.07 | 98.44±0.16 |
| usps | 92.95±0.10 | 92.95±0.12 | 92.95±0.12 | **93.49±0.18** | 93.24±0.13 | 93.28±0.14 | 93.30±0.15 |
| mnist | 95.38±0.12 | 95.40±0.13 | 95.46±0.11 | **95.53±0.13** | 95.48±0.13 | 95.49±0.13 | 95.48±0.12 |

FedAvg but worse than `FedNewton`. 3) Compared to FedProx and `FedNewton`, the convergence of FedAvg is slow and achieves the performance between DKRR and FedProx.

## B.2 Evaluation Results on Real Datasets

We compared related federated learning algorithms on both original datasets and non-iid datasets partitioned by a Dirichlet distribution $\text{Dir}_K(0.5)$. After partitioning with a Dirichlet distribution, the labels and the number of local samples on datasets are very unbalanced that decrease the generalization ability of federated learning algorithms. We report the classification accuracy in Table 4 for several public classification datasets, illustrating that:

1) The proposed `FedNewton` remarkably outperforms the compared methods on the original datasets, and more iterations improves the generalization performance. This observation coincides with results in Theorem 3.

2) The predictive accuracies of all federated learning methods in the heterogeneous setting are worse than ones in the original case, but `FedNewton` approaches still achieve the optimal results on the most datasets.

3) Similar to Figure 4, `FedNewton` with more iterations are more sensitive to the heterogeneity and more iterations hurts the generalization performance. The reason is the number of iterations augments the federated error when $\Upsilon > 1$ due to large data heterogeneity.

## C Experiments on Large Pretrained Models

In this section, we evaluate the proposed `FedNewton` method (Algorithm 2) in a federated fine-tuning setting, where each client updates only a small linear classification head on top of a large frozen pretrained vision backbone (e.g., ResNet or ViT). The code used to produce all experimental results is available at the following anonymous repository: `https://anonymous.4open.science/r/FedNewton-78B4/`.

## C.1 EXPERIMENTAL SETTINGS

**Datasets.** We consider MNIST, Fashion-MNIST, CIFAR-10 and CIFAR-100, . MNIST and Fashion-MNIST contain 60K training and 10K test grayscale images of size $28 \times 28$ and 10 classes. CIFAR-10/100 contain 50K training and 10K test RGB images of size $32 \times 32$ with 10 and 100 classes, respectively. For CIFAR-10/100 we apply standard data augmentation: random crop with 4-pixel padding, random horizontal flip, conversion to tensor and per-channel normalization using the usual dataset statistics, followed by resizing to $224 \times 224$ with bicubic interpolation to match the input size of pretrained backbones. For MNIST and Fashion-MNIST we resize images to $224 \times 224$, convert them to tensors, normalize with mean and standard deviation $0.5$, and replicate the single channel to form 3-channel inputs at run time. We report test accuracy and cross-entropy loss on the full official test sets.

**Federated data partition.** We simulate heterogeneous clients via a label-skewed non-IID partition based on a Dirichlet distribution. For each class $c \in \{1, \dots, C\}$, we sample a proportion vector

$$\boldsymbol{\pi}^{(c)} \sim \text{Dirichlet}(\alpha \mathbf{1}_m),$$

where $m$ is the number of clients and $\alpha > 0$ controls the degree of data heterogeneity. The indices of class $c$ are then split across clients according to $\boldsymbol{\pi}^{(c)}$, and each client's local dataset is formed by taking the union of its assigned indices over all classes.

For each dataset, we consider Dirichlet splits with concentration parameters $\alpha \in \{0.1, 0.5, 10\}$ to systematically vary the level of heterogeneity:

- $\alpha = 0.1$ induces highly non-IID partitions with strong label skew,

- $\alpha = 0.5$ corresponds to moderate heterogeneity,

- $\alpha = 10$ yields partitions that are close to IID.

**Model and training regime.** We use ImageNet-pretrained backbones loaded either from the CLIP library or the `timm` model zoo, with the original classification layer removed (`num_classes = 0`). On top of the frozen feature extractor $\phi_{\text{backbone}}$, we add a single linear classification head

$$f_\theta(x) = W \phi_{\text{backbone}}(x) + b,$$

where $W \in \mathbb{R}^{C \times d}$ and $b \in \mathbb{R}^C$ are trainable and $d$ is the feature dimension of the backbone output. **Crucially, we *only* fine-tune the linear head and keep the backbone parameters fixed.** In practice, all backbone parameters have `requires_grad = False` and are kept in evaluation mode, while the head is the only module in training mode.

**Federated setup and optimization.** We consider $m = 10$ clients, sample a fraction $\gamma = 0.5$ of clients (i.e., 5 clients) per communication round, and train $T = 50$ rounds. Each participating client performs one local epoch using mini-batches of size 64.

We compare the proposed `FedNewton` (Algorithm 2) with the following methods for optimizing the linear head under identical data partitions and model architectures:

- **First-order FL baselines:** FedAvg McMahan et al. (2017) and FedProx Li et al. (2020b).

- **Second-order FL baselines:** FedSophia Elbakary et al. (2024) and FedNew Elgabli et al. (2022).

All methods start from the same pretrained backbone with a randomly initialized linear head. Since only the head is trainable, all gradient and Hessian computations are restricted to the head parameters. We use cross-entropy as the local objective and optimize with SGD with momentum 0.9. Unless otherwise specified, we adopt the following default hyperparameters: local learning rate $\eta_{\text{local}} = 0.01$; Newton-step learning rate $\eta_{\text{newton}} = 0.1$ for FedNewton; server learning rate $\eta_{\text{sophia}} = 0.05$ for FedSophia and FedNew; damping $\lambda = 10^{-4}$ in FedNewton and $\mu = 10^{-2}$ in FedProx; Hessian smoothing parameter $\rho = 0.04$ and momentum coefficients $\beta_1 = 0.9$, $\beta_2 = 0.99$ for FedSophia/FedNew. The default solver for `FedNewton` is the exact Hessian.

Table 5: Test accuracy comparison on various datasets with different non-IID settings ($\alpha$). The reported results represent the average test accuracy (%) over the last 5 communication rounds (mean $\pm$ std). We repeat the experiments with three different random seeds.

| Dataset | $\alpha$ | FedAvg | FedProx | FedSophia | FedNew | FedNewton |
|---|---|---|---|---|---|---|
| **MNIST** | 0.1 | $85.27 \pm 0.69$ | $85.49 \pm 0.63$ | $89.37 \pm 0.96$ | $89.30 \pm 1.04$ | $\mathbf{94.50 \pm 0.71}$ |
| | 0.5 | $93.60 \pm 0.31$ | $93.56 \pm 0.30$ | $93.82 \pm 0.36$ | $93.78 \pm 0.37$ | $\mathbf{97.82 \pm 0.06}$ |
| | 10 | $94.79 \pm 0.04$ | $94.71 \pm 0.05$ | $94.37 \pm 0.05$ | $94.32 \pm 0.05$ | $\mathbf{97.86 \pm 0.03}$ |
| **Fashion-MNIST** | 0.1 | $76.36 \pm 2.26$ | $76.81 \pm 1.99$ | $77.62 \pm 2.58$ | $77.89 \pm 2.14$ | $\mathbf{82.32 \pm 0.59}$ |
| | 0.5 | $84.41 \pm 0.95$ | $84.38 \pm 0.92$ | $84.90 \pm 0.50$ | $84.86 \pm 0.52$ | $\mathbf{89.11 \pm 0.02}$ |
| | 10 | $86.52 \pm 0.05$ | $86.44 \pm 0.06$ | $86.38 \pm 0.05$ | $86.31 \pm 0.01$ | $\mathbf{89.24 \pm 0.05}$ |
| **CIFAR-10** | 0.1 | $77.98 \pm 1.98$ | $78.14 \pm 1.88$ | $80.87 \pm 1.90$ | $81.04 \pm 1.95$ | $\mathbf{86.35 \pm 0.39}$ |
| | 0.5 | $83.35 \pm 0.12$ | $83.33 \pm 0.10$ | $83.68 \pm 0.33$ | $83.60 \pm 0.42$ | $\mathbf{87.51 \pm 0.08}$ |
| | 10 | $85.22 \pm 0.02$ | $85.17 \pm 0.03$ | $84.83 \pm 0.02$ | $84.75 \pm 0.03$ | $\mathbf{87.93 \pm 0.07}$ |

**Backbone architectures.** In all experiments, we adopt large pretrained vision backbones and only fine-tune a small linear head on top of their frozen features. As shown in Table 7, we consider two convolutional networks with ResNet-18 and ResNet-50 He et al. (2016), and two ViT-Base models pretrained with DINOv2 Oquab et al. (2024) and CLIP Radford et al. (2021), respectively. All backbones are initialized from publicly available pretrained weights and kept frozen throughout training; only the linear classification head is updated by federated optimization.

**Implementation and metrics.** All experiments are implemented in PyTorch with fixed seeds for Python, NumPy, and PyTorch. For FedNewton, we follow the procedure in Algorithm 2 and *simulate* federated learning on a single GPU by executing clients sequentially rather than in parallel. In each communication round, every participating client first performs the local gradient computation phase and sends its gradient to the server; after the server broadcasts the aggregated (global) gradient, each client enters the second phase, where it constructs the (damped) local Hessian and computes the Newton update step. Thus, each round consists of *two sequential passes over all selected clients* (corresponding exactly to the two phases in the algorithm), followed by the server aggregation. Consequently, the reported wall-clock time per round is the total time for these two client-side phases across all clients plus the subsequent server update in that round.

After each communication round, we evaluate the global model on the full test set and record test accuracy, test loss, cumulative wall-clock time, and peak GPU memory usage. Training is stopped early if the test loss becomes numerically unstable (e.g., $> 10$) or extremely small (e.g., $< 10^{-6}$). We log per-round statistics and hyperparameters in NumPy archives for reproducibility.

## C.2 COMPARISON EXPERIMENTS

In this experiment, we adopt *ResNet-18* as the backbone network and compare the proposed `FedNewton` method in Algorithm 2 against several representative federated learning baselines, including FedAvg, FedProx, FedSophia, and FedNew. We run 50 communication rounds on three datasets: MNIST, Fashion-MNIST, and CIFAR-10. To systematically study the impact of data heterogeneity, we partition the data across clients using a Dirichlet distribution with concentration parameters $\alpha \in \{0.1, 0.5, 10\}$, where a smaller $\alpha$ indicates a more non-IID client data distribution. For each combination of dataset, method, and $\alpha$, we perform three independent runs with different random seeds and report the test accuracy, wall-clock time, and maximum GPU memory usage.

**Performance comparison.** Across all datasets and non-IID levels, FedNewton consistently matches or outperforms all baselines in terms of the average test accuracy over the last five epochs in Table 5. On MNIST, Fashion-MNIST, and CIFAR-10, it achieves the highest accuracy under all scenarios, including highly heterogeneous ($\alpha = 0.1$), moderately non-IID ($\alpha = 0.5$), and nearly IID ($\alpha = 10$) settings, with gains of more than 5 percentage points in the most heterogeneous cases compared to the strongest Newton-type competitors (e.g., FedSophia and FedNew). Moreover, Fed-Newton exhibits markedly lower variance across random seeds on Fashion-MNIST and CIFAR-10,

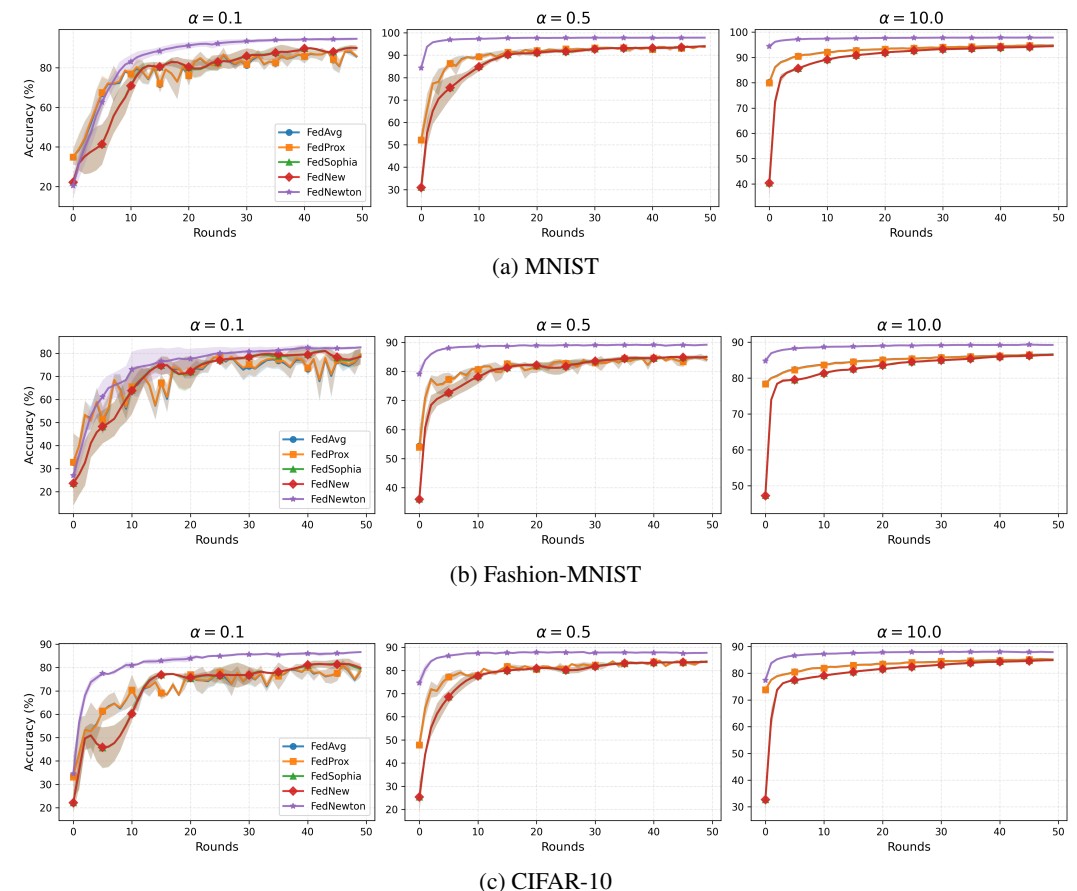

Figure 6: Test accuracy vs. communication rounds on the backbone network *Resnet-18*.

Table 6: Comparison of average training time per round (seconds) and peak GPU memory usage (MB). Results are averaged across all $\alpha$ settings. Lower is better.

| Dataset | Metric | FedAvg | FedProx | FedSophia | FedNew | FedNewton |
|---|---|---|---|---|---|---|
| **MNIST** | Time (s) | 57.89 | 55.71 | 54.15 | **53.34** | 69.05 |
| | GPU (MB) | 1507.91 | 1507.85 | 1508.08 | 1508.08 | **1463.39** |
| **Fashion-MNIST** | Time (s) | 55.91 | 53.97 | **53.29** | 53.39 | 68.96 |
| | GPU (MB) | 1507.91 | 1507.78 | 1508.08 | 1508.08 | **1463.39** |
| **CIFAR-10** | Time (s) | 68.60 | 65.57 | 55.62 | **52.18** | 90.18 |
| | GPU (MB) | 1482.33 | 1482.33 | 1482.45 | 1482.45 | **1438.89** |

suggesting that incorporating second-order curvature information yields not only higher accuracy but also more stable performance in challenging low-$\alpha$ regimes.

Figure 6 shows the test accuracy over communication rounds on MNIST, Fashion-MNIST, and CIFAR-10 under different Dirichlet concentration parameters $\alpha \in \{0.1, 0.5, 10\}$. Across all datasets and heterogeneity levels, FedNewton converges faster and attains higher final accuracy than all compared baselines (FedAvg, FedProx, FedSophia, and FedNew), with the advantage being most pronounced in the highly heterogeneous setting $\alpha = 0.1$, where the baselines converge more slowly and exhibit more oscillatory behavior. These results indicate that integrating global gradient and local Hessain information enables FedNewton to better mitigate client drift, leading to faster, more stable, and more accurate training across diverse federated learning scenarios.

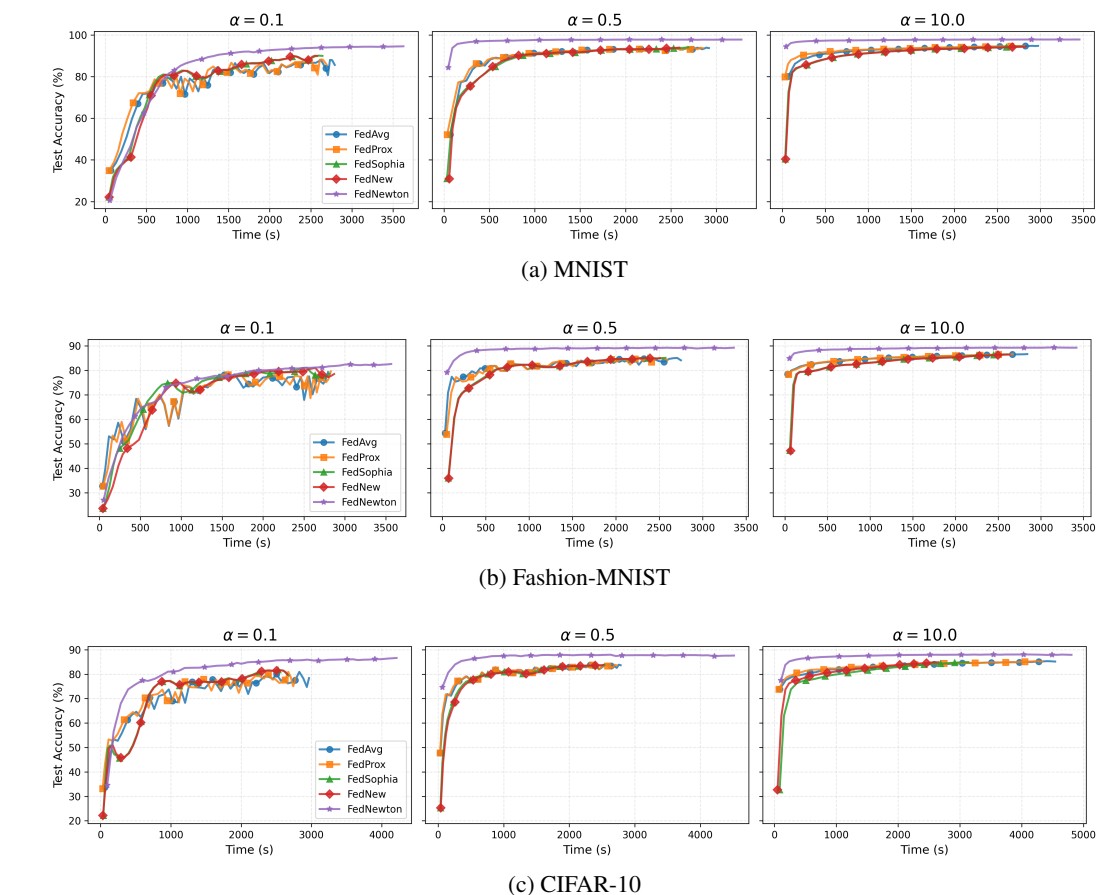

Figure 7: Test Accuracy vs. Wall-clock Time on the backbone *Resnet-18*.

Table 7: Summary of pretrained backbones used in our experiments.

| Identifier | Name | Arch. | Feat. dim | Params (M) | Pretraining scale |
|---|---|---|---|---|---|
| resnet18 | ResNet-18 | ResNet | 512 | $\sim 11$ | ImageNet-1K |
| tv_resnet50 | ResNet-50 | ResNet | 2048 | $\sim 25$ | ImageNet-1K |
| vit_base_patch14_dinov2.lvd142m | DINOv2 | ViT | 768 | $\sim 86$ | $\sim 142M$ |
| vit_base_patch16_clip_224 | CLIP | ViT | 768 | $\sim 86$ | $\sim 400M$ |

**Efficiency Comparison**   Table 6 summarizes the average training time per round and peak GPU memory usage of all methods on MNIST, Fashion-MNIST, and CIFAR-10. Relative to the fastest baseline (FedNew), FedNewton entails a moderate increase in wall-clock time (about $20\%$–$25\%$ on MNIST and Fashion-MNIST, and larger on CIFAR-10), but consistently achieves the lowest peak GPU memory consumption, reducing usage by roughly $40$–$50$ MB across all datasets. Together with the superior accuracy reported in Tables 5–6, these results indicate that FedNewton provides a favorable accuracy–efficiency trade-off and is particularly attractive in federated learning scenarios where GPU memory is a primary constraint.

Figure 7 depicts test accuracy as a function of wall-clock time on MNIST, Fashion-MNIST, and CIFAR-10 under different Dirichlet concentration parameters $\alpha \in \{0.1, 0.5, 10\}$. Across all datasets and non-IID levels, FedNewton (purple curve) consistently outperforms the baselines in terms of accuracy at most time points. In the highly heterogeneous setting $\alpha = 0.1$, FedNewton reaches higher accuracy much earlier and follows a smoother trajectory, whereas the baselines converge more slowly and exhibit pronounced oscillations. As the data distribution becomes more balanced ($\alpha = 0.5$ and $\alpha = 10$), all methods converge faster, but FedNewton still attains the best final

Table 8: Comparison of backbone models on CIFAR-10 in terms of best accuracy, the number of rounds and wall-clock time to reach a cutoff accuracy of 95%, and peak GPU memory usage.

| Backbone Model | Best Acc. (%) | Rounds@95% | Time@95% (s) | Peak GPU Mem. |
|---|---|---|---|---|
| ResNet-18 | 86.77±0.18 | > Max | 4647±98 | 1439 MB |
| ResNet-50 | 91.60±0.01 | > Max | 4845±53 | 12623 MB |
| CLIP (ViT-Base) | 96.43±0.11 | 7.3 | 916±57 | 7373 MB |
| DINOv2 (ViT-Base) | 98.07±0.30 | 5.3 | 794±58 | 7732 MB |

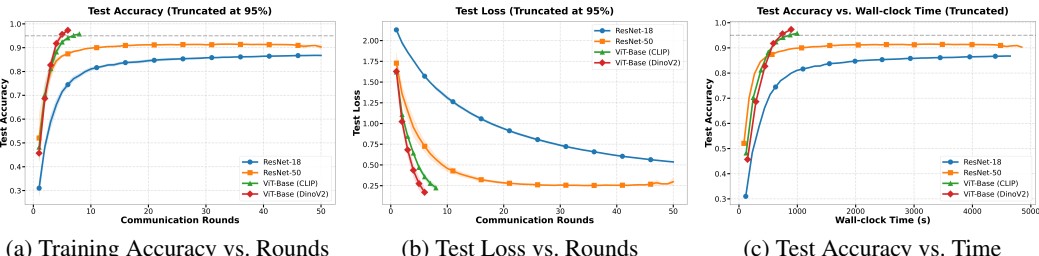

(a) Training Accuracy vs. Rounds     (b) Test Loss vs. Rounds     (c) Test Accuracy vs. Time

Figure 8: Comparison of different backbones (ResNet-18, ResNet-50, CLIP and DinoV2) on CIFAR-10 (truncated at 95%).

accuracy and remains on or above the competing curves. These observations demonstrate that Fed-Newton achieves both faster time-to-accuracy and higher final performance.

### C.3  DIFFERENT BACKBONES

We further evaluate `FedNewton` under the standard pretrain–then–finetune paradigm by freezing a powerful vision backbone and only optimizing a linear head in the federated setting. On CIFAR-10, we consider four representative backbones as fixed feature extractors shared across clients in Table 7: ResNet-18, ResNet-50, a CLIP-based ViT, and a DINOv2-based ViT. The data is partitioned among clients using a Dirichlet distribution with concentration parameter $\alpha = 0.5$ to induce moderate statistical heterogeneity. For `FedNewton`, we set the server-side Newton step size to $\eta_N = 1e - 2$ and the damping parameter to $\lambda = 1e - 5$. In this configuration, `FedNewton` only updates the linear head on top of strong pretrained features, yielding efficient second-order optimization in the federated setting.

From Table 8, we observe a clear monotonic relationship between backbone strength and the performance of `FedNewton` under the head-only federated fine-tuning setting. On CIFAR-10 with $\alpha = 0.5$, ResNet-18 and ResNet-50 achieve substantially lower best test accuracies than CLIP and DINOv2, and they do not even reach 95% accuracy within the given communication budget. In contrast, when using CLIP- and DINOv2-based ViT backbones as frozen feature extractors, `FedNewton` attains best accuracies in the 96%–98% range and crosses the 95% threshold in only a few rounds. This demonstrates that, once the backbone provides sufficiently rich and linearly separable representations, second-order optimization on top of a simple linear head is enough for `FedNewton` to fully exploit the pretrained model and deliver both higher accuracy and better accuracy–round trade-offs than with weaker CNN backbones.

The Figure 8 further illustrate the advantages of strong ViT backbones in terms of convergence speed and wall-clock efficiency. In the test accuracy and loss versus communication rounds plots, CLIP and DINOv2 exhibit much faster convergence than ResNet-18 and ResNet-50: their test losses drop sharply within the first few rounds, and their accuracies exceed 95% after roughly 5–8 rounds, whereas the ResNet curves increase more slowly and saturate at noticeably lower levels. When plotting test accuracy against wall-clock time, the ViT-based backbones remain preferable despite their higher per-round computational cost, because the total time required to reach high accuracy is significantly smaller than for ResNet backbones. Overall, these results show that, `FedNewton` is particularly well aligned with the pretrain–then–finetune paradigm: combining a frozen, high-capacity

Table 9: Efficient Solvers on CIFAR-10 with ResNet-18 backbone (Threshold: 88%)

| Method | Best Acc (%) | Rounds | Time (s) | GPU Mem |
|---|---|---|---|---|
| L-BFGS | 71.41±7.40 | >Max | 1263±547 | 1439 MB |
| Low-Rank | 76.35±0.03 | >Max | 5430±324 | 1439 MB |
| CG | 86.90±0.05 | >Max | 5860±93 | 1439 MB |
| Exact Hessian | 88.15±0.09 | 30.0 | 2939±1011 | 1439 MB |

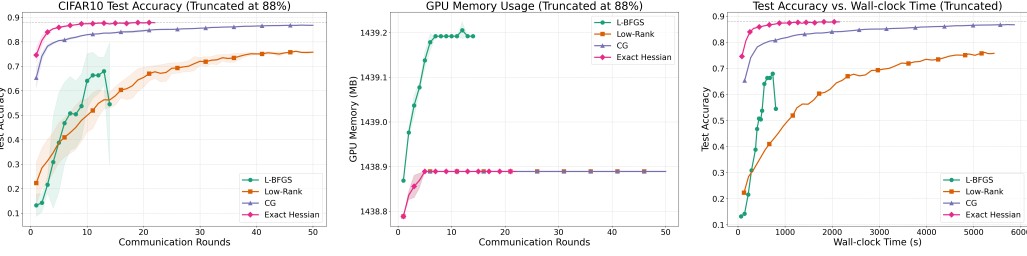

(a) Training Accuracy vs. Rounds  (b) GPU Memory Usage vs. Rounds  (c) Test Accuracy vs. Time

Figure 9: Comparison of different solvers for FedNewton (L-BFGS, Low-Rank, CG and Exact Hessian) with the backbone ResNet-18 on CIFAR-10 (truncated at 90%).

backbone with second-order optimization of a linear head yields fast, communication-efficient improvements in federated learning.

## C.4 EFFICIENT SOLVERS

As discussed in Section A, to avoid the high memory cost $\mathcal{O}(d^2)$ and computational complexity $\mathcal{O}(d^3)$ of exact Newton solvers, we implement several efficient solvers for `FedNewton`, including CG, L-BFGS, Diagonal, and low-rank variants.

We evaluate these solvers on two settings: CIFAR-10 with a ResNet-18 backbone, and the more challenging CIFAR-100 dataset with a DinoV2 backbone.

**Efficient solvers for `FedNewton` on CIFAR-10.** A main bottleneck of Newton-type methods in federated learning is how to apply the local Hessian under realistic memory and time budgets. Table 9 and Figure 9 compare several solvers for the Newton system on CIFAR-10 with a ResNet-18 backbone. *Exact Hessian* explicitly forms and inverts the full head Hessian on each client and serves as a strong reference in terms of convergence speed and final accuracy: it reaches the 88% threshold in about 30 rounds and attains the best performance, but incurs the highest wall-clock time due to expensive second-order computations.

The approximate solvers trade accuracy for efficiency to different degrees. *Conjugate Gradient (CG)* solves the Newton system using only Hessian–vector products, avoiding explicit Hessian construction while preserving rich curvature information; its curves closely track exact Newton but do not cross the 88% threshold within the given rounds. *Low-Rank* replaces the full Hessian with a low-rank approximation and shares a similar memory footprint with CG, yet its accuracy quickly saturates around 76%, indicating underfitting. *L-BFGS* is the most time-efficient in early rounds but plateaus near 71% and remains far below CG and exact Newton. Overall, CG offers the best balance between computation and performance when exact Hessian inversion is feasible but costly, whereas overly aggressive approximations (very low rank or very small L-BFGS history) substantially diminish the benefits of second-order information in `FedNewton`.

**Efficient solvers for `FedNewton` on CIFAR-100 with DinoV2.** Table 10 and Figure 10 evaluate different Newton-system solvers on the harder CIFAR-100 task using a DinoV2 backbone. Here, forming the exact Hessian is no longer feasible: the *Exact Hessian* baseline runs out of GPU memory

Table 10: Efficient Solvers on CIFAR-100 with DinoV2 backbone (Threshold: 90%)

| Method | Best Acc (%) | Rounds | Time (s) | GPU Mem |
|---|---|---|---|---|
| L-BFGS | 90.52±0.02 | 34.0 | 3615±785 | 8758 MB |
| Low-Rank | 74.61±0.62 | >Max | 8459±14 | 8258 MB |
| CG | 89.73±0.18 | >Max | 8074±16 | 8258 MB |
| Exact Hessian | / | / | / | OOM |

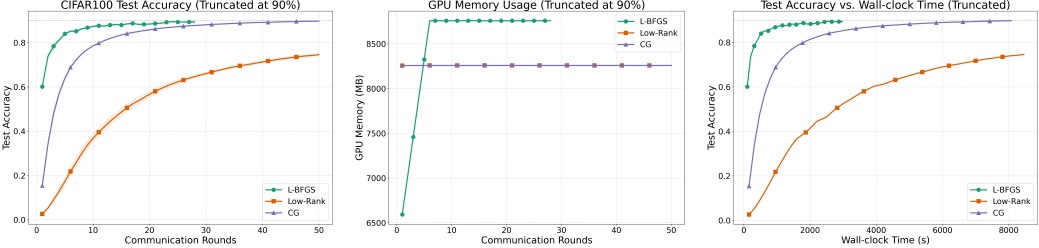

(a) Training Accuracy vs. Rounds    (b) GPU Memory Usage vs. Rounds    (c) Test Accuracy vs. Time

Figure 10: Comparison of different solvers for FedNewton (L-BFGS, Low-Rank and CG) with the backbone DinoV2 on CIFAR-100 (truncated at 90%).

(OOM), highlighting the memory bottleneck of naive second-order methods even when only a linear head is updated on top of a strong pretrained backbone. Among the approximate solvers, *L-BFGS* clearly stands out: it reaches the $90\%$ accuracy threshold in about $34$ rounds, attains the best accuracy of $90.52\% \pm 0.02$, and achieves the smallest time-to-target.

The curves further show that these gains do not come from extra memory. After a short warm-up, L-BFGS stabilizes around $\sim 8.8$ GB of GPU memory, comparable to or only slightly higher than *CG* and *Low-Rank*. In contrast, CG converges more slowly and plateaus around $89.73\%$, while the Low-Rank solver saturates near $74.61\%$ and clearly underfits, despite similar memory usage and higher wall-clock time. Overall, on this more challenging dataset with a strong DinoV2 backbone, an L-BFGS-style inverse-Hessian approximation offers the most favorable trade-off between accuracy, communication rounds, wall-clock time, and memory, whereas CG and overly restrictive low-rank approximations fail to fully exploit second-order information.

### C.5 PARAMETER TUNING FOR FEDNEWTON

We first study the sensitivity of FedNewton to its key hyperparameters on MNIST with a ResNet-18 backbone. Figure 11 reports test accuracy as a function of (a) communication rounds and (b) wall-clock time for different combinations of the Newton-step learning rate $\eta_{\text{newton}}$ (nLR) and damping $\lambda$. In all settings, we keep the local optimizer and other training configurations fixed, and vary only $(\eta_{\text{newton}}, \lambda)$.

The results show that FedNewton is robust within a reasonable range, but the choice of $(\eta_{\text{newton}}, \lambda)$ has a clear impact on both convergence speed and final accuracy. Very small Newton learning rates (e.g., $\eta_{\text{newton}} = 0.01$) lead to slow progress in early rounds, while overly aggressive learning rates combined with weak damping (e.g., $\eta_{\text{newton}} = 1.0$ with $\lambda = 10^{-4}$) cause unstable or highly oscillatory behavior. Moderate settings such as $\eta_{\text{newton}} = 0.1$ with $\lambda \in [10^{-4}, 10^{-3}]$ achieve the best trade-off: they reach high test accuracy within a few communication rounds and also dominate in terms of wall-clock time. Based on these observations, we adopt $\eta_{\text{newton}} = 0.1$ and $\lambda = 10^{-4}$ as the default configuration in our main experiments.

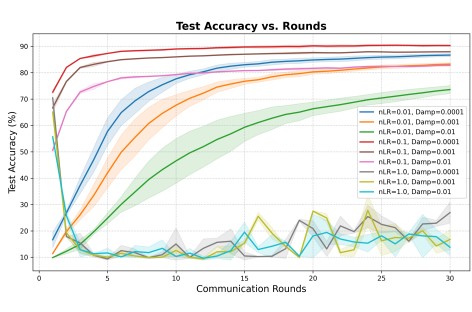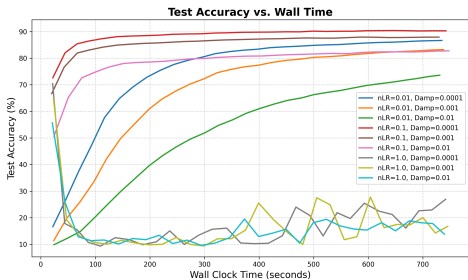

(a) Training Accuracy vs. Rounds                  (b) Training Accuracy vs. Wall-clock Time

Figure 11: Parameter tuning for FedNewton with the backbone ResNet-18 on MNIST.

## PROOFS

### C.6   PRELIMINARIES

Since KRR has closed-form solutions, the intermediate estimators $\bar{f}^t_{\mathcal{D},\lambda}, f_{\mathcal{D},\lambda}, f_\lambda, f^*$ in error decomposition can be represented by the redirection operators and their adjoint operators. In this section, we first provide useful linear operators associated with kernel $K$. Then, we measure the similarities between empirical and expected covariance operators via concentration inequalities.

**Definition 2** (Operators with kernel $K$ in terms of the global distribution $\rho_{X \times Y}$). *For any $\boldsymbol{x} \in \boldsymbol{X}, g \in L^2(\mathbb{P}), \phi : \boldsymbol{X} \to \mathcal{H}_K$ and $\boldsymbol{\beta} \in \mathcal{H}_K$, we define the following expected operators*

- $S : \mathcal{H}_K \to L^2(\mathbb{P}), \quad (S\boldsymbol{\beta})(\boldsymbol{x}) = \langle \boldsymbol{\beta}, \phi(\boldsymbol{x}) \rangle.$

- $S^* : L^2(\mathbb{P}) \to \mathcal{H}_K, \quad S^*g = \int_X \phi(\boldsymbol{x})g(\boldsymbol{x})d\rho_X(\boldsymbol{x}).$

- $L : L^2(\mathbb{P}) \to L^2(\mathbb{P}), \quad L = SS^*, \quad$ *such that* $(Lg)(\cdot) = \int_X \langle \phi(\cdot), \phi(\boldsymbol{x}) \rangle g(\boldsymbol{x})d\rho_X(\boldsymbol{x}).$

- $C : \mathcal{H}_K \to \mathcal{H}_K, \quad C = S^*S, \quad$ *such that* $C\boldsymbol{\beta} = \int_X \langle \boldsymbol{\beta}, \phi(\boldsymbol{x}) \rangle \phi(\boldsymbol{x})d\rho_X(\boldsymbol{x}).$

**Definition 3** (Empirical operators on the global dataset $\mathcal{D}$ and local datasets $\mathcal{D}_j$). *For any $\phi : \boldsymbol{X} \to \mathcal{H}_K$ and $\boldsymbol{\beta} \in \mathcal{H}_K$, we define the following empirical operators*

- $S_{\mathcal{D}} : \mathcal{H}_K \to \mathbb{R}^{|\mathcal{D}|}, \quad S_{\mathcal{D}}\boldsymbol{\beta} = \left( \langle \boldsymbol{\beta}, \phi(\boldsymbol{x}_i) \rangle \right)_{i=1}^{|\mathcal{D}|} \in \mathbb{R}^{|\mathcal{D}|}, \quad \forall (\boldsymbol{x}_i, y_i) \in \mathcal{D}.$

- $S_{\mathcal{D}}^* : \mathbb{R}^{|\mathcal{D}|} \to \mathcal{H}_K, \quad S_{\mathcal{D}}^*\alpha = \frac{1}{|\mathcal{D}|}\sum_{i=1}^{|\mathcal{D}|} \phi(\boldsymbol{x}_i)\alpha_i \in \mathcal{H}_K, \quad \forall (\boldsymbol{x}_i, y_i) \in \mathcal{D}, \alpha \in \mathbb{R}^{|\mathcal{D}|}.$

- $C_{\mathcal{D}} : \mathcal{H}_K \to \mathcal{H}_K, \quad C_{\mathcal{D}} = S_{\mathcal{D}}^*S_{\mathcal{D}},$ *such that* $C_{\mathcal{D}} \boldsymbol{\beta} = \frac{1}{|\mathcal{D}|}\sum_{i=1}^{|\mathcal{D}|} \langle \boldsymbol{\beta}, \phi(\boldsymbol{x}_i) \rangle \phi(\boldsymbol{x}_i), \quad \forall (\boldsymbol{x}_i, y_i) \in \mathcal{D}.$

- $S_{\mathcal{D}_j} : \mathcal{H}_K \to \mathbb{R}^{|\mathcal{D}_j|}, \quad S_{\mathcal{D}_j}\boldsymbol{\beta} = \left( \langle \boldsymbol{\beta}, \phi(\boldsymbol{x}_i) \rangle \right)_{i=1}^{|\mathcal{D}_j|} \in \mathbb{R}^{|\mathcal{D}_j|}, \quad \forall (\boldsymbol{x}_i, y_i) \in \mathcal{D}_j.$

- $S_{\mathcal{D}_j}^* : \mathbb{R}^{|\mathcal{D}_j|} \to \mathcal{H}_K, \quad S_{\mathcal{D}_j}^*\alpha = \frac{1}{|\mathcal{D}_j|}\sum_{i=1}^{|\mathcal{D}_j|} \phi(\boldsymbol{x}_i)\alpha_i \in \mathcal{H}_K, \quad \forall (\boldsymbol{x}_i, y_i) \in \mathcal{D}_j.$

- $C_{\mathcal{D}_j} : \mathcal{H}_K \to \mathcal{H}_K, \quad C_{\mathcal{D}_j} = S_{\mathcal{D}_j}^*S_{\mathcal{D}_j}, \quad$ *such that* $C_{\mathcal{D}_j} \boldsymbol{\beta} = \frac{1}{|\mathcal{D}_j|}\sum_{i=1}^{|\mathcal{D}_j|} \langle \boldsymbol{\beta}, \phi(\boldsymbol{x}_i) \rangle \phi(\boldsymbol{x}_i), \quad \forall (\boldsymbol{x}_i, y_i) \in \mathcal{D}_j.$

Here, we denote $S$ the inclusion operator and $S_{\mathcal{D}}, S_{\mathcal{D}_j}$ the sampling operator, while $S^*, S_{\mathcal{D}}^*, S_{\mathcal{D}_j}^*$ are their adjoint operators. Note that $C : \mathcal{H}_K \to \mathcal{H}_K$ is the covariance operator given by $S^*S$, and the integral operator $L : L^2(\mathbb{P}) \to L^2(\mathbb{P})$ given by $SS^*$. The kernel matrix $\mathbf{K}_{\mathcal{D}}, \mathbf{K}_{\mathcal{D}}$ and the covariance matrix $C_{\mathcal{D}}, C_{\mathcal{D}_j}$ are the empirical counterparts of the integral operator $L$ and the covariance operator $C$, respectively. Using Singular Value Decomposition shows that $L$ and $C$ have the same eigenvalues, and the corresponding eigenvectors are closely related Rosasco et al. (2010). Those kernels-related operators are widely used in the proof of optimal learning theory

for standard KRR. Assuming the kernel is bounded $K(\boldsymbol{x}, \boldsymbol{x}') \leq \kappa^2$, the integral operator $L$ and the covariance operator $C$ are positive trace class operators (and hence compact) and bounded by $\|L\| = \|C\| \leq \kappa^2$. For any function $f \in \mathcal{H}_K$, the estimator $f \in L^2(\mathbb{P})$ is obtained by kernel trick. Thus, for $f(\boldsymbol{x}) = \langle \boldsymbol{w}, \phi(\boldsymbol{x}) \rangle$, the RKHS norm can be related to the $L^2(\mathbb{P})$-norm by $C^{1/2}$ Bauer et al. (2007):

$$\|f\|_2 = \|Sf\|_2 = \|C^{1/2}\boldsymbol{w}\|_K, \quad \forall \boldsymbol{w} \in \mathcal{H}_K, \ f \in L^2(\mathbb{P}). \tag{13}$$

**Remark 8.** *With the assumption $K(\boldsymbol{x}, \boldsymbol{x}') \leq \kappa^2$, the integral operator $L$ is trace class Caponnetto & De Vito (2007) and $C, C_{\mathcal{D}}, C_{\mathcal{D}_j}$ are finite dimensional. Moreover we have that $L = SS^*$, $C = S^*S$, $C_{\mathcal{D}} = S_{\mathcal{D}}^* S_{\mathcal{D}}$ and $C_{\mathcal{D}_j} = S_{\mathcal{D}_j}^* S_{\mathcal{D}_j}$. Finally $L, C, C_{\mathcal{D}}, C_{\mathcal{D}_j}$ are self-adjoint and positive operators, with spectrum is $[0, \kappa^2]$.*

**Proposition 2** (Cordes Inequality Fujii et al. (1993)). *Let $A, B$ two positive semi-definite bounded linear operators on a separable Hilbert space. Then*

$$\|A^s B^s\| \leq \|AB\|^s, \qquad when \quad 0 \leq s \leq 1.$$

Here, we use Proposition 2 to obtain the inequality $\|(A + \lambda I)^{-1/2}(B + \lambda)^{1/2}\| \leq \|(A + \lambda I)^{-1}(B + \lambda)\|^{1/2}$ for linear operators $C, C_j, C_{\mathcal{D}}, C_{\mathcal{D}_j}$, and $L$.

**Proposition 3** (Lemma 2 in Smale & Zhou (2007)). *Let $\mathcal{L}$ be a separable Hilbert space and $\{\xi_1, \cdots, \xi_n\}$ be a sequence of i.i.d random variables in $\mathcal{L}$. Assume the bound be $\|\xi_i\| \leq \widetilde{M} \leq \infty$ and the variance be $\tilde{\sigma}^2 = \mathbb{E}(\|\xi_i - \mathbb{E}(\xi_i)\|^2)$ for any $i \in [n]$. For any $\delta \in (0, 1)$, with confidence $1 - \delta$,*

$$\left\| \frac{1}{n} \sum_{i=1}^{n} \xi_i - \mathbb{E}(\xi_i) \right\| \leq \frac{2\widetilde{M} \log(2/\delta)}{n} + \sqrt{\frac{2\tilde{\sigma}^2 \log(2/\delta)}{n}}. \tag{14}$$

The above Bernstein's inequality is the key to analyzing the relationship between the empirical random vector and its expected counterpart, which is used to prove Lemma 1. The above Bernstein's inequality for random vectors was provided in Smale & Zhou (2007); Rudi & Rosasco (2017) and later was extended to the random operator case in Theorem 7.3.1 in Tropp (2012) and Lemma 24 in Lin & Cevher (2020).

**Lemma 1.** *Given $K(\boldsymbol{x}, \boldsymbol{x}') = \langle \phi(\boldsymbol{x}), \phi(\boldsymbol{x}') \rangle_K$, let $\phi(\cdot)$ be i.i.d random vectors on a separable Hilbert space $\mathcal{H}_K$ such that $C, C_{\mathcal{D}}, C_{\mathcal{D}_j}$ are trace class. Then for any $\delta \in (0, 1)$ with the probability at least $1 - \delta$, the following holds*

$$\left\| (C + \lambda I)^{-1/2}(C - C_{\mathcal{D}})(C + \lambda I)^{-1/2} \right\| \leq \mathcal{R}_{\mathcal{D}, \lambda} \leq \frac{2\kappa^2 \log(2/\delta)}{\lambda |\mathcal{D}|} + \sqrt{\frac{2(\kappa^2 + 1) \log(2/\delta)}{\lambda |\mathcal{D}|}},$$

$$\left\| (C_j + \lambda I)^{-1/2}(C_j - C_{\mathcal{D}_j})(C_j + \lambda I)^{-1/2} \right\| \leq \mathcal{R}_{\mathcal{D}_j, \lambda} \leq \frac{2\kappa^2 \log(2/\delta)}{\lambda |\mathcal{D}_j|} + \sqrt{\frac{2(\kappa^2 + 1) \log(2/\delta)}{\lambda |\mathcal{D}_j|}},$$

$$\tag{15}$$

*where $\mathcal{R}_{\mathcal{D}, \lambda} = \left\| (C + \lambda I)^{-1}(C - C_{\mathcal{D}}) \right\|$ and $\mathcal{R}_{\mathcal{D}_j, \lambda} = \left\| (C_j + \lambda I)^{-1}(C_j - C_{\mathcal{D}_j}) \right\|$.*

*Proof.* We first prove the lower bound for $\mathcal{R}_{\mathcal{D}, \lambda}$. Using the Cauchy-Schwarz inequality, we have

$$\begin{aligned}
&\left\| (C + \lambda I)^{-1/2}(C - C_{\mathcal{D}})(C + \lambda I)^{-1/2} \right\| \\
&= \left\| (C + \lambda I)^{-1/2}(C - C_{\mathcal{D}})^{1/2}(C - C_{\mathcal{D}})^{1/2}(C + \lambda I)^{-1/2} \right\| \\
&\leq \left\| (C + \lambda I)^{-1/2}(C - C_{\mathcal{D}})^{1/2} \right\|^2.
\end{aligned} \tag{16}$$

Recall that the norm on a matrix or operator $A$ can be defined By

$$\|A\| := \sup_x \frac{\|Ax\|_2}{\|x\|_2}.$$

For $K > 1$ and a nonzero vector $x$, we get

$$\|A^k x\|_2 = \|AA^{k-1}x\|_2 \le \|A\|\|A^{k-1}x\|_2 \le \cdots \le \|A\|^k\|x\|_2.$$

Therefore, it holds $\frac{\|A^k x\|_2}{\|x\|_2} \le \|A\|^k$ and thus

$$\|A^k\| = \sup_x \frac{\|A^k x\|_2}{\|x\|_2} \le \|A\|^k. \tag{17}$$

Assuming $A = (C + \lambda I)^{-1/2}$ and substituting equation 17 to equation 16, we get

$$\left\|(C + \lambda I)^{-1/2}(C - C_{\mathcal{D}})(C + \lambda I)^{-1/2}\right\| \le \left\|(C + \lambda I)^{-1}(C - C_{\mathcal{D}})\right\| = \mathcal{R}_{\mathcal{D},\lambda}.$$

Then, we prove the upper bound for $\mathcal{R}_{\mathcal{D},\lambda}$. Let $\xi = (C + \lambda I)^{-1}\phi(\boldsymbol{x}) \otimes \phi(\boldsymbol{x})$, thus we have

$$\mathbb{E}(\xi) = (C + \lambda I)^{-1}\mathbb{E}[\phi(\boldsymbol{x}) \otimes \phi(\boldsymbol{x})] = (C + \lambda I)^{-1}C,$$

$$\frac{1}{|\mathcal{D}|}\sum_{i=1}^{|\mathcal{D}|}\xi_i = \frac{1}{|\mathcal{D}|}\sum_{i=1}^{|\mathcal{D}|}(C + \lambda I)^{-1}[\phi(\boldsymbol{x}_i) \otimes \phi(\boldsymbol{x}_i)] = (C + \lambda I)^{-1}C_{\mathcal{D}}.$$

The left of the desired inequality becomes

$$\left\|(C + \lambda I)^{-1}(C - C_{\mathcal{D}})\right\| = \left\|\mathbb{E}(\xi) - \frac{1}{|\mathcal{D}|}\sum_{i=1}^{|\mathcal{D}|}\xi_i\right\|.$$

Note that

$$\|(C + \lambda I)^{-1/2}\phi(\boldsymbol{x})\|^2 \le \kappa^2\lambda^{-1}.$$

To use Bernstein's inequality (Proposition 3), we need to bound $\|\xi\|$ and $\mathbb{E}\|\xi\|^2$ as follows

$$\|\xi\| = \|\langle(C + \lambda I)^{-1}\phi(\boldsymbol{x}), \phi(\boldsymbol{x})\rangle\| \le \|(C + \lambda I)^{-1/2}\phi(\boldsymbol{x})\|^2 \le \kappa^2\lambda^{-1}.$$

$$\mathbb{E}\|\xi - \mathbb{E}(\xi)\|^2 = \left\|\mathbb{E}\left[\langle(C + \lambda I)^{-1}\phi(\boldsymbol{x}), \phi(\boldsymbol{x})\rangle(C + \lambda I)^{-1}\phi(\boldsymbol{x}) \otimes \phi(\boldsymbol{x})\right] - C_\lambda^{-2}C^2\right\|$$

$$\le \kappa^2\lambda^{-1}\left\|\mathbb{E}\left[(C + \lambda I)^{-1}\phi(\boldsymbol{x}) \otimes \phi(\boldsymbol{x})\right]\right\| + \left\|C_\lambda^{-2}C^2\right\|$$

$$\le \kappa^2\lambda^{-1}\|C_\lambda^{-1}C\| + 1 \le \kappa^2\lambda^{-1} + 1 \le (\kappa^2 + 1)\lambda^{-1}.$$

Substituting the above identities to Bernstein's inequality equation 14, we obtain the upper bound for $\mathcal{R}_{\mathcal{D},\lambda}$.

The lower and upper bounds can be proven with similar proof techniques. $\qquad\square$

**Lemma 2** (Proposition 8 Rudi & Rosasco (2017)). *Let $\lambda > 0$. We define the following quantities*

$$\mathcal{P}_{\mathcal{D},\lambda} := \left\|(C_{\mathcal{D}} + \lambda I)^{-1}(C + \lambda I)\right\|, \quad \mathcal{P}_{\mathcal{D}_j,\lambda} := \left\|(C_{\mathcal{D}_j} + \lambda I)^{-1}(C_j + \lambda I)\right\|.$$

*Then, there exists the following properties*

$$\mathcal{P}_{\mathcal{D},\lambda} \le \frac{1}{1 - \beta}, \quad \mathcal{P}_{\mathcal{D}_j,\lambda} \le \frac{1}{1 - \beta},$$

*with*

$$\beta = \lambda_{max}\left[(C + \lambda I)^{-1/2}(C - C_{\mathcal{D}})(C + \lambda I)^{-1/2}\right].$$

*Note that, $\beta \le \frac{\lambda_{max}(C)}{\lambda_{max}+\lambda} < 1$.*

### C.7 ERROR DECOMPOSITION FOR FEDNEWTON

For Newton-based federated learning, there holds the following error decompositions

$$\|\bar{f}_{\mathcal{D},\lambda}^t - f^*\| \le \|\bar{f}_{\mathcal{D},\lambda}^t - f_{\mathcal{D},\lambda}\| + \|f_{\mathcal{D},\lambda} - f^*\|. \tag{18}$$

Here, the federated error term $\|\bar{f}_{\mathcal{D},\lambda}^t - f_{\mathcal{D},\lambda}\|$ is also the key to analyzing the generalization of second-order optimization based federated learning FedNewton.

*Proof of Theorem 1.* For any function $f(\boldsymbol{x}) = \langle \boldsymbol{w}, \phi(\boldsymbol{x}) \rangle_K$, the $\mathcal{H}_K$-norm can be related to the $L^2(\mathbb{P})$-norm by the inclusion $S$ Bauer et al. (2007)

$$\|f\|_2 = \|S\boldsymbol{w}\|_K = \|S(C+\lambda I)^{-1/2}(C+\lambda I)^{1/2}\boldsymbol{w}\|_K \le \|(C+\lambda I)^{1/2}\boldsymbol{w}\|_K.$$

Therefore, one can prove

$$\|\bar{f}_{\mathcal{D},\lambda}^t - f_{\mathcal{D},\lambda}\|_2 \le \|(C+\lambda I)^{1/2}(\bar{\boldsymbol{w}}_{\mathcal{D},\lambda}^t - \boldsymbol{w}_{\mathcal{D},\lambda})\|_K. \tag{19}$$

From equation 5, we have

$$
\begin{aligned}
\bar{\boldsymbol{w}}_{\mathcal{D},\lambda}^t &= \bar{\boldsymbol{w}}_{\mathcal{D},\lambda}^{t-1} - \sum_{j=1}^m p_j \boldsymbol{H}_{\mathcal{D}_j,\lambda}^{-1} \boldsymbol{g}_{\mathcal{D},\lambda}^{t-1} \\
&= \bar{\boldsymbol{w}}_{\mathcal{D},\lambda}^{t-1} - \sum_{j=1}^m p_j (C_{\mathcal{D}_j}+\lambda I)^{-1} \left[ (C_{\mathcal{D}}+\lambda I)\bar{\boldsymbol{w}}_{\mathcal{D},\lambda}^{t-1} - S_{\mathcal{D}}^* \boldsymbol{y}_{\mathcal{D}} \right] \\
&= \sum_{j=1}^m p_j (C_{\mathcal{D}_j}+\lambda I)^{-1}(C_{\mathcal{D}_j}-C_{\mathcal{D}})\bar{\boldsymbol{w}}_{\mathcal{D},\lambda}^{t-1} + \sum_{j=1}^m p_j (C_{\mathcal{D}_j}+\lambda I)^{-1} S_{\mathcal{D}}^* \boldsymbol{y}_{\mathcal{D}} \\
&= \sum_{j=1}^m p_j (C_{\mathcal{D}_j}+\lambda I)^{-1}(C_{\mathcal{D}_j}-C_{\mathcal{D}})\bar{\boldsymbol{w}}_{\mathcal{D},\lambda}^{t-1} + \sum_{j=1}^m p_j (C_{\mathcal{D}_j}+\lambda I)^{-1}(C_{\mathcal{D}}+\lambda I)\boldsymbol{w}_{\mathcal{D},\lambda}.
\end{aligned}
$$

And then, one can obtain

$$
\begin{aligned}
&\bar{\boldsymbol{w}}_{\mathcal{D},\lambda}^t - \boldsymbol{w}_{\mathcal{D},\lambda} \\
&= \sum_{j=1}^m p_j (C_{\mathcal{D}_j}+\lambda I)^{-1}(C_{\mathcal{D}_j}-C_{\mathcal{D}})\bar{\boldsymbol{w}}_{\mathcal{D},\lambda}^{t-1} + \sum_{j=1}^m p_j (C_{\mathcal{D}_j}+\lambda I)^{-1}(C_{\mathcal{D}}+\lambda I)\boldsymbol{w}_{\mathcal{D},\lambda} - \boldsymbol{w}_{\mathcal{D},\lambda} \\
&= \sum_{j=1}^m p_j (C_{\mathcal{D}_j}+\lambda I)^{-1}(C_{\mathcal{D}_j}-C_{\mathcal{D}})\bar{\boldsymbol{w}}_{\mathcal{D},\lambda}^{t-1} + \sum_{j=1}^m p_j (C_{\mathcal{D}_j}+\lambda I)^{-1}(C_{\mathcal{D}}-C_{\mathcal{D}_j})\boldsymbol{w}_{\mathcal{D},\lambda} \\
&= \sum_{j=1}^m p_j (C_{\mathcal{D}_j}+\lambda I)^{-1}(C_{\mathcal{D}_j}-C_{\mathcal{D}})(\bar{\boldsymbol{w}}_{\mathcal{D},\lambda}^{t-1} - \boldsymbol{w}_{\mathcal{D},\lambda}).
\end{aligned}
$$

We then estimate the federated error by

$$(C + \lambda I)^{1/2}(\bar{\boldsymbol{w}}_{\mathcal{D},\lambda}^t - \boldsymbol{w}_{\mathcal{D},\lambda})$$

$$= \sum_{j=1}^m p_j(C + \lambda I)^{1/2}(C_{\mathcal{D}_j} + \lambda I)^{-1}(C_{\mathcal{D}_j} - C_{\mathcal{D}})(\bar{\boldsymbol{w}}_{\mathcal{D},\lambda}^{t-1} - \boldsymbol{w}_{\mathcal{D},\lambda})$$

$$= \sum_{j=1}^m p_j(C + \lambda I)^{1/2}(C_{\mathcal{D}_j} + \lambda I)^{-1}(C_{\mathcal{D}_j} - C_j + C_j - C + C - C_{\mathcal{D}})(\bar{\boldsymbol{w}}_{\mathcal{D},\lambda}^{t-1} - \boldsymbol{w}_{\mathcal{D},\lambda})$$

$$= \sum_{j=1}^m p_j(C + \lambda I)^{1/2}(C_j + \lambda I)^{-1/2}(C_j + \lambda I)^{1/2}(C_{\mathcal{D}_j} + \lambda I)^{-1}(C_j + \lambda I)^{1/2}$$

$$(C_j + \lambda I)^{-1/2}(C_{\mathcal{D}_j} - C_j)(C_j + \lambda I)^{-1/2}(C_j + \lambda I)^{1/2}(C + \lambda I)^{-1/2}(C + \lambda I)^{1/2}(\bar{\boldsymbol{w}}_{\mathcal{D},\lambda}^{t-1} - \boldsymbol{w}_{\mathcal{D},\lambda})$$

$$+ \sum_{j=1}^m p_j(C + \lambda I)^{1/2}(C_j + \lambda I)^{-1/2}(C_j + \lambda I)^{1/2}(C_{\mathcal{D}_j} + \lambda I)^{-1}(C_j + \lambda I)^{1/2}$$

$$(C_j + \lambda I)^{-1/2}(C_j - C)(C_j + \lambda I)^{-1/2}(C_j + \lambda I)^{1/2}(C + \lambda I)^{-1/2}(C + \lambda I)^{1/2}(\bar{\boldsymbol{w}}_{\mathcal{D},\lambda}^{t-1} - \boldsymbol{w}_{\mathcal{D},\lambda})$$

$$+ \sum_{j=1}^m p_j(C + \lambda I)^{1/2}(C_j + \lambda I)^{-1/2}(C_j + \lambda I)^{1/2}(C_{\mathcal{D}_j} + \lambda I)^{-1}(C_j + \lambda I)^{1/2}$$

$$(C_j + \lambda I)^{-1/2}(C + \lambda I)^{1/2}(C + \lambda I)^{-1/2}(C - C_{\mathcal{D}})(C + \lambda I)^{-1/2}(C + \lambda I)^{1/2}(\bar{\boldsymbol{w}}_{\mathcal{D},\lambda}^{t-1} - \boldsymbol{w}_{\mathcal{D},\lambda}). \tag{20}$$

Note that, $\|(C + \lambda I)^{1/2}(C_j + \lambda I)^{-1/2}\| \leq \|I + (C_j + \lambda I)^{-1}(C - C_j)\|^{1/2} \leq \sqrt{1 + \frac{\Delta_{\mathcal{D}_j}}{\lambda}}$, $\|(C_j + \lambda I)^{1/2}(C_{\mathcal{D}_j} + \lambda I)^{-1}(C_j + \lambda I)^{1/2}\| \leq \mathcal{P}_{\mathcal{D}_j,\lambda}$, $\|(C_j + \lambda I)^{1/2}(C + \lambda I)^{-1/2}\| \leq \|I + (C + \lambda I)^{-1}(C_j - C)\|^{1/2} \leq \sqrt{1 + \frac{\Delta_{\mathcal{D}_j}}{\lambda}}$. Therefore, substituting these inequalities to equation 20 and from equation 19, there exists

$$\|\bar{f}_{\mathcal{D},\lambda}^t - f_{\mathcal{D},\lambda}\|_2$$

$$\leq \|(C + \lambda I)^{1/2}(\bar{\boldsymbol{w}}_{\mathcal{D},\lambda}^t - \boldsymbol{w}_{\mathcal{D},\lambda})\|_K$$

$$\leq \sum_{j=1}^m p_j\left(1 + \frac{\Delta_{\mathcal{D}_j}}{\lambda}\right)\mathcal{P}_{\mathcal{D}_j,\lambda}\mathcal{R}_{\mathcal{D}_j,\lambda}\left\|(C + \lambda I)^{1/2}(\bar{\boldsymbol{w}}_{\mathcal{D},\lambda}^{t-1} - \boldsymbol{w}_{\mathcal{D},\lambda})\right\|_K$$

$$+ \sum_{j=1}^m p_j\left(1 + \frac{\Delta_{\mathcal{D}_j}}{\lambda}\right)\mathcal{P}_{\mathcal{D}_j,\lambda}\frac{\Delta_{\mathcal{D}_j}}{\lambda}\left\|(C + \lambda I)^{1/2}(\bar{\boldsymbol{w}}_{\mathcal{D},\lambda}^{t-1} - \boldsymbol{w}_{\mathcal{D},\lambda})\right\|_K$$

$$+ \sum_{j=1}^m p_j\left(1 + \frac{\Delta_{\mathcal{D}_j}}{\lambda}\right)\mathcal{P}_{\mathcal{D}_j,\lambda}\mathcal{R}_{\mathcal{D},\lambda}\left\|(C + \lambda I)^{1/2}(\bar{\boldsymbol{w}}_{\mathcal{D},\lambda}^{t-1} - \boldsymbol{w}_{\mathcal{D},\lambda})\right\|_K \tag{21}$$

$$\leq \sum_{j=1}^m p_j\mathcal{P}_{\mathcal{D}_j,\lambda}\left(1 + \frac{\Delta_{\mathcal{D}_j}}{\lambda}\right)\left(\mathcal{R}_{\mathcal{D},\lambda} + \mathcal{R}_{\mathcal{D}_j,\lambda} + \frac{\Delta_{\mathcal{D}_j}}{\lambda}\right)\left\|(C + \lambda I)^{1/2}(\bar{\boldsymbol{w}}_{\mathcal{D},\lambda}^{t-1} - \boldsymbol{w}_{\mathcal{D},\lambda})\right\|_K$$

$$\leq \left(\sum_{j=1}^m p_j\mathcal{P}_{\mathcal{D}_j,\lambda}\left(1 + \frac{\Delta_{\mathcal{D}_j}}{\lambda}\right)\left(2\mathcal{R}_{\mathcal{D}_j,\lambda} + \frac{\Delta_{\mathcal{D}_j}}{\lambda}\right)\right)^t\left\|(C + \lambda I)^{1/2}(\bar{\boldsymbol{w}}_{\mathcal{D},\lambda}^0 - \boldsymbol{w}_{\mathcal{D},\lambda})\right\|_K.$$

Note that, $\mathcal{R}_{\mathcal{D},\lambda} \propto 1/|\mathcal{D}|$ and thus $\mathcal{R}_{\mathcal{D}_j,\lambda} \leq \mathcal{R}_{\mathcal{D},\lambda}$. Combing the above inequality and equation 18, we prove the final result. □

**Proposition 4.** *The following federated error bounds hold for oneshot federated learning:*

$$\|(C + \lambda I)^{1/2}(\bar{\boldsymbol{w}}_{\mathcal{D},\lambda}^0 - \boldsymbol{w}_{\mathcal{D},\lambda})\|_K$$

$$\leq \mathcal{P}_{\mathcal{D},\lambda}\sum_{j=1}^m p_j\left(2\mathcal{R}_{\mathcal{D}_j,\lambda} + \frac{(1 + \mathcal{R}_{\mathcal{D}_j,\lambda})\Delta_{\mathcal{D}_j}}{\lambda}\right)\left\|(C + \lambda I)^{1/2}(\boldsymbol{w}_{\mathcal{D}_j,\lambda} - \boldsymbol{w}_{\lambda})\right\|_K, \tag{22}$$

where $\Delta_{\mathcal{D}_j} = \|C_j - C\|$.

*Proof.* Note that, if $A, B$ are invertible operators on a Banach space, then there holds the equality

$$A^{-1} - B^{-1} = B^{-1}(B - A)A^{-1} = A^{-1}(B - A)B^{-1}.$$

From equation 2, using the facts $S_{\mathcal{D}}^* \boldsymbol{y}_{\mathcal{D}} = \sum_{j=1}^m p_j S_{\mathcal{D}_j}^* \boldsymbol{y}_{\mathcal{D}_j}$ and $A^{-1} - B^{-1} = A^{-1}(B - A)B^{-1}$, we have

$$\bar{\boldsymbol{w}}_{\mathcal{D},\lambda}^0 - \boldsymbol{w}_{\mathcal{D},\lambda}$$

$$= \sum_{j=1}^m p_j (\boldsymbol{\Phi}_{\mathcal{D}_j}^\top \boldsymbol{\Phi}_{\mathcal{D}_j} + \lambda I)^{-1} \boldsymbol{\Phi}_{\mathcal{D}_j}^\top \boldsymbol{y}_{\mathcal{D}_j} - (\boldsymbol{\Phi}_{\mathcal{D}}^\top \boldsymbol{\Phi}_{\mathcal{D}} + \lambda I)^{-1} \boldsymbol{\Phi}_{\mathcal{D}}^\top \boldsymbol{y}_{\mathcal{D}}$$

$$= \sum_{j=1}^m p_j (C_{\mathcal{D}_j} + \lambda I)^{-1} S_{\mathcal{D}_j}^* \boldsymbol{y}_{\mathcal{D}_j} - (C_{\mathcal{D}} + \lambda I)^{-1} S_{\mathcal{D}}^* \boldsymbol{y}_{\mathcal{D}}$$

$$= \sum_{j=1}^m p_j [(C_{\mathcal{D}_j} + \lambda I)^{-1} - (C_{\mathcal{D}} + \lambda I)^{-1}] S_{\mathcal{D}_j}^* \boldsymbol{y}_{\mathcal{D}_j}$$

$$= \sum_{j=1}^m p_j (C_{\mathcal{D}} + \lambda I)^{-1} (C_{\mathcal{D}} - C_{\mathcal{D}_j}) \boldsymbol{w}_{\mathcal{D}_j,\lambda}$$

$$= \sum_{j=1}^m p_j (C_{\mathcal{D}} + \lambda I)^{-1} (C_{\mathcal{D}} - C) \boldsymbol{w}_{\mathcal{D}_j,\lambda} + \sum_{j=1}^m p_j (C_{\mathcal{D}} + \lambda I)^{-1} (C - C_{\mathcal{D}_j}) \boldsymbol{w}_{\mathcal{D}_j,\lambda}$$

$$= \sum_{j=1}^m p_j (C_{\mathcal{D}} + \lambda I)^{-1} (C_{\mathcal{D}} - C)(\boldsymbol{w}_{\mathcal{D}_j,\lambda} - \boldsymbol{w}_\lambda) + \sum_{j=1}^m p_j (C_{\mathcal{D}} + \lambda I)^{-1} (C_{\mathcal{D}} - C) \boldsymbol{w}_\lambda$$

$$+ \sum_{j=1}^m p_j (C_{\mathcal{D}} + \lambda I)^{-1} (C - C_{\mathcal{D}_j}) \boldsymbol{w}_{\mathcal{D}_j,\lambda}$$

$$= \sum_{j=1}^m p_j (C_{\mathcal{D}} + \lambda I)^{-1} (C_{\mathcal{D}} - C)(\boldsymbol{w}_{\mathcal{D}_j,\lambda} - \boldsymbol{w}_\lambda) + \sum_{j=1}^m p_j (C_{\mathcal{D}} + \lambda I)^{-1} (C - C_{\mathcal{D}_j})(\boldsymbol{w}_{\mathcal{D}_j,\lambda} - \boldsymbol{w}_\lambda).$$

$$(23)$$

The last step is due to the fact $\sum_{j=1}^m p_j C_{\mathcal{D}} = \sum_{j=1}^m p_j C_{\mathcal{D}_j}$.

Combining equation 19 and equation 23, we have

$$\|\bar{f}_{\mathcal{D},\lambda}^0 - f_{\mathcal{D},\lambda}\|_2 \leq \|(C + \lambda I)^{1/2}(\bar{\boldsymbol{w}}_{\mathcal{D},\lambda}^0 - \boldsymbol{w}_{\mathcal{D},\lambda})\|_K$$

$$\leq \left\| \sum_{j=1}^m p_j (C + \lambda I)^{1/2}(C_{\mathcal{D}} + \lambda I)^{-1}(C_{\mathcal{D}} - C + C - C_{\mathcal{D}_j})(\boldsymbol{w}_{\mathcal{D}_j,\lambda} - \boldsymbol{w}_\lambda) \right\|.$$

$$(24)$$

Note that

$$(C + \lambda I)^{1/2}(C_{\mathcal{D}} + \lambda I)^{-1}(C_{\mathcal{D}} - C)$$

$$= (C + \lambda I)^{1/2}(C_{\mathcal{D}} + \lambda I)^{-1/2}(C_{\mathcal{D}} + \lambda I)^{-1/2}(C + \lambda I)^{1/2}(C + \lambda I)^{-1/2}(C_{\mathcal{D}} - C)(C + \lambda I)^{-1/2}(C + \lambda I)^{1/2}.$$

Using the inequality $\|(C + \lambda I)^{-1/2}(C_{\mathcal{D}} - C)(C + \lambda I)^{-1/2}\| \leq \mathcal{R}_{\mathcal{D},\lambda}$ from Lemma 1, we have

$$\|(C + \lambda I)^{1/2}(C_{\mathcal{D}} + \lambda I)^{-1}(C_{\mathcal{D}} - C)(\boldsymbol{w}_{\mathcal{D}_j,\lambda} - \boldsymbol{w}_\lambda)\|$$

$$\leq \mathcal{P}_{\mathcal{D},\lambda} \|(C + \lambda I)^{-1/2}(C_{\mathcal{D}} - C)(C + \lambda I)^{-1/2}(C + \lambda I)^{1/2}(\boldsymbol{w}_{\mathcal{D}_j,\lambda} - \boldsymbol{w}_\lambda)\| \quad (25)$$

$$\leq \mathcal{P}_{\mathcal{D},\lambda} \mathcal{R}_{\mathcal{D},\lambda} \|(C + \lambda I)^{1/2}(\boldsymbol{w}_{\mathcal{D}_j,\lambda} - \boldsymbol{w}_\lambda)\|.$$

Similarly, we have

$$(C + \lambda I)^{1/2}(C_\mathcal{D} + \lambda I)^{-1}(C - C_{\mathcal{D}_j})$$

$$=(C + \lambda I)^{1/2}(C_\mathcal{D} + \lambda I)^{-1}(C - C_j + C_j - C_{\mathcal{D}_j})$$

$$=(C + \lambda I)^{1/2}(C_\mathcal{D} + \lambda I)^{-1}(C + \lambda I)^{1/2}(C + \lambda I)^{-1/2}(C - C_j)(C + \lambda I)^{-1/2}(C + \lambda I)^{1/2}$$

$$\quad + (C + \lambda I)^{1/2}(C_\mathcal{D} + \lambda I)^{-1}(C + \lambda I)^{1/2}(C + \lambda I)^{-1/2}(C_j + \lambda I)^{1/2}$$

$$(C_j + \lambda I)^{-1/2}(C_j - C_{\mathcal{D}_j})(C_j + \lambda I)^{-1/2}(C_j + \lambda I)^{1/2}(C + \lambda I)^{-1/2}(C + \lambda I)^{1/2}.$$

Using $\|(C + \lambda I)^{-1/2}(C_j + \lambda I)^{1/2}\| \le \|I + (C + \lambda I)^{-1}(C_j - C)\|^{1/2} \le 1 + \frac{\Delta_{\mathcal{D}_j}}{\lambda}$, it holds

$$\|(C + \lambda I)^{1/2}(C_\mathcal{D} + \lambda I)^{-1}(C - C_{\mathcal{D}_j})(\boldsymbol{w}_{\mathcal{D}_j,\lambda} - \boldsymbol{w}_\lambda)\|$$

$$\le \frac{\mathcal{P}_{\mathcal{D},\lambda}\Delta_{\mathcal{D}_j}}{\lambda}\|(C + \lambda I)^{1/2}(\boldsymbol{w}_{\mathcal{D}_j,\lambda} - \boldsymbol{w}_\lambda)\| + \mathcal{P}_{\mathcal{D},\lambda}\mathcal{R}_{\mathcal{D}_j,\lambda}\left(1 + \frac{\Delta_{\mathcal{D}_j}}{\lambda}\right)\|(C + \lambda I)^{1/2}(\boldsymbol{w}_{\mathcal{D}_j,\lambda} - \boldsymbol{w}_\lambda)\|$$

$$\le \mathcal{P}_{\mathcal{D},\lambda}\left(\mathcal{R}_{\mathcal{D}_j,\lambda} + \frac{(1 + \mathcal{R}_{\mathcal{D}_j,\lambda})\Delta_{\mathcal{D}_j}}{\lambda}\right)\|(C + \lambda I)^{1/2}(\boldsymbol{w}_{\mathcal{D}_j,\lambda} - \boldsymbol{w}_\lambda)\|.$$

$$(26)$$

Therefore, substituting equation 25 and equation 26 to equation 24, we have

$$\|\bar{f}_{\mathcal{D},\lambda}^0 - f_{\mathcal{D},\lambda}\| \le \sum_{j=1}^m p_j \mathcal{P}_{\mathcal{D},\lambda}\left(\mathcal{R}_{\mathcal{D},\lambda} + \mathcal{R}_{\mathcal{D}_j,\lambda} + \frac{(1 + \mathcal{R}_{\mathcal{D}_j,\lambda})\Delta_{\mathcal{D}_j}}{\lambda}\right)\|(C + \lambda I)^{1/2}(\boldsymbol{w}_{\mathcal{D}_j,\lambda} - \boldsymbol{w}_\lambda)\|$$

$$\le \mathcal{P}_{\mathcal{D},\lambda}\sum_{j=1}^m p_j\left(2\mathcal{R}_{\mathcal{D}_j,\lambda} + \frac{(1 + \mathcal{R}_{\mathcal{D}_j,\lambda})\Delta_{\mathcal{D}_j}}{\lambda}\right)\|(C + \lambda I)^{1/2}(\boldsymbol{w}_{\mathcal{D}_j,\lambda} - \boldsymbol{w}_\lambda)\|.$$

$\square$

## C.8 ESTIMATING ERROR TERMS

### C.8.1 ESTIMATING FEDERATED ERROR

From Lemma 1, Lemma 4, and equation 18, there are two error terms $\|\boldsymbol{w}_{\mathcal{D}_j,\lambda} - \boldsymbol{w}_\lambda\|_K$ and $\|f_{\mathcal{D}_j,\lambda} - f_\lambda\|_2$ in federated error to be bounded. Using Bennett's inequality (Proposition 3), we first provide two useful lemmas.

**Lemma 3.** *Assume there exists $\kappa \ge 1$ such that $\|\phi(\boldsymbol{x})\|_K \le \kappa$, $\forall \boldsymbol{x} \in \mathcal{X}$ and $|y| \le B$. For $\delta \in (0, 1]$, the following holds with the probability at least $1 - \delta$*

$$\|(C + \lambda I)^{-1/2}(S_\mathcal{D}^* \boldsymbol{y}_\mathcal{D} - S^* f^*)\| \le 2B\kappa \mathcal{A}_{\mathcal{D},\lambda} \log \frac{2}{\delta},$$

$$\|(C_j + \lambda I)^{-1/2}(S_{\mathcal{D}_j}^* \boldsymbol{y}_{\mathcal{D}_j} - S_j^* f_j^*)\| \le 2B\kappa \mathcal{A}_{\mathcal{D}_j,\lambda} \log \frac{2}{\delta}.$$

*where $C_j$, $S_j^*$ are operators defined on the local distribution $\rho_j$, and*

$$\mathcal{A}_{\mathcal{D},\lambda} := \frac{1}{|\mathcal{D}|\sqrt{\lambda}} + \sqrt{\frac{\mathcal{N}(\lambda)}{|\mathcal{D}|}}, \quad \mathcal{A}_{\mathcal{D}_j,\lambda} := \frac{1}{|\mathcal{D}_j|\sqrt{\lambda}} + \sqrt{\frac{\mathcal{N}(\lambda)}{|\mathcal{D}_j|}}. \quad (27)$$

*Proof.* Let $\xi_i = (C + \lambda I)^{-1/2}\phi(\boldsymbol{x}_i)y_i$ in the Hilbert space $\mathcal{H}_K$. We see that

$$\frac{1}{|\mathcal{D}|}\sum_{i=1}^{|\mathcal{D}|}\xi_i = \frac{1}{n}\sum_{i=1}^n (C + \lambda I)^{-1/2}\phi(\boldsymbol{x}_i)y_i = (C + \lambda I)^{-1/2}S_\mathcal{D}\boldsymbol{y}_\mathcal{D},$$

$$\mathbb{E}\xi = \int_X (C + \lambda I)^{-1/2}\phi(\boldsymbol{x})f^*(\boldsymbol{x})d\rho_X(\boldsymbol{x}) = (C + \lambda I)^{-1/2}S^* f^*$$

Thus, the error term to bound can be stated as

$$\|(C + \lambda I)^{-1/2}(\widehat{S}_n^* \boldsymbol{y}_{\mathcal{D}} - S^* f^*)\| = \left\| \frac{1}{|\mathcal{D}|} \sum_{i=1}^{|\mathcal{D}|} \xi_i - \mathbb{E}\xi_i \right\|. \tag{28}$$

The rhs of the above identity can be bounded by Bennett's inequality (Proposition 3), thus we need to estimate $\|\xi_i - \mathbb{E}(\xi_i)\|$ and $\mathbb{E}\|\xi_i - \mathbb{E}(\xi_i)\|^2$ first.

We first recall the definition of effective dimension

$$\mathcal{N}(\lambda) = \mathbb{E}\langle \phi(\boldsymbol{x}), (C + \lambda I)^{-1}\phi(\boldsymbol{x})\rangle_K = \int_X \|(C + \lambda I)^{-1}\phi(\boldsymbol{x})\|_K^2 \, d\rho_X(\boldsymbol{x}).$$

By Jensen's inequality, we thus have

$$\|\xi_i - \mathbb{E}(\xi_i)\| \leq \|(C + \lambda I)^{-1/2}\phi(\boldsymbol{x}_i)\||y_i| + \mathbb{E}\|(C + \lambda I)^{-1/2}\phi(\boldsymbol{x}_i)\||y_i| \leq 2B\kappa\lambda^{-1/2}. \tag{29}$$

Note that

$$\mathbb{E}\|\xi_i - \mathbb{E}(\xi_i)\|^2 \leq 2\int_X \|(C + \lambda I)^{-1/2}\phi(\boldsymbol{x}_i)\|^2|y_i|^2 d\rho_X(\boldsymbol{x})$$
$$\leq 2B^2 \int_X \|(C + \lambda I)^{-1/2}\phi(\boldsymbol{x}_i)\|^2 d\rho_X(\boldsymbol{x}) \leq 2B^2\mathcal{N}(\lambda). \tag{30}$$

Substituting equation 29 and equation 30 to equation 28, by Bennett's inequality (Proposition 3), we have

$$\|(C + \lambda I)^{-1/2}(S_{\mathcal{D}}^* \boldsymbol{y}_{\mathcal{D}} - S^* f^*)\| \leq \frac{2B\kappa \log(2/\delta)}{|\mathcal{D}|\sqrt{\lambda}} + 2\sqrt{\frac{B^2\mathcal{N}(\lambda)\log(2/\delta)}{|\mathcal{D}|}}.$$

Similarly, we derive the bound for $\|(C_j + \lambda I)^{-1/2}(S_{\mathcal{D}_j}^* \boldsymbol{y}_{\mathcal{D}_j} - S_{\mathcal{D}_j}^* f_j^*)\|$. Thus, we prove the result.

$\square$

**Lemma 4** (From Theoreom 4 of Caponnetto & De Vito (2007)). *Assume there exists $\kappa \geq 1$ such that $\|\phi(\boldsymbol{x})\|_K \leq \kappa, \ \forall \boldsymbol{x} \in \mathcal{X}$. For $\delta \in (0,1]$, the following holds with the probability at least $1 - \delta$*

$$\|(C + \lambda I)^{-1/2}(C - C_{\mathcal{D}})\| \leq 2\kappa(\kappa + 1)\mathcal{A}_{\mathcal{D},\lambda} \log \frac{2}{\delta},$$

$$\|(C_j + \lambda I)^{-1/2}(C_j - C_{\mathcal{D}_j})\| \leq 2\kappa(\kappa + 1)\mathcal{A}_{\mathcal{D}_j,\lambda} \log \frac{2}{\delta}.$$

The above lemma is a standard method for the difference between expected and empirical covariance operators $C - C_{\mathcal{D}}$ and $C_j - C_{\mathcal{D}_j}$. Using a concentration inequality in Hilbert spaces, it have been proven in Caponnetto & De Vito (2007); Smale & Zhou (2007); Guo et al. (2017).

We define the expected estimators for local machines and centralized model as

$$\boldsymbol{w}_{j,\lambda} = \underset{\boldsymbol{w} \in \mathcal{H}_K}{\arg\min} \left\{ \int_X (\langle \boldsymbol{w}, \phi(\boldsymbol{x})\rangle - f^*(\boldsymbol{x}))^2 d\rho_j(\boldsymbol{x}) + \lambda\|\boldsymbol{w}\|_K^2 \right\}$$

$$\boldsymbol{w}_{\lambda} = \underset{\boldsymbol{w} \in \mathcal{H}_K}{\arg\min} \left\{ \int_X (\langle \boldsymbol{w}, \phi(\boldsymbol{x})\rangle - f^*(\boldsymbol{x}))^2 d\rho_X(\boldsymbol{x}) + \lambda\|\boldsymbol{w}\|_K^2 \right\}.$$

**Proposition 5.** *Assume $\|\phi(\boldsymbol{x})\|_K \leq \kappa$ and $|y| \leq B$. Under Assumption 2, for $\delta \in (0, 1/2)$, the following bound hold with the probability at least $1 - 2\delta$*

$$\|(C + \lambda I)^{1/2}(\boldsymbol{w}_{\mathcal{D}_j,\lambda} - \boldsymbol{w}_{\lambda})\| \leq C_1\sqrt{1 + \frac{\Delta_{\mathcal{D}_j}}{\lambda}}\mathcal{P}_{\mathcal{D}_j,\lambda}\mathcal{A}_{\mathcal{D}_j,\lambda} \log \frac{2}{\delta} + \frac{\kappa^2 R\Delta_{\mathcal{D}_j}}{\lambda} + \Delta_{f_j}. \tag{31}$$

*where $C_1 = 2\kappa(B + 2\kappa^3 R)$.*

*Proof.* We introduce the intermediate estimators $\boldsymbol{w}_{j,\lambda} = (C_j + \lambda I)^{-1}S_j^* f_j^*$, where $S_j^*$ and $C_j$ are operators defined on the local distribution $\rho_j$. Then, it holds

$$\|(C + \lambda I)^{1/2}(\boldsymbol{w}_{\mathcal{D}_j,\lambda} - \boldsymbol{w}_\lambda)\| \leq \|(C + \lambda I)^{1/2}(\boldsymbol{w}_{\mathcal{D}_j,\lambda} - \boldsymbol{w}_{j,\lambda})\| + \|(C + \lambda I)^{1/2}(\boldsymbol{w}_{j,\lambda} - \boldsymbol{w}_\lambda)\|$$

(32)

where $\|\boldsymbol{w}_{\mathcal{D}_j,\lambda} - \boldsymbol{w}_{j,\lambda}\|$ is the local variance and $\Delta_{f_j}$ is the model heterogeneity.

$$(C + \lambda I)^{1/2}(\boldsymbol{w}_{j,\lambda} - \boldsymbol{w}_\lambda)$$
$$= (C + \lambda I)^{1/2}\left[(C_j + \lambda I)^{-1}S_j^* f_j^* - (C + \lambda I)^{-1}S^* f^*\right]$$
$$= (C + \lambda I)^{1/2}\left[(C_j + \lambda I)^{-1}S_j^* f_j^* - (C + \lambda I)^{-1}S^* f_j^* + (C + \lambda I)^{-1}S^* f_j^* - (C + \lambda I)^{-1}S^* f^*\right]$$
$$= (C + \lambda I)^{-1/2}S^*(L - L_j)(L_j + \lambda I)^{-1}f_j^* + (C + \lambda I)^{-1/2}S^*(f_j^* - f^*)$$
$$= (C + \lambda I)^{-1/2}S^*(L - L_j)(L_j + \lambda I)^{-1}L^r L^{-r}f_j^* + (C + \lambda I)^{-1/2}S^*(f_j^* - f^*).$$

Since $\|(C + \lambda I)^{-1/2}S^*\| \leq 1$, $\|L\| \leq \kappa^2$, $\|C - C_j\| = \|L - L_j\|$ and $\Delta_{f_j} = \|f_j^* - f^*\|$, from Assumption 2, we have

$$\|(C + \lambda I)^{1/2}(\boldsymbol{w}_{j,\lambda} - \boldsymbol{w}_\lambda)\| \leq \frac{\kappa^{2r}R\Delta_{\mathcal{D}_j}}{\lambda} + \Delta_{f_j}.$$

(33)

We then decompose the local variance

$$\boldsymbol{w}_{\mathcal{D}_j,\lambda} - \boldsymbol{w}_{j,\lambda}$$
$$= (C_{\mathcal{D}_j} + \lambda I)^{-1}S_{\mathcal{D}_j}^* \boldsymbol{y}_{\mathcal{D}_j} - (C_{\mathcal{D}_j} + \lambda I)^{-1}S_j^* f_j^* + (C_{\mathcal{D}_j} + \lambda I)^{-1}S_j^* f_j^* - (C_j + \lambda I)^{-1}S_j^* f_j^*$$
$$= (C_{\mathcal{D}_j} + \lambda I)^{-1}(C_j + \lambda I)^{1/2}(C_j + \lambda I)^{-1/2}(S_{\mathcal{D}_j}^* \boldsymbol{y}_{\mathcal{D}_j} - S_j^* f_j^*) + [(C_{\mathcal{D}_j} + \lambda I)^{-1} - (C_j + \lambda I)^{-1}]S_j^* f_j^*$$
$$= (C_{\mathcal{D}_j} + \lambda I)^{-1}(C_j + \lambda I)^{1/2}(C_j + \lambda I)^{-1/2}(S_{\mathcal{D}_j}^* \boldsymbol{y}_{\mathcal{D}_j} - S_j^* f_j^*) + (C_{\mathcal{D}_j} + \lambda I)^{-1}(C_j - C_{\mathcal{D}_j})\boldsymbol{w}_{j,\lambda}$$
$$= (C_{\mathcal{D}_j} + \lambda I)^{-1}(C_j + \lambda I)^{1/2}\left[(C_j + \lambda I)^{-1/2}(S_{\mathcal{D}_j}^* \boldsymbol{y}_{\mathcal{D}_j} - S_j^* f_j^*) + (C_j + \lambda I)^{-1/2}(C_j - C_{\mathcal{D}_j})\boldsymbol{w}_{j,\lambda}\right].$$

and it holds

$$(C + \lambda I)^{1/2}(\boldsymbol{w}_{\mathcal{D}_j,\lambda} - \boldsymbol{w}_{j,\lambda})$$
$$= (C + \lambda I)^{1/2}(C_{\mathcal{D}_j} + \lambda I)^{-1}(C_j + \lambda I)^{1/2}\Big[(C_j + \lambda I)^{-1/2}(S_{\mathcal{D}_j}^* \boldsymbol{y}_{\mathcal{D}_j} - S_j^* f_j^*)$$
$$+ (C_j + \lambda I)^{-1/2}(C_j - C_{\mathcal{D}_j})\boldsymbol{w}_{j,\lambda}\Big]$$

(34)

$$= (C + \lambda I)^{1/2}(C_j + \lambda I)^{-1/2}(C_j + \lambda I)^{1/2}(C_{\mathcal{D}_j} + \lambda I)^{-1/2}(C_{\mathcal{D}_j} + \lambda I)^{-1/2}(C_j + \lambda I)^{1/2}$$
$$\Big[(C_j + \lambda I)^{-1/2}(S_{\mathcal{D}_j}^* \boldsymbol{y}_{\mathcal{D}_j} - S_j^* f_j^*) + (C_j + \lambda I)^{-1/2}(C_j - C_{\mathcal{D}_j})\boldsymbol{w}_{j,\lambda}\Big].$$

Due to Assumption 2 and $\|L_j\| \leq \kappa^2$, we obtain

$$\|\boldsymbol{w}_{j,\lambda}\|_K = \|(L_j + \lambda I)^{-1}L_j f_j^*\| = \|(L_j + \lambda I)^{-1}L_j L_j^r L_j^{-r}f_j^*\| \leq \kappa^{2r}\|L^{-r}f_j^*\| \leq \kappa^{2r}R. \quad (35)$$

Thus, substituting equation 35 to equation 34, using Lemma 3 and Lemma 4, for any $\delta \in (0, 1/2)$, we have with the probability $1 - 2\delta$

$$\|(C + \lambda I)^{1/2}(\boldsymbol{w}_{\mathcal{D}_j,\lambda} - \boldsymbol{w}_{j,\lambda})\|$$
$$\leq \mathcal{P}_{\mathcal{D}_j,\lambda}\sqrt{1 + \frac{\Delta_{\mathcal{D}_j}}{\lambda}}\left(2B\kappa\mathcal{A}_{\mathcal{D}_j,\lambda}\log\frac{2}{\delta} + 2\kappa(\kappa + 1)\mathcal{A}_{\mathcal{D}_j,\lambda}\log\frac{2}{\delta}\kappa^{2r}R\right)$$

(36)

$$\leq 2\kappa(B + 2\kappa^3 R)\sqrt{1 + \frac{\Delta_{\mathcal{D}_j}}{\lambda}}\mathcal{P}_{\mathcal{D}_j,\lambda}\mathcal{A}_{\mathcal{D}_j,\lambda}\log\frac{2}{\delta}.$$

Applying equation 33 and equation 36 to equation 32, we prove the result.

$\square$

**Theorem 5** (Detailed version of Theorem 2). *For any $\delta \in (0, 1)$, under Assumption 2, with the probability at least $1 - \delta$, the federated error holds*

$$\|\bar{f}_{\mathcal{D},\lambda}^t - f_{\mathcal{D},\lambda}\|_2 \leq C_2 \Upsilon^t \sum_{j=1}^{m} p_j \sqrt{1 + \frac{\Delta_{\mathcal{D}_j}}{\lambda}} \left(2\mathcal{R}_{\mathcal{D}_j,\lambda} + \frac{(1 + \mathcal{R}_{\mathcal{D}_j,\lambda})\Delta_{\mathcal{D}_j}}{\lambda}\right) \left(\mathcal{A}_{\mathcal{D}_j,\lambda} \log \frac{2}{\delta} + \frac{\Delta_{\mathcal{D}_j}}{\lambda} + \Delta_{f_j}\right).$$
(37)

*where $C_2 = 2\kappa(B + 2\kappa^3 R)/(1 - \beta)$, $\beta = \lambda_{max}((C + \lambda)^{-1/2}(C - C_{\mathcal{D}})(C + \lambda)^{-1/2})$ and $\mathcal{A}_{\mathcal{D}_j,\lambda} = \frac{1}{\sqrt{\lambda}|\mathcal{D}_j|} + \sqrt{\frac{\mathcal{N}(\lambda)}{|\mathcal{D}_j|}}$.*

*Proof.* Substituting equation 31 and equation 22 to Theorem 1, with the probability $1 - 2\delta$, we obtain the federated error

$$\|\bar{f}_{\mathcal{D},\lambda}^t - f_{\mathcal{D},\lambda}\|_2$$

$$\leq \Upsilon^t \left\|(C + \lambda I)^{1/2}(\bar{w}_{\mathcal{D},\lambda}^0 - w_{\mathcal{D},\lambda})\right\|_K$$

$$\leq \Upsilon^t \sum_{j=1}^{m} p_j \mathcal{P}_{\mathcal{D},\lambda} \left(2\mathcal{R}_{\mathcal{D}_j,\lambda} + \frac{(1 + \mathcal{R}_{\mathcal{D}_j,\lambda})\Delta_{\mathcal{D}_j}}{\lambda}\right) \left\|(C + \lambda I)^{1/2}(w_{\mathcal{D}_j,\lambda} - w_\lambda)\right\|_K$$

$$\leq \Upsilon^t \sum_{j=1}^{m} p_j \mathcal{P}_{\mathcal{D},\lambda} \left(2\mathcal{R}_{\mathcal{D}_j,\lambda} + \frac{(1 + \mathcal{R}_{\mathcal{D}_j,\lambda})\Delta_{\mathcal{D}_j}}{\lambda}\right) \left(C_1 \sqrt{1 + \frac{\Delta_{\mathcal{D}_j}}{\lambda}} \mathcal{P}_{\mathcal{D}_j,\lambda} \mathcal{A}_{\mathcal{D}_j,\lambda} \log \frac{2}{\delta} + \frac{\kappa^2 R \Delta_{\mathcal{D}_j}}{\lambda} + \Delta_{f_j}\right)$$

$$\leq \Upsilon^t \sum_{j=1}^{m} \frac{C_1 p_j}{1 - \beta} \sqrt{1 + \frac{\Delta_{\mathcal{D}_j}}{\lambda}} \left(2\mathcal{R}_{\mathcal{D}_j,\lambda} + \frac{(1 + \mathcal{R}_{\mathcal{D}_j,\lambda})\Delta_{\mathcal{D}_j}}{\lambda}\right) \left(\mathcal{A}_{\mathcal{D}_j,\lambda} \log \frac{2}{\delta} + \frac{\Delta_{\mathcal{D}_j}}{\lambda} + \Delta_{f_j}\right).$$
(38)

The last step is due to Lemma 2. $\qquad\square$

### C.8.2 ESTIMATING CENTRALIZED EXCESS RISK

The generalization analysis for the centralized model (the exact KRR) is standard Caponnetto & De Vito (2007); Smale & Zhou (2007), but the existing work imposed a strict assumption $r \in [1/2, 1]$ on the kernel space, which assumes the ideal estimator belongs to the kernel space $f^* \in \mathcal{H}_K$. Here, we relax this strict assumption to $r > 0$ but still obtain the identical optimal learning rates for the centralized excess risk bounds.

**Proposition 6.** *Under Assumption 2, for $\delta \in (0, 1/2)$, the following bounds hold with the probability at least $1 - 2\delta$*

$$\|f_{\mathcal{D},\lambda} - f^*\|_2 \leq C_1 \mathcal{P}_{\mathcal{D},\lambda}^{1/2} \mathcal{A}_{\mathcal{D},\lambda} \log \frac{2}{\delta} + R\lambda^r,$$
(39)

*where $C_1 = 2\kappa\left(B + 2\kappa^3 R\right)$.*

*Proof.* The excess risk term can be divided into two parts: variance and bias.

$$\|f_{\mathcal{D},\lambda} - f^*\| \leq \|f_{\mathcal{D},\lambda} - f_\lambda\| + \|f_\lambda - f^*\|.$$
(40)

Using Cauchy's inequality, Lemma 3 and Lemma 4, for $\delta \in (0, 1/2)$, with the probability at least $1 - 2\delta$ we have

$$\|f_{\mathcal{D},\lambda} - f_\lambda\|_2$$

$$=\|S(C_\mathcal{D} + \lambda I)^{-1} S_\mathcal{D}^* \boldsymbol{y}_\mathcal{D} - S(C_\mathcal{D} + \lambda I)^{-1} S^* f^* + S(C_\mathcal{D} + \lambda I)^{-1} S^* f^* - S(C + \lambda I)^{-1} S^* f^*\|_2$$

$$=\|S(C_\mathcal{D} + \lambda I)^{-1}(C + \lambda I)(C + \lambda I)^{-1/2}(C + \lambda I)^{-1/2}(S_\mathcal{D}^* \boldsymbol{y}_\mathcal{D} - S^* f^*)$$
$$+ S(C_\mathcal{D} + \lambda I)^{-1}(C + \lambda I)(C + \lambda I)^{-1/2}(C + \lambda I)^{-1/2}(C - C_\mathcal{D})(C + \lambda I)^{-1} S^* f^*\|_2$$

$$=\|S(C_\mathcal{D} + \lambda I)^{-1/2}(C_\mathcal{D} + \lambda I)^{-1/2}(C + \lambda I)^{1/2}(C + \lambda I)^{-1/2}(S_\mathcal{D}^* \boldsymbol{y}_\mathcal{D} - S^* f^*)$$
$$+ S(C_\mathcal{D} + \lambda I)^{-1/2}(C_\mathcal{D} + \lambda I)^{-1/2}(C + \lambda I)^{1/2}(C + \lambda I)^{-1/2}(C - C_\mathcal{D})(C + \lambda I)^{-1} S^* f^*\|_2$$

$$\leq 2B\kappa \log \frac{2}{\delta} \mathcal{P}_{\mathcal{D},\lambda}^{1/2} \mathcal{A}_{\mathcal{D},\lambda} + 2\kappa(\kappa + 1) \log \frac{2}{\delta} \mathcal{P}_{\mathcal{D},\lambda}^{1/2} \mathcal{A}_{\mathcal{D},\lambda} \|\boldsymbol{w}_\lambda\|_K$$

$$\leq 2\kappa \left(B + 2\kappa^3 R\right) \log \frac{2}{\delta} \mathcal{P}_{\mathcal{D},\lambda}^{1/2} \mathcal{A}_{\mathcal{D},\lambda}.$$

$$(41)$$

The last step is due $\|\boldsymbol{w}_\lambda\|_K = \|(L + \lambda I)^{-1} L f^*\| = \|(L + \lambda I)^{-1} L L^r L^{-r} f^*\| \leq \kappa^{2r} R$ due to Assumption 2.

The identity $A(A + \lambda I)^{-1} = I - \lambda(A + \lambda I)^{-1}$ holds for $\lambda > 0$ and $A$ the bounded self-adjoint positive operator. Then, under Assumption 2, it holds

$$\|f_\lambda - f^*\|_2$$
$$=\|(L + \lambda I)^{-1} L f^* - f^*\| = \|((L + \lambda I)^{-1} L - I)f^*\| = \|\lambda(L + \lambda I)^{-1} f^*\|$$
$$=\|\lambda^r \lambda^{1-r}(L + \lambda I)^{-(1-r)}(L + \lambda I)^{-r} L^r L^{-r} f^*\|$$
$$\leq \lambda^r \|\lambda^{1-r}(L + \lambda I)^{-(1-r)}\| \|(L + \lambda I)^{-r} L^r\| \|L^{-r} f^*\|$$
$$\leq R\lambda^r.$$

$$(42)$$

Substituting equation 41 and equation 42 to equation 40, we prove the result. $\qquad\square$

### C.9 Excess Risk Bounds for FedNewton

*Proof of Theorem 3.* In the homogeneous setting, we have $\Delta_{\mathcal{D}_j} = 0$ and $\Delta_{f_j} = 0$. Thus, under Assumption 2, from equation 37 and equation 39, it holds

$$\|\bar{f}_{\mathcal{D},\lambda}^t - f^*\|_2 \leq \|\bar{f}_{\mathcal{D},\lambda}^t - f_{\mathcal{D},\lambda}\|_2 + \|f_{\mathcal{D},\lambda} - f_{\mathcal{D},\lambda}\|_2$$

$$\leq \boldsymbol{O}\left(\Upsilon^t \sum_{j=1}^m p_j \mathcal{R}_{\mathcal{D}_j,\lambda} \mathcal{A}_{\mathcal{D}_j,\lambda} \log \frac{2}{\delta} + \mathcal{A}_{\mathcal{D},\lambda} \log \frac{2}{\delta} + R\lambda^r\right). \qquad (43)$$

If $|\mathcal{D}_j| > 29(\kappa^2 + 1)\log(1/\delta)/\lambda$, we have $\Upsilon < 1$. Otherwise, $\Upsilon \geq 1$.

From equation 15 and equation 27, under Assumption 1, with the probability at least $1 - 3\delta$, we have

$$\mathcal{R}_{\mathcal{D}_j,\lambda} \mathcal{A}_{\mathcal{D}_j,\lambda}$$

$$=\boldsymbol{O}\left(\left(\frac{1}{\lambda|\mathcal{D}_j|} + \sqrt{\frac{1}{\lambda|\mathcal{D}_j|}}\right) \log \frac{2}{\delta} \times \left(\frac{1}{|\mathcal{D}_j|\sqrt{\lambda}} + \sqrt{\frac{\mathcal{N}(\lambda)}{|\mathcal{D}_j|}}\right)\right)$$

$$=\boldsymbol{O}\left((|\mathcal{D}_j|^{-2}\lambda^{-1.5} + |\mathcal{D}_j|^{-1.5}\lambda^{-1-0.5\gamma} + |\mathcal{D}_j|^{-1.5}\lambda^{-1} + |\mathcal{D}_j|^{-1}\lambda^{-0.5-0.5\gamma}) \log \frac{2}{\delta}\right)$$

$$=\boldsymbol{O}\left((|\mathcal{D}_j|^{-2}\lambda^{-1.5} + |\mathcal{D}_j|^{-1.5}\lambda^{-1-0.5\gamma} + |\mathcal{D}_j|^{-1}\lambda^{-0.5-0.5\gamma}) \log \frac{2}{\delta}\right).$$

The relationships between $\lambda$ and $|\mathcal{D}_j|$ affects the value of $\mathcal{R}_{\mathcal{D}_j,\lambda} \mathcal{A}_{\mathcal{D}_j,\lambda}$.

$$\mathcal{R}_{\mathcal{D}_j,\lambda} \mathcal{A}_{\mathcal{D}_j,\lambda} = \log \frac{2}{\delta} \begin{cases} \boldsymbol{O}(|\mathcal{D}_j|^{-2}\lambda^{-1.5}), & \text{if } \lambda < \boldsymbol{O}(|\mathcal{D}_j|^{\frac{1}{\gamma-1}}). \\ \boldsymbol{O}(|\mathcal{D}_j|^{-1.5}\lambda^{-1-0.5\gamma}), & \text{if } \Omega(|\mathcal{D}_j|^{\frac{1}{\gamma-1}}) \leq \lambda < \boldsymbol{O}(|\mathcal{D}_j|^{-1}). \\ \boldsymbol{O}(|\mathcal{D}_j|^{-1}\lambda^{-0.5-0.5\gamma}), & \text{if } \lambda \geq \Omega(|\mathcal{D}_j|^{-1}). \end{cases}$$

By setting $\lambda = |\mathcal{D}|^{\frac{-1}{2r+\gamma}}$ and $2r + \gamma \geq 1$, we have

$$
\mathcal{R}_{\mathcal{D}_j,\lambda}\mathcal{A}_{\mathcal{D}_j,\lambda} = \log\frac{2}{\delta}
\begin{cases}
\boldsymbol{O}\left(|\mathcal{D}_j|^{-2}|\mathcal{D}|^{\frac{1.5}{2r+\gamma}}\right), & \text{if } |\mathcal{D}_j| \lesssim |\mathcal{D}|^{\frac{1-\gamma}{2r+\gamma}}. \\[2mm]
\boldsymbol{O}\left(|\mathcal{D}_j|^{-1.5}|\mathcal{D}|^{\frac{1+0.5\gamma}{2r+\gamma}}\right), & \text{if } |\mathcal{D}|^{\frac{1-\gamma}{2r+\gamma}} \lesssim |\mathcal{D}_j| \lesssim |\mathcal{D}|^{\frac{1}{2r+\gamma}}. \\[2mm]
\boldsymbol{O}\left(|\mathcal{D}_j|^{-1}|\mathcal{D}|^{\frac{1+\gamma}{4r+2\gamma}}\right), & \text{if } |\mathcal{D}_j| \gtrsim |\mathcal{D}|^{\frac{1}{2r+\gamma}}.
\end{cases}
\tag{44}
$$

and

$$
\mathcal{A}_{\mathcal{D},\lambda} = |\mathcal{D}|^{\frac{1-4r-2\gamma}{4r+\gamma}} + |\mathcal{D}|^{\frac{-r}{2r+\gamma}} \leq 2|\mathcal{D}|^{\frac{-r}{2r+\gamma}}.
\tag{45}
$$

Substituting equation 44 and equation 45 to equation 43, we have

$$\|\bar{f}_{\mathcal{D},\lambda}^t - f^*\|_2$$

$$
\lesssim |\mathcal{D}|^{\frac{-r}{2r+\gamma}}\log\frac{2}{\delta} + \Upsilon^t \log^2\frac{2}{\delta}\sum_{j=1}^m p_j
\begin{cases}
|\mathcal{D}_j|^{-2}|\mathcal{D}|^{\frac{1.5}{2r+\gamma}}, & \text{if } |\mathcal{D}_j| \lesssim |\mathcal{D}|^{\frac{1-\gamma}{2r+\gamma}} \\[2mm]
|\mathcal{D}_j|^{-1.5}|\mathcal{D}|^{\frac{1+0.5\gamma}{2r+\gamma}}, & \text{if } |\mathcal{D}|^{\frac{1-\gamma}{2r+\gamma}} \lesssim |\mathcal{D}_j| \lesssim |\mathcal{D}|^{\frac{1}{2r+\gamma}} \\[2mm]
|\mathcal{D}_j|^{-1}|\mathcal{D}|^{\frac{1+\gamma}{4r+2\gamma}}, & \text{if } |\mathcal{D}|^{\frac{1}{2r+\gamma}} \lesssim |\mathcal{D}_j| \lesssim |\mathcal{D}|^{\frac{2r+\gamma+1}{4r+2\gamma}} \\[2mm]
|\mathcal{D}|^{\frac{-r}{2r+\gamma}}, & \text{if } |\mathcal{D}_j| \gtrsim |\mathcal{D}|^{\frac{2r+\gamma+1}{4r+2\gamma}}
\end{cases}
\tag{46}
$$

where

$$
\begin{cases}
t = 0, \Upsilon \geq 1, & \text{if } |\mathcal{D}_j| \lesssim |\mathcal{D}|^{\frac{1}{2r+\gamma}} \\[2mm]
t > 0, \Upsilon^t \lesssim \left(\frac{|\mathcal{D}|^{\frac{1}{2r+\gamma}}}{|\mathcal{D}_j|}\right)^{0.5t}, & \text{otherwise.}
\end{cases}
\tag{47}
$$

Note that, $\Upsilon = 2\sum_{j=1}^m p_j \mathcal{P}_{\mathcal{D}_j,\lambda}\mathcal{R}_{\mathcal{D}_j,\lambda} \lesssim \sum_{j=1}^m p_j \mathcal{R}_{\mathcal{D}_j,\lambda}$. When $|\mathcal{D}_j| \gtrsim |\mathcal{D}|^{\frac{2r+\gamma+1}{4r+2\gamma}}$, we thus have $\mathcal{R}_{\mathcal{D}_j,\lambda} \lesssim \sqrt{\frac{1}{\lambda|\mathcal{D}_j|}} \lesssim |\mathcal{D}|^{\frac{1-2r-\gamma}{8r+4\gamma}}$. $\qquad\square$

*Proof of Theorem 4.* Under Assumption 2, from equation 37 and equation 39, it holds

$$\|\bar{f}_{\mathcal{D},\lambda}^t - f^*\|_2 \leq \|\bar{f}_{\mathcal{D},\lambda}^t - f_{\mathcal{D},\lambda}\|_2 + \|f_{\mathcal{D},\lambda} - f_{\mathcal{D},\lambda}\|_2$$

$$\leq \boldsymbol{O}\left(\Upsilon^t \sum_{j=1}^m p_j\sqrt{1+\frac{\Delta_{\mathcal{D}_j}}{\lambda}}\left(2\mathcal{R}_{\mathcal{D}_j,\lambda} + \frac{(1+\mathcal{R}_{\mathcal{D}_j,\lambda})\Delta_{\mathcal{D}_j}}{\lambda}\right)\left(\mathcal{A}_{\mathcal{D}_j,\lambda}\log\frac{2}{\delta} + \frac{\Delta_{\mathcal{D}_j}}{\lambda} + \Delta_{f_j}\right) + \mathcal{A}_{\mathcal{D},\lambda}\log\frac{2}{\delta} + R\lambda^r\right).$$

Let $\lambda = |\mathcal{D}|^{\frac{-1}{2r+\gamma}}$ and $2r + \gamma \geq 1$. When $|\mathcal{D}_j| \leq \boldsymbol{O}(|\mathcal{D}|^{\frac{1}{2r+\gamma}})$, we have $\frac{1}{\lambda|\mathcal{D}_j|} \geq \sqrt{\frac{1}{\lambda|\mathcal{D}_j|}} \geq 1$ and $\mathcal{R}_{\mathcal{D}_j,\lambda} \lesssim \frac{1}{\lambda|\mathcal{D}_j|} + \sqrt{\frac{1}{\lambda|\mathcal{D}_j|}} \lesssim \frac{1}{\lambda|\mathcal{D}_j|}$ from equation 15. Thus,

$$\|\bar{f}_{\mathcal{D},\lambda}^t - f^*\|_2$$

$$\leq \boldsymbol{O}\left(\Upsilon^t \sum_{j=1}^m p_j\left(1+\frac{\Delta_{\mathcal{D}_j}}{\lambda}\right)^{1.5}\left(\mathcal{R}_{\mathcal{D}_j,\lambda}\mathcal{A}_{\mathcal{D}_j,\lambda}\log\frac{2}{\delta} + \frac{\mathcal{R}_{\mathcal{D}_j,\lambda}\Delta_{\mathcal{D}_j}}{\lambda} + \mathcal{R}_{\mathcal{D}_j,\lambda}\Delta_{f_j}\right) + \mathcal{A}_{\mathcal{D},\lambda}\log\frac{2}{\delta} + R\lambda^r\right)$$

$$\leq \boldsymbol{O}\left(\Upsilon^t \sum_{j=1}^m p_j\left(1+\frac{\Delta_{\mathcal{D}_j}}{\lambda}\right)^{1.5}\left(\mathcal{R}_{\mathcal{D}_j,\lambda}\mathcal{A}_{\mathcal{D}_j,\lambda}\log\frac{2}{\delta} + \frac{\Delta_{\mathcal{D}_j}}{\lambda^2|\mathcal{D}_j|} + \frac{\Delta_{f_j}}{\lambda|\mathcal{D}_j|}\right) + \mathcal{A}_{\mathcal{D},\lambda}\log\frac{2}{\delta} + R\lambda^r\right)$$

$$\leq \boldsymbol{O}\left(\Upsilon^t \sum_{j=1}^m p_j\left(1+\frac{\Delta_{\mathcal{D}_j}}{\lambda}\right)^{1.5}\left(\mathcal{R}_{\mathcal{D}_j,\lambda}\mathcal{A}_{\mathcal{D}_j,\lambda} + \frac{|\mathcal{D}|^{\frac{2}{2r+\gamma}}}{|\mathcal{D}_j|}\Delta_{\mathcal{D}_j} + \frac{|\mathcal{D}|^{\frac{1}{2r+\gamma}}}{|\mathcal{D}_j|}\Delta_{f_j}\right)\log\frac{2}{\delta} + |\mathcal{D}|^{\frac{-r}{2r+\gamma}}\right).$$

When $|\mathcal{D}_j| \geq \Omega(|\mathcal{D}|^{\frac{1}{2r+\gamma}})$, we have $\mathcal{R}_{\mathcal{D}_j,\lambda} \lesssim \sqrt{\frac{1}{\lambda|\mathcal{D}_j|}} \leq 1$, $\mathcal{A}_{\mathcal{D}_j,\lambda} \lesssim |\mathcal{D}_j|^{-1/2}|\mathcal{D}|^{\frac{\gamma/2}{2r+\gamma}}$ and

$$\|\bar{f}_{\mathcal{D},\lambda}^t - f^*\|_2$$

$$\leq O\left(\Upsilon^t \sum_{j=1}^m p_j \sqrt{1 + \frac{\Delta_{\mathcal{D}_j}}{\lambda}} \left(\mathcal{R}_{\mathcal{D}_j,\lambda} + \frac{\Delta_{\mathcal{D}_j}}{\lambda}\right)\left(\mathcal{A}_{\mathcal{D}_j,\lambda}\log\frac{2}{\delta} + \frac{\Delta_{\mathcal{D}_j}}{\lambda} + \Delta_{f_j}\right) + |\mathcal{D}|^{\frac{-r}{2r+\gamma}}\right)$$

$$\leq O\left(\Upsilon^t \sum_{j=1}^m p_j \sqrt{1 + \frac{\Delta_{\mathcal{D}_j}}{\lambda}} \left(\mathcal{R}_{\mathcal{D}_j,\lambda}\mathcal{A}_{\mathcal{D}_j,\lambda} + |\mathcal{D}|^{\frac{1}{2r+\gamma}}\Delta_{\mathcal{D}_j} + \Delta_{f_j} + \frac{|\mathcal{D}|^{\frac{\gamma+2}{4r+2\gamma}}}{\sqrt{|\mathcal{D}_j|}}\Delta_{\mathcal{D}_j} + |\mathcal{D}|^{\frac{2}{2r+\gamma}}\Delta_{\mathcal{D}_j}^2\right.\right.$$

$$\left.\left. + |\mathcal{D}|^{\frac{1}{2r+\gamma}}\Delta_{\mathcal{D}_j}\Delta_{f_j}\right)\log\frac{2}{\delta} + |\mathcal{D}|^{\frac{-r}{2r+\gamma}}\right)$$

$$\leq O\left(\Upsilon^t \sum_{j=1}^m p_j \sqrt{1 + \frac{\Delta_{\mathcal{D}_j}}{\lambda}} \left(\mathcal{R}_{\mathcal{D}_j,\lambda}\mathcal{A}_{\mathcal{D}_j,\lambda} + |\mathcal{D}|^{\frac{1}{2r+\gamma}}\Delta_{\mathcal{D}_j} + \Delta_{f_j} + |\mathcal{D}|^{\frac{2}{2r+\gamma}}\Delta_{\mathcal{D}_j}^2 + |\mathcal{D}|^{\frac{1}{2r+\gamma}}\Delta_{\mathcal{D}_j}\Delta_{f_j}\right)\log\frac{2}{\delta}\right.$$

$$\left. + |\mathcal{D}|^{\frac{-r}{2r+\gamma}}\right)$$

$$\leq O\left(\Upsilon^t \sum_{j=1}^m p_j \sqrt{1 + \frac{\Delta_{\mathcal{D}_j}}{\lambda}} \left(\mathcal{R}_{\mathcal{D}_j,\lambda}\mathcal{A}_{\mathcal{D}_j,\lambda} + (1 + |\mathcal{D}|^{\frac{1}{2r+\gamma}}\Delta_{\mathcal{D}_j})(|\mathcal{D}|^{\frac{1}{2r+\gamma}}\Delta_{\mathcal{D}_j} + \Delta_{f_j})\right)\log\frac{2}{\delta} + |\mathcal{D}|^{\frac{-r}{2r+\gamma}}\right).$$

Combing with equation 44, we complete the proof

$$\|\bar{f}_{\mathcal{D},\lambda}^t - f^*\|_2 \lesssim \Upsilon^t \sum_{j=1}^m p_j \sqrt{1 + \frac{\Delta_{\mathcal{D}_j}}{\lambda}}(\aleph_j + \Pi_j)\log^2\frac{2}{\delta} + |\mathcal{D}|^{\frac{-r}{2r+\gamma}}\log\frac{2}{\delta}.$$

Here, $\aleph_j$ and $\Pi_j$ have different values w.r.t local sample size

$$\aleph_j = \begin{cases} |\mathcal{D}_j|^{-2}|\mathcal{D}|^{\frac{1.5}{2r+\gamma}}, & \text{if } |\mathcal{D}_j| \lesssim |\mathcal{D}|^{\frac{1-\gamma}{2r+\gamma}} \\ |\mathcal{D}_j|^{-1.5}|\mathcal{D}|^{\frac{1+0.5\gamma}{2r+\gamma}}, & \text{if } |\mathcal{D}|^{\frac{1-\gamma}{2r+\gamma}} \lesssim |\mathcal{D}_j| \lesssim |\mathcal{D}|^{\frac{1}{2r+\gamma}} \\ |\mathcal{D}_j|^{-1}|\mathcal{D}|^{\frac{1+\gamma}{4r+2\gamma}}, & \text{if } |\mathcal{D}|^{\frac{1}{2r+\gamma}} \lesssim |\mathcal{D}_j| \lesssim |\mathcal{D}|^{\frac{2r+\gamma+1}{4r+2\gamma}} \\ |\mathcal{D}|^{\frac{-r}{2r+\gamma}}, & \text{if } |\mathcal{D}_j| \gtrsim |\mathcal{D}|^{\frac{2r+\gamma+1}{4r+2\gamma}}, \end{cases}$$

and

$$\Pi_j = \begin{cases} \frac{|\mathcal{D}|^{\frac{2}{2r+\gamma}}}{|\mathcal{D}_j|}\Delta_{\mathcal{D}_j} + \frac{|\mathcal{D}|^{\frac{1}{2r+\gamma}}}{|\mathcal{D}_j|}\Delta_{f_j}, & \text{if } |\mathcal{D}_j| \lesssim |\mathcal{D}|^{\frac{1}{2r+\gamma}} \\ (1 + |\mathcal{D}|^{\frac{1}{2r+\gamma}}\Delta_{\mathcal{D}_j})(\Delta_{f_j} + |\mathcal{D}|^{\frac{1}{2r+\gamma}}\Delta_{\mathcal{D}_j}), & \text{if } |\mathcal{D}_j| \gtrsim |\mathcal{D}|^{\frac{1}{2r+\gamma}}. \end{cases}$$

$\square$