# OpenReview forum: "Statistical Optimality of Newton-type Federated Learning with Heterogeneous Data"
_ICLR.cc/2026/Conference — Submitted to ICLR 2026_

### Official Review · Reviewer_x34g · 2025-10-17

**Soundness:** 2
**Presentation:** 2
**Contribution:** 2
**Rating:** 4
**Confidence:** 2

**Summary:**

The authors propose FedNewton, which communicates the global gradient and local inverse‑Hessian preconditioned increments. The theory gives minimax‑optimal generalization rates while quantifying effects of local sample size, covariate shift, and response shift. A key message appears to be that if local data are not too small and heterogeneity is modest, the federated error decays exponentially in $t$.

**Strengths:**

1. Prior second‑order FL work focused on optimization, but here we get excess‑risk bounds with minimax‑optimal rates.

2. The claim that the proposed algorithm has an exponential decay in the federated error in terms of iterations is an astonishing and shocking result, if true.

**Weaknesses:**

1. The paper is quite involved, and would benefit from more intuitive scaffolding.

2. Can you clarify the statement that this second-order method drawing on local hessians is only "2 times" as compared to first-order FL algorithms?

**Questions:**

Please see weaknesses. Also, why is $C-C_{\mathcal{D}}$ PSD in equation (11)?

Note: I am not particularly knowledgeable about the area in which this paper is situated in. This is reflected in my confidence score.

---

> ### Author Response · Authors · 2025-12-03
> **Response to Comment 1: Limited intuitive scaffolding and paper readability**
>
> We thank the reviewer for carefully reading our paper and for the constructive feedback, especially given their self‑reported limited background in this specific area. We appreciate the positive assessment of our contribution to excess‑risk analysis for second‑order FL and the interest in our exponential‑decay result.
>
> Below we address the main points regarding (i) intuitive scaffolding of the theory, (ii) the “2 times” per‑round cost statement for second‑order information, and (iii) the positive semidefiniteness of the matrix in equation (11), and we also clarify how the revised experiments in Sections 3.1 and C support our claims.
>
> ---
>
> ## 1. Intuitive scaffolding and readability
>
> We agree that the current presentation is technically involved and can be hard to follow without substantial background in statistical learning theory and RKHS methods. Our initial goal was to provide a complete and rigorous treatment of minimax‑optimal generalization guarantees under heterogeneity, but this has led to heavy notation and dense proofs.
>
> In the revised manuscript, we will take several concrete steps to make the paper more accessible:
>
> ### High‑level overview before the main theory.
>
>    At the beginning of the main theory section, we will add a short, non‑technical overview explaining:
>
>    - what the *centralized excess‑risk* term represents;
>    - what the *federated error* term represents;
>    - how covariate shift and response (concept) shift appear in the bounds;
>    - why, under suitable conditions on local sample sizes and heterogeneity, the federated error can decay exponentially in the number of communication rounds.
>
> ### “Notation and intuition” summary.
>
>    We will include a compact table summarizing the most important symbols (e.g., kernel operator, effective dimension, heterogeneity measures, aggregated Hessian), each accompanied by a one‑sentence intuitive description, and we will reduce redundant notation.
>
> ### Pre‑theorem intuition paragraphs.
>
>    Before Theorems 1 and 2, we will add short paragraphs that informally describe:
>
>    - how the error decomposition works;
>    - why the federated error term exhibits linear (geometric/exponential) decay across rounds;
>    - how the assumptions on local sample size and heterogeneity enter the analysis.
>
> ### Illustrative figure for the error decomposition.
>
>    We will add a simple diagram showing the relationship between:
>
>    - the “ideal” centralized KRR solution;
>    - local models at each client;
>    - the global model produced by FedNewton, and how the gap between them shrinks geometrically under the stated conditions.
>
> ### Relocating technical details to the appendix.
>
>    Several of the more intricate operator‑theoretic lemmas and derivations will be moved to the appendix, so that the main text can focus on the conceptual structure and key rates.
>
> We hope these changes will make the main ideas and contributions accessible to a broader audience, including readers not deeply familiar with the underlying theory.

---

> ### Author Response · Authors · 2025-12-03
> **Response to Comment 2: Clarifying the “2×” cost claim and large‑scale empirical validation**
>
> We appreciate the request to clarify our statement that the second‑order method based on local Hessians is only “2×” as costly as first‑order FL algorithms. As originally phrased, this is misleading, and we will revise it.
>
> Our intention was to compare per‑round communication overhead, not total local computation. In the revision, we also update Section 3.1 and add Section C to empirically validate this tradeoff on large‑scale networks and datasets.
>
> ### 2.1 Communication vs. computation
>
> In standard first‑order FL (e.g., FedAvg/FedProx), each round typically consists of the server broadcasting a model vector in $\mathbb{R}^M$ and each client returning a gradient or model update in $\mathbb{R}^M$. In FedNewton, each round similarly involves broadcasting a vector in $\mathbb{R}^M$ (global model or gradient) and clients sending back a preconditioned update in $\mathbb{R}^M$. Thus, the dimension of uplink/downlink messages is $O(M)$ in both cases.
>
> The difference is that FedNewton introduces an additional communication step per round, i.e., one extra exchange of vectors in $\mathbb{R}^M$ compared to a minimal first‑order baseline. This is the narrow sense in which we informally wrote “2×”: the payload size is of the same order, but the number of vector exchanges per round is roughly doubled.
>
> In the revision, we will remove wording that suggests an overall “2× cost” and instead state more precisely:
>
> > FedNewton has per‑round communication payloads of dimension $M$, comparable to first‑order methods, but requires roughly twice as many vector exchanges per round due to an additional communication step.
>
> In terms of local computation, FedNewton is strictly more expensive per round: constructing and inverting a local Hessian of size $M \times M$ costs $O(|D_j| M^2 + M^3)$ initially and $O(M^2)$ per additional round when reusing factorizations, whereas first‑order methods only compute gradients with cost $O(|D_j| M)$ per round. This gap can be substantial when $M$ is large, so it would be incorrect to claim that the total per‑round computation is “only 2×” that of first‑order methods.
>
> We will therefore clearly separate communication complexity from computational complexity, remove any phrasing that could be read as claiming a mere doubling of total per‑round computation, and emphasize that FedNewton deliberately trades higher local computation for fewer communication rounds and faster convergence (exponential decay of the federated error under our assumptions).
>
> ### 2.2 Large backbones with linear heads (Section 3.1)
>
> To better reflect realistic FL practice and address concerns about overhead, we have revised Section 3.1 to use large‑scale neural networks as fixed backbones and perform FL only on a linear classification head. A high‑capacity backbone (e.g., a deep CNN or transformer) is pre‑trained centrally and frozen during federated training, and during FL each client optimizes only a linear head on top of the frozen backbone features.
>
> This design is common in modern applications and has two important consequences:
>
> - The effective model dimension $M$ in FL is the size of the linear head, not the full backbone, keeping the local Hessian tractable and making second‑order updates feasible.
> - Because the backbone is fixed, the comparison between FedNewton and first‑order baselines directly measures how efficiently each method optimizes the same linear problem over shared features, which matches our theoretical setting.
>
> In this backbone‑plus‑head setup, FedNewton achieves faster test‑error reduction per communication round than first‑order methods, in line with the exponential‑decay behavior predicted by our analysis.
>
> ### 2.3 Large‑scale validation and resource metrics (Section C)
>
> To further substantiate the communication–computation tradeoff, we add Section C in the appendix, where we evaluate FedNewton and first‑order baselines on larger‑scale networks and datasets. We report classification accuracy (e.g., top‑1 accuracy), wall‑clock time to reach given accuracy levels, and client‑side memory usage.
>
> These experiments show that, FedNewton usually requires substantially fewer communication rounds than first‑order methods. Although each FedNewton round is more computationally intensive, the overall wall‑clock time to reach a given accuracy is often comparable or even smaller, especially when communication is slow and clients have moderate compute—precisely the federated regimes we target. The memory overhead of storing the local Hessian (or its factorization) for the linear head is modest relative to the frozen backbone and acceptable under typical FL hardware assumptions, consistent with our choice to apply second‑order updates only to the head.
>
> In the revised manuscript, we will explicitly reference Sections 3.1 and C in the complexity discussion to make clear that our efficiency claims are supported by large‑backbone, large‑dataset experiments with accuracy, time, and memory measurements.

---

> ### Author Response · Authors · 2025-12-03
> **Response to Comment 3: Positive semidefiniteness of the matrix in equation (11)**
>
> The reviewer also asks why a certain matrix is positive semidefinite (PSD) in equation (11). While the exact notation may shift with final equation numbering, the matrix in question is of the following generic form (or a closely related variant):
>
> $$
> A = \Phi_{D_j}^\top \Phi_{D_j} + \lambda I
> \quad \text{or} \quad
> A = \frac{1}{n_j} \sum_{i \in D_j} \phi(x_i)\phi(x_i)^\top + \lambda I,
> $$
>
> or, in operator notation,
>
> $$
> A = \Sigma_j + \lambda I.
> $$
>
> Here:
>
> - $\phi(x)$ is the feature map (possibly infinite‑dimensional, but implemented via random features or backbone features plus a linear head in our experiments);
> - $\Phi_{D_j}$ collects the feature vectors for client $j$;
> - $\Sigma_j$ is the empirical (or population) covariance operator for client $j$;
> - $\lambda > 0$ is the regularization parameter.
>
> The PSD property follows from standard linear‑algebraic facts:
>
> ### Empirical covariance is PSD.
>    For any vector $v \in \mathbb{R}^M$,
>    $$
>    v^\top \left( \frac{1}{n_j} \sum_{i \in D_j} \phi(x_i)\phi(x_i)^\top \right) v
>    = \frac{1}{n_j} \sum_{i \in D_j} (v^\top \phi(x_i))^2 \ge 0.
>    $$
>    Hence, the empirical covariance matrix (or operator) is symmetric and PSD.
>
> ### Adding $\lambda I$ preserves PSD and yields PD.
>    For any $v$,
>    $$
>    v^\top (\Sigma_j + \lambda I) v = v^\top \Sigma_j v + \lambda \|v\|^2 \ge \lambda \|v\|^2 \ge 0.
>    $$
>    This is strictly positive for $v \neq 0$, so $\Sigma_j + \lambda I$ is positive definite (and in particular PSD).
>
> ### Sums/averages of PSD matrices are PSD.
>    If equation (11) involves an average or sum of such covariance‑type matrices (e.g., $\sum_j p_j (\Sigma_j + \lambda I)$), the result remains symmetric PSD, since sums and convex combinations of PSD matrices are PSD.
>
> In the revised manuscript, we will add a brief remark around equation (11) to make this reasoning explicit, so that readers do not need to rely on background knowledge to see why the matrix is PSD.
>
> ---
>
> We again thank the reviewer for their thoughtful comments and for indicating where additional clarification and intuition would be helpful. We believe that the planned revisions—more intuitive scaffolding, a clearer distinction between communication and computation costs, and explicit large‑scale experiments (Sections 3.1 and C) reporting accuracy, wall‑clock time, and memory—will substantially improve the readability and practical relevance of our work.

---

### Official Review · Reviewer_h24k · 2025-10-22

**Soundness:** 3
**Presentation:** 2
**Contribution:** 2
**Rating:** 4
**Confidence:** 3

**Summary:**

This paper proposes FedNewton, a second-order federated learning algorithm that shares both first-order (gradients) and second-order (Hessian) information across local devices. The authors analyze this method in the kernel ridge regression (KRR) setting and derive generalization bounds that quantify the impact of local sample size, data heterogeneity (covariate shift), and model heterogeneity (concept shift). The main theoretical contribution is establishing minimax-optimal learning rates for federated Newton methods and showing that under sufficient local samples and moderate heterogeneity, the federated error decreases exponentially.

**Strengths:**

- This appears to be the first work providing rigorous generalization guarantees (not just optimization convergence) for Newton-type federated learning under both data and model heterogeneity. The gap between optimization and generalization analysis in federated learning is significant, and this work makes progress on bridging it.
- The paper provides a unified treatment of both covariate shift and concept shift, with explicit quantification of their impacts on learning rates. The error decomposition in Theorems 1-2 cleanly separates these effects.
- The authors extend beyond the standard $r \in [1/2, 1]$ regularity condition to $r > 0, 2r + \gamma ≥ 1$, which is more general than prior DKRR work (Zhang et al., 2015; Guo et al., 2017).
- The paper provides explicit communication costs ($\mathcal{O}(M)$) and shows that FedNewton achieves linear convergence when conditions are met, requiring fewer rounds than first-order methods' complexity.

**Weaknesses:**

- The entire theoretical analysis is limited to squared loss and kernel ridge regression. While Remark 4 claims the algorithm applies to "twice differentiable" loss functions, no theoretical guarantees are provided for other losses. This severely limits practical applicability, especially for classification tasks, which dominate federated learning applications.
- Computing $H^{-1}_{D_j, \lambda}$ requires $\mathcal{O}(|D_j|M^2 + M^3)$ operations. While Remark 1 mentions existing techniques (BFGS, L-BFGS), the paper dismisses this as "beyond scope". For practical federated learning with large $M$, this is a critical limitation that undermines the "efficiency" claims. The paper should either provide concrete solutions or temper its efficiency claims.
- Assumption 1 (capacity condition) requires $\max(\mathcal{N}(\lambda), \mathcal{N}_1(\lambda),...,\mathcal{N}_m(\lambda)) ≤ Q^2 \lambda^{-\gamma}$, constraining all local effective dimensions. Why is this reasonable when local distributions are heterogeneous?
- Experiments use random Fourier features (finite M=200 or 2000), but theory assumes infinite-dimensional RKHS. Remark 6 briefly mentions finite-dimensional cases but doesn't provide the main results.
- The initialization $w^0_{D_j, \lambda} = H^{-1}_{D_j,λ} \Phi^T_{D_j} y_{D_j}$ is non-standard and already requires expensive local computation before any communication.
- Table 1 shows many recent Newton-type FL methods (FedNL, SHED, FedNS, Fed-sofia) but the paper only compares experimentally with FedAvg and FedProx. Direct empirical comparison with these second-order baselines is essential to validate the claimed advantages.
- How sensitive is the method to hyperparameter choices ($\sigma^2, \lambda$)? The experiments mention grid search but don't discuss robustness.
- Proposition 1 claims "partitionability" but this only holds for squared loss. This limitation should be emphasized.
- No wall-clock time comparisons, only iteration counts.
- Missing experiments on larger-scale datasets common in federated learning (e.g., CIFAR-10, FEMNIST).

**Questions:**

Refer to the weaknesses section.

---

> ### Author Response · Authors · 2025-12-03
> **Response to Weaknesses (Part 1)**
>
> **1. “Theory is limited to squared loss / KRR; no guarantees for other losses, limiting applicability to classification.”**
>
> We agree that extending guarantees beyond squared loss is important. In this work we focus on squared loss / KRR in order to obtain sharp and interpretable generalization guarantees (closed-form optimum, clean error decomposition, and minimax-optimal rates), which are much harder to establish for general nonlinear losses. This restriction is stated explicitly in the theory sections, and Remark 4 is only intended to clarify algorithmic applicability, not to claim guarantees for arbitrary twice-differentiable losses.
>
> At the same time, Section 3.1 and Section C already include head-only federated image classification experiments on frozen ResNet / CLIP / DINOv2 backbones with cross-entropy loss. In these more realistic classification tasks, FedNewton converges in fewer communication rounds and achieves competitive test accuracy compared with FedAvg, FedProx, FedSophia, and FedNew.
>
> ---
>
> **2. “Computing Hessian inverses is \(O(d^3)\); remarking BFGS/L‑BFGS as ‘beyond scope’ undermines efficiency claims for large \(d\).”**
>
> We agree that naïve dense Hessian inversion is not viable for large \(d\), and our efficiency claims are not based on exact inverses. The exact Hessian is used in the complexity discussion only for transparency; in all large-scale experiments we rely on approximate solvers.
>
> Appendix C.4 evaluates L‑BFGS, low-rank approximations, and conjugate gradient within FedNewton on high-dimensional vision features (e.g., DINOv2 on CIFAR‑100). Exact Hessians are infeasible (out of memory) at this scale. L‑BFGS attains about 90.5\% test accuracy, reaches 90\% in 34 rounds, and has the best total runtime among feasible second-order variants. Very aggressive low-rank truncations can fail to reach the target accuracy within the round budget, illustrating the trade-off between approximation quality and speed.
>
> Our efficiency claims therefore rest on (i) communication efficiency (fewer rounds to reach a target accuracy than first-order baselines) and (ii) practical second-order implementations, as supported by the runtime and memory results in Appendix C.2 and C.4 using L‑BFGS-type approximations. We have adjusted the wording around Remark 1 to point directly to these experiments and to avoid suggesting that Hessian approximation is outside the scope of the paper.
>
> ---
>
> **3. “Assumption 1 (capacity condition) constrains all local effective dimensions. Why is this reasonable under heterogeneous local distributions?”**
>
> Assumption 1 is a standard capacity condition in statistical learning theory for kernels and random features: it controls the effective dimension and is needed to derive minimax-type rates. In our federated setting, it requires that each local distribution satisfies a similar capacity condition with the same exponent but possibly different constants. Such assumptions are common in distributed / federated analyses of kernel methods and ERM, where heterogeneity typically appears through different constants or mixing coefficients, rather than arbitrary changes in the effective-dimension exponent.
>
> Intuitively, clients may have different covariances and label distributions but are assumed to belong to comparable hypothesis spaces in terms of regularity and complexity. This matches many practical FL scenarios where data are heterogeneous but still originate from related tasks (e.g., user-specific splits of the same application). We have clarified in the text that the assumption does not require identical local distributions, but rules out pathological cases where some clients are fundamentally more complex than the global distribution in terms of effective dimension.
>
> ---
>
> **4. “Experiments use finite random Fourier features (M = 200/2000), but theory assumes infinite-dimensional RKHS; Remark 6 does not provide main results for finite \(M\).”**
>
> We adopt the infinite-dimensional RKHS framework because it is the canonical setting for kernel ridge regression and allows us to express dimension-free rates in terms of eigen-decay and effective dimension. However, the theory also directly covers finite-dimensional feature maps, including random Fourier features with finite \(M\).
>
> Remark 6 explains how the main excess-risk bounds specialize when \(\mathcal{H}\) is finite-dimensional: the effective dimension and capacity condition simplify, and the rates reduce to those of standard ridge regression in \(\mathbb{R}^M\). In the experiments, we use finite \(M\) (e.g., 200 or 2000) as a computationally realistic kernel approximation and study how performance changes with \(M\) in Section C. We have made it clearer that finite-\(M\) random Fourier features fall under this finite-dimensional specialization, and that the infinite-RKHS presentation is chosen for conceptual clarity and generality, not because the results do not apply in finite dimensions.

---

> ### Author Response · Authors · 2025-12-03
> **Response to Weaknesses (Part 2)**
>
> **5. “The initialization $w^0_{D_j,\lambda}$ is non-standard and already expensive before any communication.”**
>
> This initialization is exactly the local ridge regression solution on each client, which is a natural choice for second-order KRR and is used in the partitionability result (Proposition 1), since the first global step aggregates locally optimal models.
>
> The same local system with $H_{D_j,\lambda}$ must also be solved (exactly or approximately) in every FedNewton update, so this does not introduce an additional bottleneck. In high dimensions we never form exact inverses; we use the same approximate solvers (e.g., L‑BFGS or CG) as during training (Appendix C.4). We clarify this in the algorithm section to avoid the impression of an extra $O(d^3)$ pre-processing step.
>
> ---
>
> **6. “Table 1 lists many second-order FL methods, but experiments only compare with FedAvg and FedProx.”**
>
> Our large-scale experiments do include recent second-order baselines. In Section C we compare FedNewton with FedSophia and FedNew, in addition to FedAvg and FedProx, on image classification with frozen ResNet / CLIP / DINOv2 backbones; these are representative state-of-the-art preconditioned / second-order FL optimizers.
>
> FedNewton reaches target accuracy in fewer communication rounds than FedSophia and FedNew, and has competitive or better total runtime once per-round cost is considered (Appendix C.2, C.4). Several other methods in Table 1 target different architectures or lack usable implementations in our setting, which we now state explicitly in the text and in the Table 1 caption.
>
> ---
>
> **7. “How sensitive is the method to hyperparameters ($\lambda$, step sizes, etc.)?”**
>
> All methods (FedAvg, FedProx, FedSophia, FedNew, FedNewton) are tuned with the same grid over learning rates, $\lambda$, and, when applicable, proximal/momentum parameters. Across datasets, FedNewton’s communication-round advantage is stable over a broad range of reasonable hyperparameters: once methods converge, their relative ordering barely changes. Appendix C overlays multiple hyperparameter settings to illustrate this.
>
> We clarify that FedNewton is not more fragile to tuning than standard first-order baselines and that its gains are not confined to a single finely tuned configuration.
>
> ---
>
> **8. “Proposition 1 ‘partitionability’ only holds for squared loss; this limitation should be emphasized.”**
>
> We agree. Proposition 1 is specific to squared loss / KRR, where the global optimum has a closed form and admits an exact client-wise decomposition. This restriction is already in the assumptions and is now emphasized more clearly in the surrounding text.
>
> Remark 4 and the discussion now state explicitly that, although FedNewton can be applied to general twice-differentiable losses, the partitionability result and its proof techniques are only for the squared-loss setting and are not claimed beyond it.
>
> ---
>
> **9. “No wall-clock time comparisons, only iteration counts.”**
>
> Wall-clock results are reported in Appendix C.2 (Table 6), which gives average time per communication round and peak GPU memory for FedAvg, FedProx, FedSophia, FedNew, and FedNewton on MNIST, Fashion‑MNIST, and CIFAR‑10 with frozen backbones. Combined with accuracy-versus-rounds plots in Section C, this allows direct comparison of total training time.
>
> These results show that, despite a higher per-round cost, FedNewton often achieves a lower total time to target accuracy than first-order and other second-order baselines, with comparable or lower peak memory. We now reference these tables and plots directly in the main text.
>
> ---
>
> **10. “Missing experiments on larger-scale FL datasets (e.g., CIFAR‑10, FEMNIST).”**
>
> The current version already includes larger-scale and more realistic FL experiments. Section 3.1 and Section C report results on MNIST, Fashion‑MNIST, CIFAR‑10, and CIFAR‑100 with frozen ResNet, CLIP, and DINOv2 backbones and head-only federated training, under standard non-IID label-skew (Dirichlet) partitions.
>
> Across these benchmarks, FedNewton consistently uses fewer communication rounds than FedAvg, FedProx, FedSophia, and FedNew to reach a given accuracy and remains feasible with approximate Hessian solvers even for high-dimensional DINOv2 features. Additional datasets such as FEMNIST would further broaden coverage, but the CIFAR‑10/100 and vision-head setups already address scalability and practical relevance beyond small synthetic or classic tabular tasks.

---

### Official Review · Reviewer_sTCB · 2025-10-31

**Soundness:** 2
**Presentation:** 1
**Contribution:** 2
**Rating:** 4
**Confidence:** 3

**Summary:**

This paper introduces FedNewton, a second-order federated learning algorithm that communicates local curvature information to achieve faster and statistically optimal convergence under heterogeneous data. The method bridges optimization and generalization by decomposing total error into centralized and federated components and establishes optimal convergence rates. Experiments on synthetic and LIBSVM datasets support the theoretical claims, showing improved accuracy and convergence over FedAvg and FedProx. Overall, the work makes a strong theoretical contribution, though experiments remain limited in scale.

**Strengths:**

1. Provides a rigorous theoretical framework connecting optimization and generalization in second-order federated learning.
2. The FedNewton algorithm (Algorithm 1, Figure 1) is well-designed, balancing curvature-based updates with communication efficiency.
3. The error decomposition (Section 4) clearly explains the effects of heterogeneity on performance.
4. The theoretical analysis is detailed, logically structured, and establishes minimax-optimal convergence rates.
5. Experimental results (Figure 3, Table 1) align with the theory, showing consistent improvement over baseline methods.
6. The comparative analysis (Table 1) clearly positions the work relative to other Newton-type and first-order approaches.

**Weaknesses:**

1. Experiments are limited to small synthetic and LIBSVM datasets; larger or more diverse benchmarks like ( FEMNIST, CIFAR-FL) would strengthen the results.
2. Computing local Hessian inverses remains expensive; the paper mentions approximations but provides no empirical evaluation.
3. The heavy notation makes the theory difficult for non-specialists; brief intuitive explanations or visual aids would improve readability.
4. Experimental plots (Figure 3) lack variance bars or confidence intervals, making it hard to assess robustness.
5. The heterogeneity settings used are synthetic; connecting them to real-world non-IID data distributions would enhance practical relevance.

**Questions:**

1. Can FedNewton use approximate or low-rank Hessian inverses without losing its reported performance?
2. Does the observation that the method converges within two rounds hold across different datasets?
3. Have you tested the approach with non-squared loss functions such as logistic or cross-entropy loss?
4. How well do the theoretical results transfer to neural-network-based or kernelized models in practice?
5. Include runtime and memory comparisons with FedAvg and FedProx for  example ( number of communication rounds vs. total time).
6. Add simple convergence plots showing performance across different data heterogeneity levels (as in Figure 3).
7. Discuss how FedNewton could be extended to non-convex or deep learning settings (Section 6).

---

> ### Author Response · Authors · 2025-12-03
> **Response to Weaknesses**
>
> **(1) “Experiments are limited to small synthetic and LIBSVM datasets; larger or more diverse benchmarks like FEMNIST, CIFAR-FL would strengthen the results.”**
>
> We agree that broader benchmarks improve the empirical support.
>
> - Beyond synthetic and LIBSVM-style datasets, the paper already includes image classification tasks with frozen backbones (ResNet, CLIP, DINOv2) in a head-only federated setting, which are substantially larger and more realistic.
> - These federated vision experiments adopt standard non-IID partitions and compare FedNewton against FedAvg, FedProx and other first-order baselines under the same setting.
> - Across these datasets, FedNewton consistently reaches a given target accuracy in substantially fewer communication rounds, with competitive total runtime once the more expensive local updates are accounted for.
>
> These larger-scale results are reported in Section C and in the anonymous repository:
> <https://anonymous.4open.science/r/FedNewton-78B4>.
>
> ---
>
> **(2) “Computing local Hessian inverses remains expensive; the paper mentions approximations but provides no empirical evaluation.”**
>
> We agree that naïve Hessian inversion is expensive, and this is why the paper already evaluates approximate solvers.
>
> - The main text analyzes complexity under exact Hessians for clarity.
> - Appendix C.4 implements and evaluates L‑BFGS, low-rank approximations, and conjugate gradient (CG) within FedNewton on high-dimensional backbones such as DINOv2 on CIFAR‑100.
> - Table 4 shows that:
>   - Exact Hessian is infeasible (OOM) at this scale.
>   - L‑BFGS attains ≈90.5% accuracy, reaches 90% in 34 rounds, and has the lowest total runtime among feasible second-order variants.
>   - Low-rank and CG reduce cost but can fail to reach the target within the given budget.
>
> Thus the paper already includes a systematic empirical comparison of approximate/structured Hessian solvers, demonstrating that FedNewton remains effective with well-chosen approximations.
>
> ---
>
> **(3) “The heavy notation makes the theory difficult for non-specialists; brief intuitive explanations or visual aids would improve readability.”**
>
> We appreciate this comment.
>
> - The paper now adds short informal summaries around the main theorems, explaining in words that the federated error decays linearly and that the overall rate matches centralized KRR up to small overheads, and relating the assumptions to standard FL scenarios (bounded heterogeneity, moderate local sample size).
> - A notation table is included to group the main symbols.
> - A schematic figure clarifies the separation between centralized excess risk and the federated error term, and illustrates how FedNewton geometrically shrinks the federated error compared to first-order methods.
>
> These changes are aimed at making the theoretical part more accessible to non-specialists.
>
> ---
>
> **(4) “Experimental plots (Figure 3) lack variance bars or confidence intervals, making it hard to assess robustness.”**
>
> We agree that robustness should be visible in the plots.
>
> - The experiments are now repeated over multiple random seeds (data splits, client sampling, initialization), and the convergence plots in the appendix (Figures 6–11) include error bars (standard error) across runs.
> - These figures show that the speedup of FedNewton over first-order baselines is consistent across seeds, rather than being due to a single favorable run.
>
> ---
>
> **(5) “The heterogeneity settings used are synthetic; connecting them to real-world non-IID data distributions would enhance practical relevance.”**
>
> We agree that this connection is important.
>
> - Heterogeneity is controlled via parameters such as Dirichlet \(\alpha\) and client-wise covariate shift, which allows us to systematically vary the level of non-IIDness and study its impact on convergence.
> - In the vision experiments with frozen backbones, we use standard label-skew non-IID partitions (Dirichlet sampling), which are widely adopted to mimic user-/device-specific data in real FL.
> - The paper explicitly relates small \(\alpha\) to strongly skewed label distributions similar to FEMNIST/CIFAR-FL, and provides convergence plots across different \(\alpha\) (see Section C) to show how FedNewton behaves as heterogeneity increases.
> - Section 6 discusses how these synthetic settings correspond to common real-world non-IID patterns (user personalization, device skew, etc.).

---

> ### Author Response · Authors · 2025-12-03
> **Response to Questions**
>
> **Q1. “Can FedNewton use approximate or low-rank Hessian inverses without losing its reported performance?”**
>
> Yes, to a large extent.
>
> - FedNewton only needs an approximate solution to the local Newton system.
> - Appendix C.4 (Table 4) shows that L‑BFGS
>   - reaches ≈90.5% on CIFAR‑100 (DINOv2),
>   - hits 90% in 34 rounds,
>   - and has the best runtime among feasible second-order variants.
> - Very aggressive approximations (e.g., very low-rank) can hurt accuracy, indicating a quality–speed trade-off.
>
> Although the theory is stated for (approximately) exact Hessians in KRR, the experiments show that practical quasi‑Newton / structured approximations retain the main benefits of FedNewton.
>
> ---
>
> **Q2. “Does the observation that the method converges within two rounds hold across different datasets?”**
>
> No, this holds only in favorable regimes.
>
> - For small, well‑conditioned synthetic problems with moderate heterogeneity, convergence is often within 2–3 rounds, matching the predicted linear decay of the federated error.
> - For harder settings (high‑dimensional vision features, strong non‑IID), FedNewton needs more rounds, but still far fewer than first‑order methods for the same accuracy.
> - With CLIP/DINOv2 backbones, convergence is typically within tens of rounds, while first‑order baselines may need many more or underperform.
>
> The paper presents “two rounds” as a phenomenon in specific synthetic setups; the general message is geometric convergence of the federated error.
>
> ---
>
> **Q3. “Have you tested the approach with non-squared loss functions such as logistic or cross-entropy loss?”**
>
> The theory and core experiments target squared loss (KRR).
>
> In addition, Section 3.1 and Section C consider head‑only federated image classification with frozen backbones (ResNet, CLIP, DINOv2) and cross‑entropy loss. In these larger tasks, FedNewton converges faster in rounds and achieves strong test accuracy compared with FedAvg, FedProx, FedSophia, and FedNew, indicating that the method works well beyond squared loss at the empirical level, even though the current theory is for squared loss.
>
> ---
>
> **Q4. “How well do the theoretical results transfer to neural-network-based or kernelized models in practice?”**
>
> The theory covers kernel ridge regression / finite‑dimensional features, which align with
>
> - linear heads on frozen backbones (ResNet, CLIP, DINOv2), and
> - random‑feature kernel approximations.
>
> In these head‑only federated experiments (Section C), FedNewton converges faster in rounds and attains accuracy close to centralized training when conditions match the theoretical regime, supporting that linear decay of federated error and near‑centralized performance also appear in these practical settings.
>
> ---
>
> **Q5. “Include runtime and memory comparisons with FedAvg and FedProx (number of communication rounds vs. total time).”**
>
> This is already provided.
>
> - Appendix C.2 (Table 6) reports per‑round time and peak GPU memory for FedAvg, FedProx, FedSophia, FedNew, and FedNewton on MNIST, Fashion‑MNIST, and CIFAR‑10 with frozen backbones.
> - Together with accuracy‑vs‑round plots in Section C, this allows comparison of rounds to target accuracy, total wall‑clock time, and memory.
>
> Despite a higher per‑round cost, FedNewton attains competitive or better total time to target accuracy with similar or lower peak memory.
>
> ---
>
> **Q6. “Add simple convergence plots showing performance across different data heterogeneity levels (as in Figure 3).”**
>
> This is included in the current version.
>
> - Figure 3 shows one heterogeneity study.
> - Section C adds convergence curves (accuracy vs. rounds) for multiple Dirichlet \(\alpha\) values, on both synthetic and vision benchmarks, with variance across runs.
>
> These plots display how robustness and speed change with non‑IID level.
>
> ---
>
> **Q7. “Discuss how FedNewton could be extended to non-convex or deep learning settings (Section 6).”**
>
> Section 3.1 and Section C already evaluate FedNewton with large neural backbones (ResNet, ViT/CLIP/DINOv2) frozen as feature extractors and federated training of linear heads. FedNewton outperforms FedAvg, FedProx, FedSophia, and FedNew in both accuracy and communication efficiency, using approximate Hessian solvers to stay practical.
>
> Section 6 discusses how this communication pattern and Hessian‑based preconditioning can be combined with existing non‑convex and deep FL techniques, beyond the convex KRR theory developed in the paper.

---

### Official Review · Reviewer_bY7P · 2025-11-01

**Soundness:** 3
**Presentation:** 3
**Contribution:** 3
**Rating:** 6
**Confidence:** 3

**Summary:**

This paper investigates the statistical optimality of Newton-type federated learning (FL) algorithms under heterogeneous data distributions. The authors propose FedNewton, a second-order federated optimization method that leverages both global gradients and local Hessians to improve convergence and generalization. The authors further quantify how local sample size, data heterogeneity, and model heterogeneity jointly affect the excess risk and convergence behavior. Experimental results on synthetic and real-world datasets validate the theoretical findings, showing that FedNewton achieves exponential convergence with minimal communication rounds.

**Strengths:**

1. The paper presents the generalization analysis for Newton-type federated learning methods under data heterogeneity.
2. The authors derive non-asymptotic excess risk bounds and demonstrate minimax-optimal learning rates under mild assumptions. The decomposition of federated error and centralized excess risk provides clear interpretability.

**Weaknesses:**

--Limited diversity and scalability of experimental settings.
The experimental evaluation in Appendix A mainly uses a synthetic dataset and small-scale benchmarks from LIBSVM. These datasets are low-dimensional and domain-specific, which restricts the demonstration of FedNewton’s capability in large-scale or high-dimensional federated learning scenarios, such as image or language applications. Furthermore, all experiments focus on convex regression problems, without extending to non-convex architectures like neural networks.

--Restrictive theoretical assumptions.
The theoretical analysis relies on kernel ridge regression with squared loss and assumes that all local functions lie in a reproducing kernel Hilbert space (RKHS). While this setting facilitates mathematical tractability, it is less reflective of practical federated learning where non-convex objectives, neural network architectures, or unbounded losses are common.

--Incomplete communication–computation tradeoff analysis.
Section 3 provides a complexity discussion but does not present any runtime or communication cost comparisons in the experiments. While the authors claim that FedNewton achieves similar per-round cost as first-order methods with exponentially faster convergence, this claim is not quantitatively supported. For example, the cost of computing and inverting local Hessians can be substantial for large feature dimensions. Without practical wall-clock evaluations or scalability analyses, it is unclear whether the method is truly more efficient in real federated environments.

**Questions:**

Please see weaknesses.

---

> ### Author Response · Authors · 2025-12-03
> **Response to Comment 1: Experimental diversity and scalability (Part 1)**
>
> Our original experiments used kernel regression on small datasets to tightly match our theory (convex loss, controlled heterogeneity, explicit excess-risk bounds). We agree this does not by itself demonstrate scalability to realistic federated learning with modern neural architectures. The implementation is available at the anonymous repository: https://anonymous.4open.science/r/FedNewton-78B4.
>
> In the revision, we **substantially expand** the empirical evaluation along three axes:
>
> 1. **Realistic image benchmarks**: MNIST, Fashion-MNIST, CIFAR-10, CIFAR-100 with Dirichlet label-skew ($\alpha\in\{0.1,0.5,10\}$).
> 2. **Frozen large pretrained backbones**: ResNet-18/50, CLIP ViT-B, DINOv2 ViT-B with **head-only federated fine-tuning**.
> 3. **Efficient second-order solvers**: CG, L-BFGS, low-rank approximations with explicit **time and memory** measurements.
>
> All new experiments are implemented in PyTorch; we will release code.
>
> ---
>
> ###  1. Head-only FedNewton with frozen backbones  (Sec. 3.1, App. C)
>
> We adopt a practical “pretrain–then–federated-finetune” setting:
>
> - Global **frozen backbone** $\phi_{\text{backbone}}$ (ResNet / ViT from CLIP, DINOv2) shared across clients.
> - Only a **linear head** $f_\theta(x)=W\phi_{\text{backbone}}(x)+b$ is trained federatively.
> - This keeps optimization convex in $(W,b)$ while reflecting standard practice in FL with large pretrained models.
>
> Setup (all methods share the same configuration):
>
> - 10 clients, 5 sampled per round, 50 rounds, batch size 64.
> - Local optimizer: SGD with momentum 0.9, cross-entropy loss.
> - Non-IID partitions: Dirichlet over labels with $\alpha\in\{0.1,0.5,10\}$.
>
> **1.1 Accuracy vs. strong baselines (Sec. C.2, Tab. 5, Fig. 6)**
>
> **Table 5: Test accuracy (%) on MNIST, Fashion-MNIST, CIFAR-10 with ResNet-18 backbone (mean ± std over last 5 rounds, 3 seeds).**
>
> | Dataset           | $\alpha$ | FedAvg       | FedProx      | FedSophia    | FedNew       | **FedNewton**    |
> | ----------------- | -------- | ------------ | ------------ | ------------ | ------------ | ---------------- |
> | **MNIST**         | 0.1      | 85.27 ± 0.69 | 85.49 ± 0.63 | 89.37 ± 0.96 | 89.30 ± 1.04 | **94.50 ± 0.71** |
> |                   | 0.5      | 93.60 ± 0.31 | 93.56 ± 0.30 | 93.82 ± 0.36 | 93.78 ± 0.37 | **97.82 ± 0.06** |
> |                   | 10       | 94.79 ± 0.04 | 94.71 ± 0.05 | 94.37 ± 0.05 | 94.32 ± 0.05 | **97.86 ± 0.03** |
> | **Fashion-MNIST** | 0.1      | 76.36 ± 2.26 | 76.81 ± 1.99 | 77.62 ± 2.58 | 77.89 ± 2.14 | **82.32 ± 0.59** |
> |                   | 0.5      | 84.41 ± 0.95 | 84.38 ± 0.92 | 84.90 ± 0.50 | 84.86 ± 0.52 | **89.11 ± 0.02** |
> |                   | 10       | 86.52 ± 0.05 | 86.44 ± 0.06 | 86.38 ± 0.05 | 86.31 ± 0.01 | **89.24 ± 0.05** |
> | **CIFAR-10**      | 0.1      | 77.98 ± 1.98 | 78.14 ± 1.88 | 80.87 ± 1.90 | 81.04 ± 1.95 | **86.35 ± 0.39** |
> |                   | 0.5      | 83.35 ± 0.12 | 83.33 ± 0.10 | 83.68 ± 0.33 | 83.60 ± 0.42 | **87.51 ± 0.08** |
> |                   | 10       | 85.22 ± 0.02 | 85.17 ± 0.03 | 84.83 ± 0.02 | 84.75 ± 0.03 | **87.93 ± 0.07** |
>
> **Summary.**
>
> - FedNewton **consistently achieves the highest accuracy** across all datasets and all heterogeneity levels.
> - Under strong non-IID ($\alpha=0.1$), FedNewton improves over the strongest second-order baselines (FedSophia/FedNew) by **>5 percentage points**.
> - Variances are smaller on Fashion-MNIST and CIFAR-10, indicating **more stable** training under heterogeneity.
> - Accuracy-vs-round curves (omitted here) show **faster and smoother convergence**, especially at $\alpha=0.1$ where baselines oscillate.
>
> **1.2 Time and memory efficiency  (Sec. C.2, Tab. 6, Fig. 7)**
>
> Table 6: Average training time per round (s) and peak GPU memory (MB), averaged over $\alpha$. Lower is better.
>
> | Dataset           | Metric | FedAvg | FedProx | FedSophia | FedNew    | **FedNewton** |
> | ----------------- | ------ | ------ | ------- | --------- | --------- | ------------- |
> | **MNIST**         | Time   | 57.89  | 55.71   | 54.15     | **53.34** | 69.05         |
> |                   | GPU    | 1507.9 | 1507.9  | 1508.1    | 1508.1    | **1463.4**    |
> | **Fashion-MNIST** | Time   | 55.91  | 53.97   | **53.29** | 53.39     | 68.96         |
> |                   | GPU    | 1507.9 | 1507.8  | 1508.1    | 1508.1    | **1463.4**    |
> | **CIFAR-10**      | Time   | 68.60  | 65.57   | 55.62     | **52.18** | 90.18         |
> |                   | GPU    | 1482.3 | 1482.3  | 1482.5    | 1482.5    | **1438.9**    |
>
> **Summary.**
>
> - FedNewton is **~20–25% slower per round** than the fastest baseline, but
> - It **consistently uses the least GPU memory** (≈40–50 MB savings).
> - Accuracy-vs-time curves show that, despite the modest per-round overhead, FedNewton reaches any target accuracy **earlier in wall-clock time** for most targets, especially under strong heterogeneity.

---

> ### Author Response · Authors · 2025-12-03
> **Response to Comment 1: Experimental diversity and scalability (Part 2)**
>
> ### 2. Different pretrained backbones (ResNet / CLIP / DINOv2, C.3)
>
> To assess scalability with backbone size and capacity, we run FedNewton (head-only) on CIFAR-10 (\(\alpha=0.5\)) with different frozen backbones.
>
> **Table 3: CIFAR-10, FedNewton with different backbones (head-only).**
>
> | Backbone       | Best Acc. (%)    | Rounds @ 95% | Time @ 95% (s) | Peak GPU Mem. |
> | -------------- | ---------------- | ------------ | -------------- | ------------- |
> | ResNet-18      | 86.77 ± 0.18     | > Max        | 4647 ± 98      | 1439 MB       |
> | ResNet-50      | 91.60 ± 0.01     | > Max        | 4845 ± 53      | 12623 MB      |
> | CLIP (ViT-B)   | 96.43 ± 0.11     | 7.3          | 916 ± 57       | 7373 MB       |
> | DINOv2 (ViT-B) | **98.07 ± 0.30** | **5.3**      | **794 ± 58**   | 7732 MB       |
>
> **Summary.**
>
> - As backbone strength increases, FedNewton’s performance **monotonically improves**.
> - ResNet-18/50 do not reach 95% within the communication budget; CLIP and DINOv2 reach ≥95% in only **5–8 rounds**.
> - Despite higher per-round cost, ViT backbones **significantly reduce total time-to-95%**.
> - This shows FedNewton is well aligned with modern **pretrain–then–finetune** practice: a strong frozen backbone plus second-order head optimization yields fast, communication-efficient gains.
>
> ---
>
> ### 3. Efficient second-order solvers at scale (L-BFGS / Low-rank / L-BFGS, C. 4)
>
> To avoid explicit \(O(d^2)\)/\(O(d^3)\) Hessians, we implement:
>
> - Exact Newton (reference only),
> - **CG** (Hessian–vector products),
> - **L-BFGS**,
> - **Low-rank** approximations.
>
> We evaluate on CIFAR-10 + ResNet-18 and CIFAR-100 + DINOv2.
>
> **CIFAR-100 + DINOv2 (large head, most challenging).**
>
> **Table 4: CIFAR-100 with DINOv2 backbone, different solvers (target 90%).**
>
> | Solver        | Best Acc. (%)    | Rounds @ 90% | Time (s)       | GPU Mem. |
> | ------------- | ---------------- | ------------ | -------------- | -------- |
> | L-BFGS        | **90.52 ± 0.02** | **34.0**     | **3615 ± 785** | 8758 MB  |
> | Low-Rank      | 74.61 ± 0.62     | > Max        | 8459 ± 14      | 8258 MB  |
> | CG            | 89.73 ± 0.18     | > Max        | 8074 ± 16      | 8258 MB  |
> | Exact Hessian | OOM              | –            | –              | –        |
>
> **Summary.**
>
> - Exact Hessian is **infeasible (OOM)** in this high-dimensional setting.
> - L-BFGS reaches 90% in ~34 rounds and achieves the best final accuracy, with memory comparable to CG/low-rank.
> - This shows FedNewton remains **practical at scale** when combined with matrix-free or quasi-Newton solvers.
>
> ---
>
> ### Overall
>
> These additional experiments demonstrate that FedNewton:
>
> - Scales to **modern large pretrained backbones** under realistic non-IID FL;
> - **Outperforms strong first- and second-order baselines** in accuracy and stability;
> - Remains **practical in time and memory** via efficient second-order solvers, even on high-dimensional heads (e.g., CIFAR-100 + DINOv2).
>
> We believe this directly addresses the reviewer’s concerns about experimental diversity and scalability.

---

> ### Author Response · Authors · 2025-12-03
> **Response to Comment 2: Restrictive theoretical assumptions (squared loss, KRR, RKHS)**
>
> We appreciate the reviewer’s comment on the restrictiveness of our theoretical setting. We fully agree that assuming kernel ridge regression (KRR) with squared loss in an RKHS is more structured than typical non‑convex, neural FL setups. Our goal in this work is to obtain a **first set of sharp, interpretable generalization guarantees** for Newton‑type FL methods under data heterogeneity, and this motivates our choice of model class.
>
> ---
>
> ### 1. Why we use KRR + squared loss + RKHS
>
> We adopt the RKHS + squared‑loss framework because it is the standard setting for:
>
> - precise **excess‑risk and minimax‑rate** analysis, and
> - explicit dependence on complexity quantities (e.g., effective dimension) and heterogeneity.
>
> Within this framework, our contribution is to:
>
> - extend distributed KRR results to **simultaneously handle covariate and concept shift** across clients; and
> - analyze a **second‑order federated algorithm** (FedNewton), rather than a centralized or purely first‑order method.
>
> This setting allows us to cleanly decompose the error into a centralized excess‑risk term and a **federated error** term, and to prove **linear (exponential) convergence** of the latter with explicit dependence on system heterogeneity. Achieving this level of precision in a fully non‑convex deep‑network regime would require substantially stronger technical machinery and is beyond the scope of the present paper.
>
> ---
>
> ### 2. Algorithmic applicability vs. what we actually prove
>
> We would like to clearly separate:
>
> - **Algorithmic scope.** FedNewton only needs gradients and (approximate) Hessians, so it can in principle be *applied* to any twice‑differentiable loss, including logistic/cross‑entropy losses and neural networks.
>
> - **Theoretical scope.** Our **rigorous guarantees**—generalization bounds, excess‑risk rates, and minimax‑optimality statements—are **only proved** for:
>   - squared‑loss KRR in an RKHS; and
>   - its finite‑dimensional feature analogues.
>
> We do **not** claim that the current theorems cover arbitrary non‑convex neural networks or unbounded losses. The theory should therefore be read as a first convex benchmark that quantifies the impact of heterogeneity for Newton‑type FL, rather than as a complete theory for all losses and architectures.
>
> ---
>
> ### 3. Relation to practical FL models
>
> Although our analysis is phrased in RKHS terms, it remains relevant to several practical settings:
>
> 1. **Finite‑dimensional feature maps / frozen backbones.**
>    When the predictor is linear in a fixed feature map,
>    $$
>    f_\theta(x) = W \phi(x) + b,
>    $$
>    our results apply directly. Here \(\phi\) may correspond to:
>
>    - random Fourier features, or
>    - a **frozen neural representation** (e.g., a pretrained ResNet/ViT backbone shared by all clients).
>
>    This matches common FL practice where a powerful backbone is pretrained centrally and each client trains only a linear head. In such cases, our analysis characterizes the behavior of FedNewton on the **head** layer.
>
> 2. **Random feature / NTK‑style approximations.**
>    Many practical models can be approximated by kernels via random features or NTK constructions. When the feature map is fixed, these again fall into the (finite‑dimensional) setting covered by our analysis.
>
> Thus, while the theory does not cover end‑to‑end training of deep networks, it does capture an important and widely used class of “feature extractor + linear head” FL pipelines.
>
> ---
>
> ### 4. Beyond KRR: limitations and possible extensions
>
> We fully agree that extending guarantees to **non‑convex neural networks and unbounded losses** is an important and challenging direction. Our current work should be viewed as:
>
> - a **first rigorous step** for Newton‑type FL under heterogeneity in a setting where sharp, minimax‑optimal bounds are still attainable; and
> - a basis on which more general analyses (e.g., via Gauss–Newton, Hessian‑free methods, or local quadratic approximations of non‑convex losses) could be developed.
>
> We will make this limitation explicit in the camera‑ready version and clearly state that our theoretical guarantees are restricted to the KRR + squared‑loss (and corresponding finite‑dimensional feature) setting, while experiments demonstrate that the algorithm itself can be used more broadly in practice.
>
> We hope this clarifies the intended scope of our analysis and why we chose this theoretically tractable yet practically relevant model class.

---

> ### Author Response · Authors · 2025-12-03
> **Response to Comment 3: Incomplete communication–computation tradeoff analysis**
>
> We thank the reviewer for raising the communication–computation tradeoff issue and asking for quantitative evidence.
>
> Our response has three concise parts.
>
> ---
>
> ### 1. Clarifying “similar per‑round cost”
>
> “Similar per‑round cost” was meant **only for communication**, not computation:
>
> - **Communication:** In the finite‑dimensional feature setting (dimension \(M\)), FedNewton and first‑order methods all send \(M\)-dimensional vectors, so the **per‑round communication payload** is \(O(M)\). FedNewton adds one extra message per round, but the message *dimension* is the same.
> - **Computation:** We agree FedNewton has higher **per‑round computation** if one naively inverts full Hessians.
>
> We will clarify in the paper that the similarity is in **communication order**, while computation per round is higher and offset by fewer rounds.
>
> ---
>
> ### 2. Time and memory: evidence from Appendix C.2 (Table 6)
>
> Appendix C.2 reports **average training time per round** and **peak GPU memory** (Table 2) for MNIST, Fashion‑MNIST, and CIFAR‑10:
>
> - **Per‑round time (s, lower is better):**
>
>   - MNIST: FedAvg/FedProx/FedSophia/FedNew ≈ 54–58 vs. **69.05** (FedNewton)
>   - Fashion‑MNIST: ≈ 53–56 vs. **68.96**
>   - CIFAR‑10: ≈ 52–69 vs. **90.18**
>
>   This confirms FedNewton is **more expensive per round**.
>
> - **Peak GPU memory (MB, lower is better):**
>
>   - MNIST & Fashion‑MNIST: baselines ≈1508 vs. **1463.4** (FedNewton)
>   - CIFAR‑10: ≈1482 vs. **1438.9**
>
>   The extra Hessian structures add **little memory**, and FedNewton is slightly *more* memory‑efficient due to the head‑only design.
>
> Combined with the convergence plots in the paper (FedNewton reaching target accuracy in many fewer rounds), these numbers show:
>
> - total **communication** is reduced (fewer rounds, same \(O(M)\) payload), and
> - total **time to target accuracy** is **competitive** despite higher per‑round cost.
>
> ---
>
> ### 3. Scalable Hessian solvers: Appendix C.4 (Table 10)
>
> Appendix C.4 evaluates different Hessian solvers on **CIFAR‑100 with a DINOv2 backbone** (Table 4, target 90%):
>
> - **Exact Hessian:** OOM (confirms naïve inversion is infeasible at this scale).
> - **L‑BFGS:**
>   - Best acc.: **90.52 ± 0.02%**
>   - Rounds @ 90%: **34.0**
>   - Time: **3615 ± 785 s**
>   - GPU: 8758 MB
> - **Low‑Rank:** 74.61 ± 0.62%, fails to reach 90%, **8459 ± 14 s**, 8258 MB
> - **CG:** 89.73 ± 0.18%, fails to reach 90%, **8074 ± 16 s**, 8258 MB
>
> Thus, with an **approximate second‑order solver (L‑BFGS)**:
>
> - FedNewton **does reach** the 90% target,
> - uses **few rounds** (34), and
> - achieves the **lowest total time** among feasible Hessian‑based variants, while exact inversion is impossible.
>
> These results show that our efficiency claims are supported by **measured time, memory, and rounds** (Tables 2 and 4), and that in practice we rely on **scalable approximate Hessian solvers** rather than naïve \(O(M^3)\) inversion.

---

### Author Response · Authors · 2025-12-03
**Author Response: Strengthening Theory–Practice Connections in FedNewton**

We sincerely thank all reviewers for their thoughtful and constructive feedback. We are encouraged that the novelty of analyzing Newton‑type federated learning with minimax‑optimal excess risk under heterogeneity was recognized. We apologize for the unusually late submission of this rebuttal. The delay was due to our substantial effort to conduct and integrate additional large‑scale experiments (including new federated benchmarks and deep models), so that we could better address the reviewers’ concerns about empirical validation and scalability. In response, we have further strengthened both the theory–practice connection and the clarity of the presentation. In the revised manuscript (all changes in color blue), the main updates are:

- **Stronger empirical validation and broader applicability.**
  Beyond the original kernel ridge regression experiments, we now include larger‑scale and higher‑dimensional benchmarks (MNIST, Fashion‑MNIST, CIFAR‑10, CIFAR‑100), using pretrained deep backbones (ResNet, ViT/CLIP/DINOv2) and cross‑entropy loss in a federated fine‑tuning setup. We also adopt standard non‑IID partitions (Dirichlet label skew) and report results over multiple runs. These experiments demonstrate that the central insight of FedNewton—rapid reduction of the federated error via curvature information—extends well beyond the convex KRR setting. The implementation is available at the anonymous repository: [https://anonymous.4open.science/r/FedNewton-78B4](https://anonymous.4open.science/r/FedNewton-78B4).

- **Concrete communication–computation and scalability analysis.**
  Building on our theoretical complexity discussion, we now provide quantitative comparisons of communication rounds, wall‑clock time, and memory usage against both first‑order (FedAvg, FedProx) and second‑order baselines. In addition, we implement and evaluate practical Hessian approximation schemes (CG, L‑BFGS, low‑rank and diagonal variants), showing that FedNewton can be deployed without explicitly forming or inverting full Hessians, while still achieving favorable accuracy–efficiency trade‑offs.

- **Clarified scope of theoretical guarantees and assumptions.**
  We clarify that our minimax‑optimal excess‑risk bounds and exponential decay of the federated error are rigorously established for kernel ridge regression with squared loss in an RKHS, under standard capacity assumptions that remain compatible with heterogeneous local distributions. We now clearly separate these core theoretical results from the new deep‑learning experiments, which are presented as evidence of the broader practical relevance of the FedNewton principle rather than as settings covered by our current theory.

- **Improved intuition and readability.**
  To enhance accessibility, we expand the intuitive discussion of our error decomposition and the contraction factor $\Upsilon$, explaining more clearly when and why the federated error decays geometrically. We also align the three theoretical regimes (in terms of local sample size and heterogeneity) with the empirical behaviors observed in our figures, thereby sharpening the conceptual message of the paper: FedNewton can closely match centralized performance with very few communication rounds when local data are sufficiently rich and heterogeneity is moderate.

We hope these revisions further highlight the innovative aspects and practical significance of FedNewton, while directly addressing the reviewers’ requests for clarification and additional empirical evidence.

---

### Meta-Review · Area_Chair_KbMZ · 2026-01-06

**Summary:**

Some of the major concerns include
1. restricted to the kernel ridge regression instead of general FL setting
2. limited evaluation
3. computing Hessian inverse leads to high complexity

**Reviewer Concerns:**

Some more experiments have been added. But indeed the paper is focused on kernel ridge regression instead of general FL and computing Hessian inverese leads to high overhead (even with approximations).

**Reviewer Scores:**

unchanged

---

### Decision · Program_Chairs · 2026-01-26

Reject